# REALIGN: Regularized Procedure Alignment with Matching Video Embeddings via Partial Gromov-Wasserstein Optimal Transport

## Abstract

Learning from procedural videos remains a core challenge in self-supervised representation learning, as real-world instructional data often contains background segments, repeated actions, and steps presented out of order. Such variability violates the strong monotonicity assumptions underlying many alignment methods. Prior state-of-the-art approaches, such as OPEL and RGWOT, leverage Kantorovich Optimal Transport (KOT) and Gromov–Wasserstein Optimal Transport (GWOT) to build frame-to-frame correspondences, but operate only on local feature similarity and pairwise relational structure, without explicit temporal priors, which limits their ability to capture the higher-order temporal structure of a task. In this paper, we introduce **REALIGN**, an unsupervised framework for procedure learning based on *Regularized Fused Partial Gromov-Wasserstein Optimal Transport* (R-FPGWOT). In contrast to RGWOT, our formulation jointly models visual correspondences and temporal relations under a partial alignment scheme, enabling robust handling of irrelevant frames, repeated actions, and non-monotonic step orders common in instructional videos. To stabilize training, we integrate FPGWOT distances with inter-sequence contrastive learning, avoiding the need for multiple regularizers and preventing collapse to degenerate solutions. Across egocentric (EgoProceL) and third-person (ProceL, CrossTask) benchmarks, REALIGN achieves up to **18.9% (7.62pp)** average F1-score improvements and over **30% (7.74pp)** temporal IoU gains, while producing more interpretable transport maps that preserve key-step orderings and filter out noise.

## 1 Introduction

A central goal in modern AI applications-such as household robotics, augmented reality assistance, and industrial automation-is to enable agents to reliably replicate multi-step human demonstrations. Achieving this requires not only recognizing individual steps but also understanding how they form coherent procedures, such as preparing a salad (Fig. 1) with steps like peeling, chopping, and mixing. Unlike simple one-off actions, these procedures require models to reason about both sequence and structure, making the problem far more complex. Early approaches tried to solve this problem with hand-crafted rules that defined each step and its transitions. While intuitive, these rule-based systems have struggled to generalize across different domains, often breaking down when faced with visual variability, background noise, or steps appearing in unexpected orders. Real-world demonstrations are simply too diverse and messy to capture with explicit rules (for example, there are countless ways of cooking pasta or assembling furniture). This gap between rigid rules and messy real-world data is what motivates the shift toward learning-based methods. To overcome these limitations, the community has increasingly turned to *procedure learning (PL)*-the discovery of key steps and their temporal arrangement directly from raw instructional videos, without dense human supervision (Bansal et al. (2022; 2024); Elhamifar & Huynh (2020)). Large, uncurated repositories (e.g., YouTube tutorials, egocentric recordings, assembly demos) provide rich but noisy supervision (Alayrac et al. (2016); Kukleva et al. (2019)), offering both the scale and diversity needed to learn procedures in realistic settings.

Unlike short-term action recognition which focuses on isolated clips (e.g., classifying 'cutting' vs. 'stirring') (Carreira & Zisserman (2017); Simonyan & Zisserman (2014); Piergiovanni et al. (2017);

Figure 1: Key-step preparation of a salad bowl (De la Torre et al. (2009)) with alignment challenges: (a) *background* frames (gray blocks), (b) *non-monotonic* frames (curved arrows), and (c) *redundant* frames. Two videos are aligned via a transport matrix $T$, where the optimal path is obtained by comparing embedding similarities. This alignment groups frames into steps, each represented by a distinct color. While KOT relies solely on inter-domain costs, GWOT additionally enforces intra-domain structural consistency, producing smoother temporal mappings. In contrast, the proposed R-FPGWOT relaxes the balanced-mass constraint by introducing a virtual sink node (gray), which absorbs background or redundant frames (darker shades) and preserves clean, step-wise correspondences across videos.

Kumar et al. (2022)), procedure learning (PL) analyzes collections of demonstrations to infer both the key-steps and their temporal sequencing. This is challenging because different demonstrations of the same procedure may present steps in different orders (e.g., adding dressing before or after chopping vegetables), repeat certain steps, or include irrelevant background segments of idle motion. Related directions in instructional video understanding have explored planning (Zhao et al. (2022)), correctness verification (Qian et al. (2022)), and instructional summarization (Narasimhan et al. (2022)). In contrast, PL uniquely focuses on aligning demonstrations into a coherent sequence of key-steps. Prior research has approached PL in supervised and weakly supervised settings. Supervised PL methods (Naing & Elhamifar (2020); Zhou et al. (2018); Zhukov et al. (2019)) depend on costly frame-level annotations, while weakly supervised approaches (Li & Todorovic (2020); Richard et al. (2018); Chang et al. (2019)) rely on predefined step lists, limiting scalability. Self-supervised approaches (Bansal et al. (2022); Dwibedi et al. (2019)) exploit procedural structure via monotonic alignment assumptions (Hadji et al. (2021)). Real-world instructional videos, however, often deviate from these assumptions and exhibit temporal irregularities (Fig. 1): (a) *background* frames with irrelevant content (e.g., waiting, idle motion, or showing ingredients), (b) *non-monotonic* sequences where steps occur out of order (e.g., add sauce before chopping all vegetables), and (c) *redundant* segments capturing repeated or unnecessary steps, complicating alignment.

Early self-supervised methods like TCC (Dwibedi et al. (2019)) and CnC (Bansal et al. (2022)) introduced cycle-consistency or contrastive learning but struggled with clutter. OT-based methods reframed alignment of frames as an assignment problem. Methods such as VAVA (Shen et al. (2021)) combined OT with contrastive loss but failed at balancing multiple losses and handling repeated actions. OPEL (Chowdhury et al. (2024)) used Kantorovich OT (KOT) (Thorpe (2018)) with temporal priors yet remained sensitive to irrelevant frames. Recent techniques such as ASOT (Xu & Gould (2024)), VASOT (Ali et al. (2025)), and RGWOT (Mahmood et al. (2025)) leveraged Gromov-Wasserstein OT (GWOT) (Peyré et al. (2016)) for relational matching and reordering. However, their *fully balanced* formulations enforced strict one-to-one correspondences between frames, causing background segments (e.g., waiting, camera motion, idle actions) to be wrongly aligned with actual key-steps, thereby hindering accurate discovery.

In this paper, we propose *Regularized Fused Partial Gromov-Wasserstein Optimal Transport* (R-FPGWOT), a partial, regularized extension of Fused GWOT (FGWOT) that builds directly on RGWOT by relaxing the balanced marginal constraints and introducing a virtual sink node for unmatched or redundant frames. Unlike KOT and GWOT, this unbalanced partial transport (Bai et al. (2025)) allows irrelevant or background frames to be mapped to a single shared "*null*" mass instead of being forced into spurious correspondences. The formulation provides three main benefits: (i) exclusion of background frames using virtual sink, (ii) robustness to temporal and structural ordering variations via Laplace-shaped priors, and (iii) an adaptive fusion trade-off between semantic similarity (KOT-style feature costs) and structural consistency (GWOT-style relational costs).

To further improve stability, we integrate these temporal smoothness priors and the C-IDM regularizer into a unified loss, which prevents degenerate collapse of all frames into a single cluster and sharpens the transport plan around a near-diagonal procedural ridge. Finally, for each video, the key-steps are clustered using graphcut segmentation (Boykov et al. (2002)) in the embedding space. **REALIGN** (*Regularized Procedure Alignment with Matching Video Embeddings via Partial Gromov-Wasserstein Optimal Transport*) achieves **18.9% (7.62pp)** higher F1 and **30% (7.74pp)** higher IoU on both egocentric (EgoProceL (Bansal et al. (2022))) and third-person (ProceL (Elhamifar & Huynh (2020)), CrossTask (Zhukov et al. (2019))) datasets, producing semantically faithful alignments.

In summary, our main contributions are as follows:

- We introduce **REALIGN**, a new OT formulation for unsupervised PL that combines the semantic matching ability of classical Kantorovich OT with the structural consistency of Gromov-Wasserstein OT, while relaxing balanced constraints to better handle instructional videos.
- REALIGN supports flexible partial assignments, enabling robust alignment of demonstrations that contain background clutter, step re-orderings, or redundant actions-cases where fully balanced OT methods (e.g. OPEL, RGWOT) often fail.
- We design a unified alignment loss that integrates temporal smoothness, optimal regularization, and a novel inter-video contrastive term, preventing degenerate matches and improving stability in OT-based training.
- REALIGN achieves substantial performance gains over SOTA baselines, with an average improvement of **11.6% F1-score (4.45pp)** and **19.6% IoU (4.73pp)** on the EgoProceL benchmark.

## 2 RELATED WORKS

**Self-Supervised Representation Learning for Videos.** Self-supervised learning derives supervisory signals directly from data. Early work focused on images with tasks such as colorization (Larsson et al. (2016); Huang et al. (2016)), object counting (Liu et al. (2018)), jigsaw puzzle solving (Carlucci et al. (2019); Kim et al. (2018; 2019)), rotation prediction (Gidaris et al. (2018); Feng et al. (2019)), image inpaintings (Jenni et al. (2020)) and image clustering (Caron et al. (2018; 2019)). More recently, video-based methods exploit spatial and temporal cues through tasks like frame prediction (Ahsan et al. (2018); Diba et al. (2019); Han et al. (2019); Srivastava et al. (2015)), maintaining temporal consistency (Goroshin et al. (2015); Mobahi et al. (2009); Zou et al. (2011)), ordering frames (Fernando et al. (2017); Lee et al. (2017); Misra et al. (2016); Xu et al. (2019)), detecting the flow of time (Pickup et al. (2014); Wei et al. (2018)), estimating action speed (Benaim et al. (2020); Wang et al. (2020); Yao et al. (2020)), and clustering (Kumar et al. (2022); Tran et al. (2024)). Unlike these methods that often derive signals from a subset of videos, PL aims to uncover the key steps of a task and their order across multiple videos for broader generalization.

**Representations for Procedure Learning (PL).** PL emphasizes frame-level feature learning, using relative frame timestamps (Kukleva et al. (2019)), temporal prediction (VidalMata et al. (2021)), attention (Elhamifar & Huynh (2020)), or cross-video correspondences (Bansal et al. (2022)) to derive robust embeddings. Graph-based methods (Bansal et al. (2024)) further cluster semantically related frames but often require preprocessing (e.g., background removal) to mitigate noise and redundancy. Beyond purely visual methods, multi-modal PL has incorporated narrated text (Alayrac et al. (2016); Damen et al. (2014); Doughty et al. (2020); Fried et al. (2020); Malmaud et al. (2015); Yu et al. (2014)), optical flow, depth, or gaze signals (Shah et al. (2023)). These modalities enrich supervision but suffer from stream misalignment (Elhamifar & Huynh (2020); Elhamifar & Naing (2019)), automatic speech recognition (ASR) errors requiring manual fixes, and high memory and computation costs. Recent purely visual OT-based works (Chowdhury et al. (2024); Xu & Gould (2024); Ali et al. (2025); Mahmood et al. (2025)) laid the foundation on which we build our novel OT formulation for egocentric visual PL.

**Video Alignment.** Classical alignment methods like Canonical Correlation Analysis (CCA) (Andrew et al. (2013)) and soft-Dynamic Time Warping (DTW) (Haresh et al. (2021)), assume synchronization, while TCC (Dwibedi et al. (2019)) and GTCC (Donahue & Elhamifar (2024)) enforce local cycle-consistency. For global alignment, LAV (Haresh et al. (2021)) leverages DTW assuming monotonic sequences, whereas KOT-based methods (Liu et al. (2022); Chowdhury et al. (2024)) remain sensitive to repeated actions and loss balancing. Recent GWOT-based methods (Ali et al. (2025); Mahmood et al. (2025); Xu & Gould (2024)) handle reordering and redundancy but risk degenerate solutions. In this work, we propose a *regularized fused partial OT formulation*, incorporating Laplace priors and inter-video contrastive loss for more robust unsupervised PL.

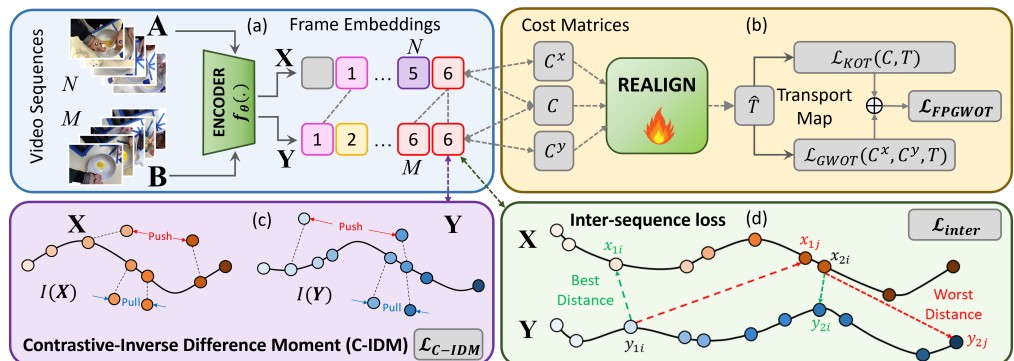

Figure 2: REALIGN framework. (a) An encoder generates frame-level embeddings from two video sequences, which serve as inputs for alignment. (b) A fused partial Gromov-Wasserstein optimal transport (FPGWOT) module, guided by structural priors, computes the transport map to establish frame-to-frame correspondences. (c) A contrastive regularization term (C-IDM) pushes apart dissimilar frames while pulling together temporally coherent ones. (d) An inter-sequence loss further stabilizes training by penalizing degenerate alignments, encouraging both the best and worst distances to be respected. Forward and backward arrows represent computation and gradient flows, while grey indicates temporal alignment and purple/green denote regularization components.

**Learning Key-step Ordering.** Most prior work in PL overlooks the variability in task execution, often assuming a fixed sequential order of key-steps (Elhamifar & Naing (2019); Kukleva et al. (2019); VidalMata et al. (2021)) or ignoring the ordering altogether (Elhamifar & Huynh (2020); Shen et al. (2021)). As shown in Figure 1, a task can be completed in different valid ways, with steps rearranged or substituted. Our method captures this variability by building a tailored key-step sequence for each video, letting the model adapt to the specific ordering.

## 3 METHODOLOGY

Our goal in REALIGN is to align instructional videos in a way that preserves both semantic meaning and temporal structure, while staying robust to background noise and redundancy. To achieve this, we design a framework that extends optimal transport with partial matching, structural priors, and contrastive regularization. The following subsections describe how each component contributes to reliable procedural alignment and key-step discovery.

### 3.1 REGULARIZED PARTIAL GROMOV-WASSERSTEIN OPTIMAL TRANSPORT (R-FPGWOT)

Optimal Transport (OT) compares two probability distributions by moving mass from one to another while minimizing transportation cost (Villani et al. (2009)). Let two instructional videos $A$ and $B$ with $N$ and $M$ frames be encoded by $f_\theta$ (Fig. 2(a)) into frame embeddings $\boldsymbol{X} = \{x_i\}_{i=1}^N \in \mathbb{R}^{N \times D}$ and $\boldsymbol{Y} = \{y_j\}_{j=1}^M \in \mathbb{R}^{M \times D}$. Each video is modeled as an empirical distribution: $\mu = \sum_{i=1}^N \alpha_i \delta_{x_i}$ and $\nu = \sum_{j=1}^M \beta_j \delta_{y_j}$ with uniform weights $\alpha_i = \frac{1}{N}, \beta_j = \frac{1}{M}$. The transportation polytope (Cuturi (2013)), $U(\boldsymbol{\alpha}, \boldsymbol{\beta}) := \{\boldsymbol{T} \in \mathbb{R}_+^{N \times M} : \boldsymbol{T}\mathbf{1}_N = \boldsymbol{\alpha}, \boldsymbol{T}^\top \mathbf{1}_M = \boldsymbol{\beta}\}$ leads to a feasible set of weight matrices. Learning procedural alignment reduces to finding a coupling $\boldsymbol{T}$ between $\mu$ and $\nu$ that best preserves semantic and temporal consistency.

Classical *Kantorovich OT (KOT)* aligns frames based on direct feature similarity, while *Gromov-Wasserstein OT (GWOT)* aligns their structural relations. Their complementary strengths under the common objective motivate *Fused GWOT* as shown in Fig. 2(b), which produces alignments that are semantically faithful and temporally coherent as shown:

$$\mathcal{L}_{\text{FGWOT}}(\boldsymbol{T}) = \arg \min_{\boldsymbol{T} \in U(\alpha,\beta)} (1-\rho)\mathcal{L}_{\text{KOT}}(C, \boldsymbol{T}) + \rho \mathcal{L}_{\text{GWOT}}(\boldsymbol{C^x}, \boldsymbol{C^y}, \boldsymbol{T})$$

$$= \arg \min_{\boldsymbol{T} \in U(\alpha,\beta)} (1-\rho)\langle \boldsymbol{C}, \boldsymbol{T} \rangle + \rho \sum_{i,k=1}^N \sum_{j,l=1}^M L(\boldsymbol{C}_{ik}^x, \boldsymbol{C}_{jl}^y) T_{ij} T_{kl}, \quad (1)$$

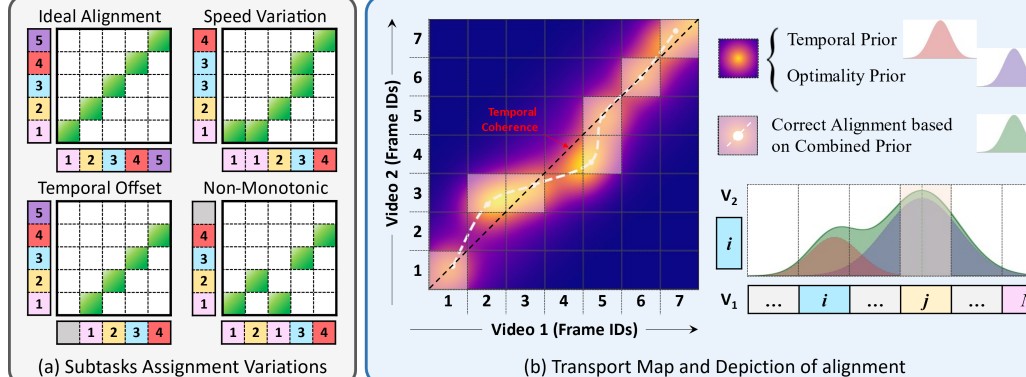

Figure 3: (a) Examples of pairwise alignment scenarios captured by the assignment matrix. (b) Visualization of the OT map in 2D, along with a 1D illustration showing how $i$-th frame from Video 2 aligns with its best-matched $j$-th frame from Video 1.

where $C_{ij} = \|x_i - y_j\|_2$ captures appearance cost, and $C^x \in \mathbb{R}^{N \times N}$ and $C^y \in \mathbb{R}^{M \times M}$ capture intra-sequence distances in $X$ and $Y$. Here $T_{ij}$ reflects how much mass of frame $x_i$ is transported to frame $y_j$. Setting $\rho = 0$ recovers KOT, and $\rho = 1$ recovers GWOT. However, real instructional videos often contain background content, idle moments, or repeated segments. Enforcing strict one-to-one matching between every frame of the two videos can push these irrelevant frames into misleading correspondences. To address this, we extend FGWOT (Mahmood et al. (2025)) with unbalanced OT penalties, leading to *Partial FGWOT* (FPGWOT):

$$\min_{T \geq 0}(1 - \rho)\langle C, T\rangle + \rho\sum_{i,k}\sum_{j,l} L(C^x_{ik}, C^y_{jl})T_{ij}T_{kl} + \tau\Big(\mathrm{KL}(T\mathbf{1}\|\alpha) + \mathrm{KL}(T^\top\mathbf{1}\|\beta)\Big) - \epsilon h(T), \quad (2)$$

where $\tau > 0$ controls how strict marginal constraints are enforced. This formulation allows unmatched frames to be softly assigned to a 'null' sink instead of forced matches, improving robustness. To make optimization computationally feasible, entropy regularization $-\epsilon h(T)$ (Cuturi (2013); Peyré et al. (2016)) is added, where $h(T) = -\sum_{i=1}^{N}\sum_{j=1}^{M} t_{ij} \log t_{ij}$ and $\epsilon > 0$.

**Regularization using Priors.** Gromov-Wasserstein OT aligns sequences by matching their pairwise relational structure, but it is agnostic to absolute time and does not inherently prefer near-diagonal (time-consistent) alignments. As a result, GW can assign semantically similar but temporally distant frames to each other, leading to off-diagonal, procedurally incoherent transport plans, especially under repeated actions, camera motion, or clutter. In instructional videos, however, key steps typically unfold in roughly the same order across demonstrations, so semantically similar frames should also lie close along the time axis. We therefore introduce *Temporal* and *Optimality* Laplace-shaped priors follow the formulation of (Liu et al. (2022); Chowdhury et al. (2024)) that inject an explicit diagonal inductive bias: they softly encourage mass to concentrate near the main diagonal (time-consistent matches) in transport matrix $T$, while still allowing deviations to account for early starts, speed variations, and non-monotonic executions (Fig. 3(a)). Concretely, the prior $Q$ is defined as:

$$Q(i, j) = \phi \exp\Big(-\frac{|d_t(i, j)|}{b}\Big) + (1 - \phi) \exp\Big(-\frac{|d_o(i, j)|}{b}\Big), \quad \phi : 1 \to 0.5 \text{ training.}$$
$$d_t(i, j) = \frac{|i/N - j/M|}{\sqrt{1/N^2 + 1/M^2}}; \qquad d_o(i, j) = \frac{|i/N - i_o/N| + |j/M - j_o/M|}{2\sqrt{1/N^2 + 1/M^2}} \quad (3)$$

where $d_t(i, j)$ preserves global temporal order, and $d_o(i, j)$ captures optimal alignment likelihood to center $(i_o, j_o)$. The mixing factor $\phi$ is annealed from 1 to 0.5 during training as in VAVA (Liu et al. (2022)), balancing temporal structure with non-monotonic flexibility.

**Virtual frame for background.** To handle background or redundant frames and avoid spurious matches, we append a *virtual frame* to both axes of the transport matrix, yielding $\hat{T} \in \mathbb{R}_+^{(N+1)\times(M+1)}$. If the matching probability of $x_i$ ($i \leq N$) with all $y_j$ ($j \leq M$) falls below a threshold $\zeta$, $x_i$ is assigned to the virtual frame $y_{M+1}$, and symmetrically for $y_j$. Virtual frames and their assignments act as sinks and are excluded from supervision and loss computation as formulated in (Liu et al. (2022); Chowdhury et al. (2024)).

**IDM-style structural regularization (with FPGWOT).** To further stabilize training, we regularize $\boldsymbol{T}$ using inverse-distance moments (IDM) (Albregtsen et al. (2008); Liu et al. (2022)):

$$M(\hat{\boldsymbol{T}}) = \phi \sum_{ij} \frac{t_{ij}}{(\frac{i}{N} - \frac{j}{M})^2 + 1} + (1 - \phi) \sum_{ij} \frac{t_{ij}}{\frac{1}{2}d_m + 1}, \qquad d_m = \left(\frac{i - i_o}{N + 1}\right)^2 + \left(\frac{j - j_o}{M + 1}\right)^2 \quad (4)$$

where the first term promotes diagonal concentration (temporal smoothness) and the second enforces sharp ridges (alignment confidence).

**Constrained feasible set.** We embed these priors into the feasible set of the partial FGWOT formulation. Unlike balanced OT, which enforces $\boldsymbol{T}\mathbf{1} = \alpha$ and $\boldsymbol{T}^\top \mathbf{1} = \beta$, our relaxation permits mass imbalance while constraining the structure of $\hat{\boldsymbol{T}}$. Specifically, we require (i) sufficiently high structural score $M(\hat{\boldsymbol{T}}) \geq \xi_1$, and (ii) proximity to a prior matrix $\hat{\boldsymbol{Q}}$ measured by $\mathrm{KL}(\hat{\boldsymbol{T}}\|\hat{\boldsymbol{Q}}) \leq \xi_2$:

$$U_{\xi_1, \xi_2}(\boldsymbol{\alpha}, \boldsymbol{\beta}) = \left\{ \hat{\boldsymbol{T}} \geq 0 : \hat{\boldsymbol{T}}\mathbf{1}_{M+1} \approx \boldsymbol{\alpha}, \ \hat{\boldsymbol{T}}^\top \mathbf{1}_{N+1} \approx \boldsymbol{\beta}, \ M(\hat{\boldsymbol{T}}) \geq \xi_1, \ \mathrm{KL}(\hat{\boldsymbol{T}}\|\hat{\boldsymbol{Q}}) \leq \xi_2 \right\}. \quad (5)$$

The approximate marginal constraints (Xu & Gould (2024); Bai et al. (2025)) allows unmatched or redundant frames to be softly assigned to the null sink rather than forced into noisy matches. Introducing Lagrange multipliers $\lambda_1, \lambda_2 > 0$ for the IDM and KL penalties yields the dual-*Regularized Fused Partial GWOT* (R-FPGWOT) program:

$$\ell^{\text{R-FPGW}}_{\lambda_1, \lambda_2, \tau} = \min_{\hat{\boldsymbol{T}} \geq 0} \left\langle \hat{\boldsymbol{T}}, \widetilde{\boldsymbol{D}}(\hat{\boldsymbol{T}}) \right\rangle - \lambda_1 M(\hat{\boldsymbol{T}}) + \lambda_2 \, \mathrm{KL}(\hat{\boldsymbol{T}}\|\hat{\boldsymbol{Q}}) + \tau \Big( \mathrm{KL}(\hat{\boldsymbol{T}}\mathbf{1}_{M+1}\|\alpha) + \mathrm{KL}(\hat{\boldsymbol{T}}^\top \mathbf{1}_{N+1}\|\beta) \Big).$$
$$(6)$$

where $\widetilde{\boldsymbol{D}}(\hat{\boldsymbol{T}}) = (1 - \rho)\boldsymbol{C} + \rho \, G(\hat{\boldsymbol{T}})$ is the fused cost matrix combining appearance cost $\boldsymbol{C}$ and the linearized GW gradient $G(\hat{\boldsymbol{T}}) = 2\,\boldsymbol{C}^x \hat{\boldsymbol{T}} \boldsymbol{C}^y$. Because $\widetilde{\boldsymbol{D}}$ depends on $\hat{\boldsymbol{T}}$, We iteratively solve a KL-regularized *linearized* OT subproblem for $\hat{\boldsymbol{T}}^{(s+1)}$ at outer step $s$ by freezing $G(\hat{\boldsymbol{T}}^{(s)})$. The inner solution retains a Sinkhorn-like scaling form: $\hat{\boldsymbol{T}}^{(s+1)} = \mathrm{Diag}(u^{(s)}) \, \boldsymbol{K}^{(s)} \, \mathrm{Diag}(v^{(s)})$.

$$\boldsymbol{K}^{(s)} = \left[ q_{ij} \exp\left( \frac{1}{\lambda_2}\left( s_{ij}^{\lambda_1} - \widetilde{D}_{ij}^{(s)} \right) \right) \right]_{ij}, \qquad s_{ij}^{\lambda_1} = \lambda_1 \left( \frac{1}{(\frac{i}{N+1} - \frac{j}{M+1})^2 + 1} + \frac{1}{\frac{1}{2}d_m + 1} \right) \quad (7)$$

and $(u^{(s)}, v^{(s)})$ updated using *unbalanced Sinkhorn iterations* to satisfy relaxed marginal constraints under penalty $\tau$. This procedure inherits FGWOT's ability to couple semantic and structural cues, while the partial relaxation and virtual frame allow irrelevant mass to be safely discarded.

**Contrastive stabilization.** To avoid trivial or collapsed mappings, the *intra-sequence* C-IDM loss from (Haresh et al. (2021); Liu et al. (2022)) (Eq. 8) enforces temporal smoothness by pulling adjacent frames together while pushing apart distant ones (Fig. 2(c));

$$I(\boldsymbol{X}) = \sum_{i,j} \left(1 - \mathcal{N}(i,j)\right) \gamma(i,j) \max\{0, \lambda_3 - d(i,j)\} + \mathcal{N}(i,j) \frac{d(i,j)}{\gamma(i,j)}, \quad (8)$$

with $\mathcal{N}(i,j) = \mathbf{1}\{|i - j| \leq \delta\}$, $\gamma(i,j) = (i - j)^2 + 1$, $d(i,j) = \|\boldsymbol{x}_i - \boldsymbol{x}_j\|$.

The *inter-sequence* CL (Chowdhury et al. (2024)) (Eq. 9) uses $\hat{\boldsymbol{T}}^{(s+1)}$ to select best & worst matches across videos, minimizing distances for best pairs while maximizing for worst (Fig 2(d)).

$$\mathcal{L}_{\text{inter}} = \mathrm{CE}\left( \begin{bmatrix} \text{best\_dist} \\ \text{worst\_dist} \end{bmatrix}, \begin{bmatrix} 0 \\ 1 \end{bmatrix} \right), \quad \text{(best/worst) from arg max/min of } \hat{\boldsymbol{T}}^{\text{R-FPGW}}_{\lambda_1, \lambda_2} \text{ along rows/cols.}$$
$$(9)$$

Intuitively, this objective ensures that embeddings connected by strong transport weights remain close, while those with negligible alignment are pushed apart. Together with the intra-sequence C-IDM term, it prevents degenerate clustering and yields robust, discriminative alignment across videos. The overall objective of REALIGN combines the regularized OT loss (Eq. 6) with contrastive regularization terms, which together enable fused appearance-structure alignment with partial mass handling, enforce IDM-style temporal shape, anchor plans to Laplace priors, and preserve both diversity and cross-video separability.

$$\mathcal{L}_{\text{REALIGN}} = c_1 \, \mathcal{L}_{\text{R-FPGWOT}} + c_2 \mathcal{L}_{\text{C-IDM}} + c_3 \, \mathcal{L}_{\text{inter}} = c_1 \, \ell^{\text{R-FPGW}}_{\lambda_1, \lambda_2, \tau}(\boldsymbol{X}, \boldsymbol{Y}) + c_2 \big( I(\boldsymbol{X}) + I(\boldsymbol{Y}) \big) + c_3 \, \mathcal{L}_{\text{inter}}.$$
$$(10)$$

Table 1: Results on EgoProceL comparing REALIGN with OT-based and prior baselines. Best and second-best scores are in bold and underlined. STEPS (Shah et al. (2023)) (purple) uses extra modalities (flow, gaze, depth), while our method relies only on visuals. OT-based SOTA methods are shown in gray, and our work REALIGN is highlighted in blue.

| | EgoProceL | | | | | | | | | | | |
| | CMU-MMAC | | EGTEA-GAZE+ | | MECCANO | | EPIC-Tents | | PC Assembly | | PC Disassembly | |
| | F1 | IoU | F1 | IoU | F1 | IoU | F1 | IoU | F1 | IoU | F1 | IoU |
|---|---|---|---|---|---|---|---|---|---|---|---|---|
| Random | 15.7 | 5.9 | 15.3 | 4.6 | 13.4 | 5.3 | 14.1 | 6.5 | 15.1 | 7.2 | 15.3 | 7.1 |
| Uniform | 18.4 | 6.1 | 20.1 | 6.6 | 16.2 | 6.7 | 16.2 | 7.9 | 17.4 | 8.9 | 18.1 | 9.1 |
| CnC (Bansal et al. (2022)) | 22.7 | 11.1 | 21.7 | 9.5 | 18.1 | 7.8 | 17.2 | 8.3 | 25.1 | 12.8 | 27.0 | 14.8 |
| GPL-2D (Bansal et al. (2024)) | 21.8 | 11.7 | 23.6 | 14.3 | 18.0 | 8.4 | 17.4 | 8.5 | 24.0 | 12.6 | 27.4 | 15.9 |
| UG-I3D (Bansal et al. (2024)) | 28.4 | 15.6 | 25.3 | 14.7 | 18.3 | 8.0 | 16.8 | 8.2 | 22.0 | 11.7 | 24.2 | 13.8 |
| GPL-w BG (Bansal et al. (2024)) | 30.2 | 16.7 | 23.6 | 14.9 | 20.6 | 9.8 | 18.3 | 8.5 | 27.6 | 14.4 | 26.9 | 15.0 |
| GPL-w/o BG (Bansal et al. (2024)) | 31.7 | 17.9 | 27.1 | 16.0 | 20.7 | 10.0 | 19.8 | 9.1 | 27.5 | 15.2 | 26.7 | 15.2 |
| STEPS (Shah et al. (2023)) | 28.3 | 11.4 | 30.8 | 12.4 | 36.4 | 18.0 | **42.2** | 21.4 | 24.9 | 15.4 | 25.9 | 14.6 |
| OPEL (Chowdhury et al. (2024)) | 36.5 | 18.8 | 29.5 | 13.2 | 39.2 | 20.2 | 20.7 | 10.6 | 33.7 | 17.9 | 32.2 | 16.9 |
| RGWOT (Mahmood et al. (2025)) | 54.4 | 38.6 | 37.4 | 22.9 | 59.5 | 42.7 | 39.7 | 24.9 | **43.6** | **28.0** | **45.9** | **30.1** |
| *REALIGN* (R-FGWOT) (Ours) | 58.3 | 42.5 | 62.4 | 47.2 | 59.1 | 42.3 | 39.1 | 24.4 | 40.9 | 25.4 | 41.9 | 28.1 |
| *REALIGN* (R-FPGWOT) (Ours) | **59.7** | **43.7** | **64.2** | **49.3** | **59.6** | **42.7** | 39.8 | **25.0** | 41.4 | 26.3 | 42.5 | 28.6 |

**Clustering and Key-Step Ordering.** Using frame embeddings from our R-FPGWOT alignment framework, we localize key steps and infer their order to capture procedural structure. As in prior work, we frame key-step localization as a multi-label graph-cut segmentation problem (Greig et al. (1989)), where terminal nodes represent $K$ candidate steps and non-terminal nodes represent the frame embeddings. T-links connect frames to steps, while n-links enforce temporal smoothness. We solve the resulting energy with the $\alpha$-Expansion algorithm (Boykov et al. (2002)), assigning each frame to one of the $K$ clusters. The underlying multi-label Potts energy with data (T-link) and temporal smoothness (N-link) terms is defined in Eq. (A17). For ordering, we normalize timestamps within each video and compute the mean time of frames in each cluster, following (Bansal et al. (2022); Chowdhury et al. (2024)). Sorting these means gives the predicted sequence, which we then aggregate across videos of the same task, selecting the most frequent sequence as the canonical procedure. Importantly, this graph-cut stage is applied *after* the transport matrix $T$ is fully learned and does *not* influence the optimization of R-FPGWOT. The alignment itself is entirely determined by the differentiable OT-based objective; graph-cut simply converts the learned transport map and embeddings into contiguous temporal segments. As shown in Appendix A.8.1, and supported by ablations in Table 4, alternative post-hoc segmenters such as K-Means and Subset-Selection underperform, indicating that graph-cut yields cleaner segment boundaries without altering the fundamental behavior of the learned alignment model.

## 4 EXPERIMENTS AND RESULTS

**Datasets.** We evaluate REALIGN across both egocentric and third-person perspectives. For third-person analysis, we use **CrossTask** (Zhukov et al. (2019)), which contains 213 hours of video spanning 18 primary tasks (2763 videos), and **ProceL** (Elhamifar & Huynh (2020)), with 720 videos covering 12 tasks over 47.3 hours. For egocentric evaluation, we adopt the large-scale **EgoProceL** benchmark (Bansal et al. (2022)), featuring 62 hours of head-mounted recordings from 130 users performing 16 tasks. Dataset statistics are summarized in Appendix Table A2.

**Evaluation.** We follow the evaluation practices of current state-of-the-art (Chowdhury et al. (2024); Mahmood et al. (2025)), reporting both F1-score and temporal Intersection-over-Union (IoU). Framewise scores are computed per key step and averaged across steps. Precision measures the proportion of correctly predicted key-step frames among all predicted, while recall measures the proportion of ground-truth key-step frames correctly retrieved. Following (Bansal et al. (2022); Elhamifar & Huynh (2020); Shen et al. (2021)), the Hungarian algorithm (Kuhn (1955)) is used to establish a one-to-one mapping between predicted and ground-truth steps.

**Experimental Setup.** We use a ResNet-50 backbone (pretrained on ImageNet) for frame-level feature extraction, following (Bansal et al. (2022); Chowdhury et al. (2024)). The encoder is trained on pairs of videos, with random frame sampling and optimization of our proposed $L_{\text{REALIGN}}$ until convergence. Features are taken from the Conv4c layer and stacked with a two-frame temporal context. This representation is passed through two 3D convolutional layers, a global max pooling layer, two fully connected layers, and a final linear projection producing 128-d embeddings. Implementation hyperparameters are given in Appendix Table A1. All reported numbers are averaged over three independent runs to account for variance. Code will be released on acceptance.

**Results on Egocentric View.** Table 1 provides comparative evaluation of *REALIGN* against SOTA baselines on EgoProceL (Bansal et al. (2022)). This benchmark is designed for egocentric PL and remains a challenging testbed. Our method surpasses previous works across nearly all tasks, achieving consistent gains of up to **11.6% F1 (4.45pp)** and **19.6% IoU (4.73pp)** over the SOTA baseline (Mahmood et al. (2025)). As shown in Appendix Table A2, datasets with higher background content (e.g., EGTEA-GAZE+ and CMU-MMAC) exhibit greater sink-mass fractions. Our partial fusion leads to proportionally higher relative F1 gains (e.g., 71% or 26.8pp for EGTEA-GAZE+ and 9.8% or 5.3pp for CMU-MMAC), confirming the benefit of partial fusion in mitigating redundant frames. Detailed task-wise results within CMU-MMAC and EGTEA-GAZE+ have been aggregated in Appendix Table A4. These improvements highlight the effectiveness of Fused Partial GWOT in handling redundant frames, order variations, and viewpoint-specific artifacts in egocentric video.

**Results on Third-person View.** We further evaluate on ProceL (Elhamifar & Huynh (2020)) and CrossTask (Zhukov et al. (2019)) (Table 2), following identical protocols from prior self-supervised procedure learning models. Competing approaches (Kukleva et al. (2019); Elhamifar & Huynh (2020)) often map most frames to a single degenerate solution. *REALIGN* consistently improves performance and outperforms existing models like RGWOT (Mahmood et al. (2025)) by **30.0% (13.3pp)** on ProceL and **51.9% (21pp)** on CrossTask on F1-score. Detailed breakdowns for CMU-MMAC, ProceL, and CrossTask subtasks are reported in Appendix Tables A3 and A5.

Table 2: PL results on third-person datasets. P (Precision), R (Recall), and F1-score. The **best** and second-best results are highlighted.

|  | ProceL | | | CrossTask | | |
|---|---|---|---|---|---|---|
|  | P | R | F1 | P | R | F1 |
| Uniform | 12.4 | 9.4 | 10.3 | 8.7 | 9.8 | 9.0 |
| Alayrac et al. (2016) | 12.3 | 3.7 | 5.5 | 6.8 | 3.4 | 4.5 |
| Kukleva et al. (2019) | 11.7 | 30.2 | 16.4 | 9.8 | 35.9 | 15.3 |
| Elhamifar & Huynh (2020) | 9.5 | 26.7 | 14.0 | 10.1 | 41.6 | 16.3 |
| Fried et al. (2020) | - | - | - | - | 28.8 | - |
| Shen et al. (2021) | 16.5 | 31.8 | 21.1 | 15.2 | 35.5 | 21.0 |
| CnC | 20.7 | 22.6 | 21.6 | 22.8 | 22.5 | 22.6 |
| UG-I3D | 21.3 | 23.0 | 22.1 | 23.4 | 23.0 | 23.2 |
| GPL | 22.4 | 24.5 | 23.4 | 24.9 | 24.1 | 24.5 |
| STEPS | 23.5 | 26.7 | 24.9 | 26.2 | 25.8 | 25.9 |
| OPEL | 33.6 | 36.3 | 34.9 | 35.6 | 34.8 | 35.1 |
| RGWOT | 42.2 | 46.7 | 44.3 | 40.4 | 40.7 | 40.4 |
| *REALIGN* (R-FGWOT) | 53.5 | 60.4 | 56.7 | 60.2 | 61.2 | 60.6 |
| *REALIGN* (R-FPGWOT) | **54.4** | **61.5** | **57.6** | **60.9** | **61.9** | **61.4** |

**Comparison with Multimodal Methods.**
We further compare REALIGN with multimodal Procedure learning approaches that leverage richer input signals such as depth, gaze, or narration. Table 1 contrasts our RGB-only framework with **STEPs** (Shah et al. (2023)) (purple), which incorporates depth and gaze in addition to RGB. Despite relying solely on visual frames, REALIGN surpasses STEPs on most datasets, and while STEPs achieves a slightly higher F1 score on EPIC-Tents (Jang et al. (2019)), our model still delivers stronger IoU, indicating more consistent temporal alignment. In addition, REALIGN outperforms narration-augmented approaches (Alayrac et al. (2016); Shen et al. (2021)) (yellow) in Table 2, underscoring that carefully designed transport-based regularization can rival or exceed methods using multimodal supervision.

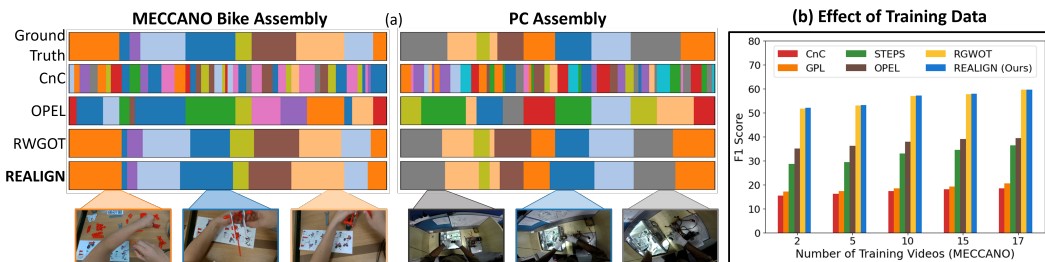

Figure 4: (a) Qualitative outcomes on MECCANO and PC Assembly, where color highlights distinguish sub-tasks across key-steps. REALIGN achieves superior alignment compared to existing SOTA methods by introducing a virtual frame to effectively manage unmatched frames. (b) Influence of training data volume on encoder performance.

**Qualitative Results.** Fig. 4(a) compares REALIGN with prior baselines. CnC (Bansal et al. (2022)) tends to over-segment, while OPEL (Chowdhury et al. (2024)) and RGWOT (Mahmood et al. (2025)) still misalign steps and fail to manage redundant frames. In contrast, REALIGN handles mass imbalance by routing background to the virtual sink, leading to faithful key-step localization and interpretable transport maps.

## 5 ABLATION STUDY

**Effectiveness of Model Components of $L_{\textbf{REALIGN}}$.** Table 3 reports the impact of different loss components by systematically removing them from the *REALIGN* model. The complete configuration, combining contrastive regularization, Laplace temporal and optimal priors, structural prior, and virtual fusion frame, achieves the best results. Removing the partial penalty ($\tau$) leads to the sharpest drop ($\sim$4–5 F1 points), highlighting the importance of handling background frames and mass imbalance allowing redundant frames to flow into a sink. Excluding Laplace-based temporal or structural priors produces a consistent 1.5–3.0pp degradation in F1, underscoring their role in enforcing near-diagonal, structurally coherent alignments and preventing the off-diagonal drift typical of unconstrained GW. The Contrastive IDM regularizer sharpens the transport plan by discouraging overly diffuse, off-diagonal matches. Contrastive regularization yields smaller gains on some datasets, but is crucial for enforcing a consistent embedding geometry across videos, thereby stabilizing the OT kernel and avoiding degenerate mappings. The KL divergence contributes marginally on its own (less than 1 point), since $\hat{T}$ and $\hat{Q}$ are already close by construction, but it further stabilizes optimization when combined with other terms. Importantly, all components are differentiable and computationally lightweight, and the ablations show that removing any one of them consistently hurts performance. Overall, while individual factors vary in influence, their cumulative effect yields up to $\sim$6 point gains in F1/IoU, justifying the inclusion of all components in the proposed, yet still simple-to-optimize, REALIGN formulation.

Table 3: Ablation study of REALIGN loss components. We analyze the contribution of *contrastive regularizers* (intra C-IDM and inter-sequence), *regularizer priors* (temporal (T) and optimal (O) Laplace priors), *structural prior* (fused GWOT term), *virtual frame* and *partial penalty* ($\tau$).

| Contrastive Regularizers | Regularizer Priors | KL-Divergence | Virtual Frame | Structural Prior | Partial Penalty ($\tau$) | MECCANO F1 | MECCANO IoU | CMU-MMAC F1 | CMU-MMAC IoU |
|---|---|---|---|---|---|---|---|---|---|
| | ✓(T+O) | ✓ | ✓ | | | 36.8 | 17.1 | 36.1 | 16.5 |
| ✓ | | ✓ | ✓ | | | 35.8 | 16.1 | 32.6 | 14.4 |
| ✓ | ✓(T+O) | | ✓ | | | 38.1 | 19.1 | 35.2 | 17.3 |
| ✓ | ✓(T+O) | ✓ | | | | 38.6 | 19.6 | 33.8 | 16.4 |
| ✓ | ✓(T+O) | ✓ | ✓ | | | 39.2$^\dagger$ | 20.2$^\dagger$ | 36.5$^\dagger$ | 18.8$^\dagger$ |
| ✓ | | ✓ | ✓ | ✓ | | 51.8 | 35.5 | 50.5 | 33.7 |
| | ✓(T) | ✓ | ✓ | ✓ | | 57.3 | 41.2 | 53.5 | 36.9 |
| ✓ | ✓(T) | ✓ | ✓ | ✓ | | 59.5* | 42.7* | 54.4* | 38.6* |
| ✓ | ✓(T+O) | ✓ | ✓ | ✓ | | 59.1 | 42.3 | 58.3 | 42.5 |
| ✓ | ✓(T+O) | ✓ | ✓ | ✓ | ✓ | **59.6** | **42.3** | **59.7** | **43.7** |

$^\dagger$ OPEL and * RGWOT highlights the baslines.

**Effect of Clustering Methods.** We assess the impact of different clustering strategies by replacing our approach with K-Means, Subset Selection (SS), and a random assignment baseline. As summarized in Table 4, alternative clustering methods consistently underperform, while our proposed OT-based graph cut segmentation achieves the highest scores across all datasets. These results underscore the importance of jointly leveraging transport-based embeddings with structured clustering for accurate key-step discovery.

Table 4: Analysis of clustering algorithm across various datasets.

| | CMU-MMAC F1 | IoU | EGTEA-GAZE+ F1 | IoU | MECCANO F1 | IoU | EPIC-Tents F1 | IoU | ProceL F1 | IoU | CrossTask F1 | IoU |
|---|---|---|---|---|---|---|---|---|---|---|---|---|
| Random | 16.0 | 7.1 | 15.6 | 6.9 | 13.8 | 6.4 | 14.4 | 6.8 | 15.5 | 7.4 | 15.6 | 7.3 |
| OT + K-means | 38.2 | 22.1 | 32.4 | 21.9 | 32.5 | 20.4 | 26.1 | 15.4 | 34.1 | 20.9 | 36.7 | 21.6 |
| OT + SS | 46.0 | 32.7 | 49.1 | 38.6 | 46.2 | 31.1 | 29.2 | 18.1 | 45.0 | 31.7 | 46.6 | 34.2 |
| *REALIGN* (R-FPGWOT) | **59.7** | **43.7** | **64.2** | **49.3** | **59.6** | **42.7** | **39.8** | **25.0** | **57.6** | **42.6** | **61.4** | **46.7** |

**Number of key-steps.** Table 5 quantifies the effect of the number of clusters $k$ on REALIGN, showing that performance is stable over a broad range and peaks at $k = 7$, while increasing $k$ to 10 or higher leads to a sharp degradation. This behavior is expected, since the choice of $k$ is task- and semantics-dependent: closely related actions (e.g., pouring oil vs. pouring water) may reasonably be represented by a single cluster, whereas setting $k$ larger than the true number of distinct subtasks in-

Table 5: Results for key-steps $k$.

| $k$ | PC Assembly R | F1 | IoU | PC Disassembly R | F1 | IoU |
|---|---|---|---|---|---|---|
| 7 | **45.1** | **41.4** | **26.3** | **47.2** | **42.5** | **28.6** |
| 10 | 39.8 | 37.9 | 23.7 | 42.1 | 38.4 | 25.1 |
| 12 | 38.5 | 36.8 | 22.9 | 40.2 | 37.3 | 24.2 |
| 15 | 36.2 | 35.6 | 20.3 | 38.1 | 36.7 | 22.7 |

duces cluster fragmentation into near-duplicate states, which in turn perturbs the segmentation and lowers the overall scores (Appendix Fig. A2). Crucially, $k$ is only used in the post-hoc segmentation

stage and does not enter the OT alignment, so the learned transport geometry is unaffected by its precise value. This decoupling, together with the empirical robustness in Table 5, indicates that explicit dynamic inference of $k$ yields at most marginal gains and is not essential for the effectiveness of REALIGN.

**Impact of Training Data Quantity.** Fig. 4(b) shows how performance on the MECCANO dataset varies with the number of training videos. Across all data scales, our REALIGN model outperforms previous state-of-the-art methods. Even with only 2-5 videos per task, it achieves higher F1-scores than competing approaches. Performance continues to rise with more data, reaching **59.7** F1 at 17 videos. In contrast, prior methods improve more slowly and remain consistently behind, underscoring the data efficiency, scalability, and robustness of our approach.

**Comparison with Action Segmentation Methods.** While related, Procedure learning (PL) differs from action segmentation (AS). PL identifies a consistent set of $K$ key steps across multiple videos of the same task, while AS only partitions a single video into actions without cross-video reasoning. Table 6 reports results of REALIGN compared with leading unsupervised AS models (Dvornik et al. (2023)) and OT-models (Chowdhury et al. (2024); Mahmood et al. (2025)). On ProceL (Elhamifar & Huynh (2020)) and CrossTask (Zhukov et al. (2019)), RE-

Table 6: Comparison with SOTA unsupervised AS methods. Note '-' denotes that the authors have not provided any data on those metrics.

| Action Segmentation (AS) benchmark | ProceL | | | CrossTask | | |
|---|---|---|---|---|---|---|
| | P | R | F1 | P | R | F1 |
| Elhamifar & Naing (2019) | - | - | 29.8 | - | - | - |
| Elhamifar & Huynh (2020) | 9.5 | 26.7 | 14.0 | 10.1 | 41.6 | 16.3 |
| Fried et al. (2020) | - | - | - | - | 28.8 | - |
| Shen et al. (2021) | 16.5 | 31.8 | 21.1 | 15.2 | 35.5 | 21.0 |
| Dvornik et al. (2022) | - | - | - | - | - | 25.3 |
| StepFormer | 18.3 | 28.1 | 21.9 | 22.1 | 42 | 28.3 |
| OPEL | 33.6 | 36.3 | 34.9 | 35.6 | 34.8 | 35.1 |
| RGWOT | 42.2 | 46.7 | 44.3 | 40.4 | 40.7 | 40.4 |
| *REALIGN* (R-FGWOT) | 53.5 | 60.4 | 56.7 | 60.2 | 61.2 | 60.6 |
| *REALIGN* (R-FPGWOT) | 54.4 | 61.5 | 57.6 | 60.9 | 61.9 | 61.4 |

ALIGN achieves the highest precision (**60.9**), recall (**61.9**), and F1 score (**61.4**), significantly outperforming prior approaches. REALIGN strikes a good balance between precision and recall, showing its strength in avoiding degenerate solutions.

**Additional analyses and results.** Several additional studies in Appendix include detailed hyperparameter settings (App. Sec. A.2), sensitivity analyses (e.g., loss weights and number of clusters $k$) (App.Sec. A.8.5), runtime comparisons (App.Sec. A.3), the explicit graph-cut energy formulation (App.Sec. A.8.1), diagnostics for avoiding degenerate solutions, and full quantitative results (App.Sec. A.6)on all subtasks of EgoProceL (Bansal et al. (2022)), ProceL (Elhamifar & Huynh (2020)), and CrossTask (Zhukov et al. (2019)).

## 6 CONCLUSION

In this work, we presented REALIGN, an unsupervised procedure learning based on *Regularized Fused Partial Gromov-Wasserstein Optimal Transport*. By jointly modeling feature similarity and temporal structure under relaxed marginal constraints, our method overcomes shortcomings of prior OT-based approaches that relied on strictly balanced frame-to-frame mappings or monotonic assumptions. Through the integration of Laplace priors, structural regularization, and contrastive stabilization, REALIGN achieves robust alignment while discarding background or redundant frames. Results across large-scale egocentric and third-person benchmarks demonstrate consistent improvements, with up to **11.6% (4.45pp)** gains in F1-score and **19.6% (4.73pp)** in IoU on EgoProceL, and an average **41% (17.15pp)** F1 boost on ProceL and CrossTask compared to existing SOTA, while producing interpretable transport maps that faithfully preserve key-step ordering. Beyond alignment accuracy, our formulation proves to be data-efficient and scalable, achieving superior performance with limited training data. Our proposed framework establishes a strong foundation for procedure learning under real-world conditions, opening avenues for future extensions in multi-modal alignment and continual learning.

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

# A APPENDIX

## A.1 DERIVATION OF THE R-FPGWOT'S OPTIMAL TRANSPORT MATRIX $(\hat{T}_{\lambda_1, \lambda_2})$

This appendix provides a complete derivation of our optimization scheme for *Regularized-Fused Partial Gromov-Wasserstein Optimal Transport* (R-FPGWOT). We (i) fix notation, (ii) state the objective, (iii) derive a majorization-minimization (MM) inner problem with a Sinkhorn-like solution, (iv) cover unbalanced (partial) transport, (v) treat the 'virtual' sink frame, and (vi) give convergence statements for the inner loop (unbalanced Sinkhorn) and the outer MM iterations. We also clarify the positive semidefiniteness (PSD) requirement for temporal structure matrices and provide two safe choices.

**Notation.** Let $X = \{x_i\}_{i=1}^N$ and $Y = \{y_j\}_{j=1}^M$ denote frame embeddings for two videos, stacked as $\boldsymbol{X} \in \mathbb{R}^{N \times d}$ and $\boldsymbol{Y} \in \mathbb{R}^{M \times d}$. We optimize a nonnegative coupling $\hat{T} \in \mathbb{R}_+^{(N+1) \times (M+1)}$ augmented by an extra row/column (index $N{+}1$ and $M{+}1$) that represents a 'virtual' sink for unmatched mass. Let $\mathbf{1}_k$ be the $k$-vector of ones.

**Costs.**

- *Appearance* (inter-sequence) cost $\boldsymbol{C} \in \mathbb{R}_+^{(N+1) \times (M+1)}$, e.g., cosine/Euclidean distances between frame embeddings (with a large finite cost to/from the virtual entry).

- *Structure* (intra-sequence) matrices $\boldsymbol{C}^x \in \mathbb{R}^{(N+1) \times (N+1)}$ and $\boldsymbol{C}^y \in \mathbb{R}^{(M+1) \times (M+1)}$ encoding temporal proximity.

**Priors and marginals.** Let $\hat{\boldsymbol{Q}} \in \mathbb{R}_{++}^{(N+1) \times (M+1)}$ be a strictly positive prior (constructed from our mixed Laplace priors plus a virtual row/column; see main text). Let $\boldsymbol{\alpha} \in \Delta^{N+1}$ and $\boldsymbol{\beta} \in \Delta^{M+1}$ be target row/column marginals (including virtual mass). We write $\mathrm{KL}(\boldsymbol{A}\|\boldsymbol{B}) = \sum_{ij} A_{ij} \log \frac{A_{ij}}{B_{ij}} - A_{ij} + B_{ij}$ and extend KL to vectors entrywise.

I. R-FPGWOT OBJECTIVE.

From the duality theory, for each pair $(\xi_1, \xi_2)$ there exists a corresponding pair $(\lambda_1, \lambda_2)$ with $\lambda_1 > 0, \lambda_2 > 0$, such that

$$l_{\xi_1, \xi_2}^R(\boldsymbol{X}, \boldsymbol{Y}) = l_{\lambda_1, \lambda_2}^R(\boldsymbol{X}, \boldsymbol{Y}).$$

We minimize a fused cost that combines appearance ($\boldsymbol{C}$) with a Gromov-Wasserstein style relational term built from $\boldsymbol{C}^x, \boldsymbol{C}^y$ under *partial* (unbalanced) marginal penalties and two regularizers: an IDM-style structural reward and a prior KL tether. In the partial (unbalanced) setting, equality constraints are replaced by marginal KL penalties with weight $\tau > 0$ and the MM (majorization–minimization) subproblem can be written in the *constrained* form:

$$l_{\xi_1, \xi_2}^{R-FPGW}(\boldsymbol{X}, \boldsymbol{Y}) = \min_{\hat{T} \geq 0} (1-\rho) \langle \boldsymbol{C}, \hat{T} \rangle + \rho \underbrace{\langle \boldsymbol{C}^x \hat{T} \boldsymbol{C}^y, \hat{T} \rangle}_{\text{GW term}} + \tau \Big[ \mathrm{KL}(\hat{T}\mathbf{1}\|\alpha) + \mathrm{KL}(\hat{T}^\top \mathbf{1}\|\beta) \Big]$$

$$\text{s.t. } M(\hat{T}) \geq \xi_1, \qquad \mathrm{KL}(\hat{T}\|\hat{Q}) \leq \xi_2. \tag{A1}$$

Taking the Lagrangian of Eq. A1 introduces multipliers $\lambda_1, \lambda_2$, yielding the equivalent *penalized* R-FPGWOT objective:

$$\min_{\hat{T} \geq 0} (1-\rho)\langle \boldsymbol{C}, \hat{T} \rangle + \rho \underbrace{\langle \boldsymbol{C}^x \hat{T} \boldsymbol{C}^y, \hat{T} \rangle}_{\text{GW-like fused term}} - \lambda_1 M(\hat{T}) + \lambda_2 \mathrm{KL}(\hat{T}\|\hat{Q})$$

$$+ \tau \Big( \mathrm{KL}(\hat{T}\mathbf{1}_{M+1}\|\alpha) + \mathrm{KL}(\hat{T}^\top \mathbf{1}_{N+1}\|\beta) \Big), \tag{A2}$$

where $M(\hat{T})$ is the IDM mixture reward used in the main paper to promote near-diagonal concentration and sharp ridges (we treat it as a linear reward in $\hat{T}$).

**Remark (PSD requirement and two safe choices).** The standard convex linearization that leads to $\nabla_{\boldsymbol{T}} \langle \boldsymbol{C}^x \boldsymbol{T} \, \boldsymbol{C}^y, \boldsymbol{T} \rangle = \boldsymbol{C}^x \boldsymbol{T} \, (\boldsymbol{C}^y)^\top + (\boldsymbol{C}^x)^\top \boldsymbol{T} \, \boldsymbol{C}^y$ simplifies to $2 \, \boldsymbol{C}^x \boldsymbol{T} \, \boldsymbol{C}^y$ only when $\boldsymbol{C}^x, \boldsymbol{C}^y$ are symmetric. Moreover, classical global convexity arguments on the quadratic form $t^\top (\boldsymbol{C}^y \otimes \boldsymbol{C}^x) t$ assume $\boldsymbol{C}^x, \boldsymbol{C}^y \succeq 0$. Temporal *distance* matrices are generally *not* PSD. We therefore adopt one of the following two remedies (both are acceptable in theory and practice):

(A) **Kernelized structure (default).** Define $\boldsymbol{C}^x = \big[ k(|i - i'|) \big]_{i,i'}$ and $\boldsymbol{C}^y = \big[ k(|j - j'|) \big]_{j,j'}$ using a PSD kernel $k$, e.g., Gaussian or Laplace. Then $\boldsymbol{C}^x, \boldsymbol{C}^y$ are symmetric PSD, and all convexity statements below hold globally.

(B) **Surrogate MM majorizer (no PSD needed).** Keep arbitrary bounded $\boldsymbol{C}^x, \boldsymbol{C}^y$ (e.g., raw temporal distances) and treat the fused quadratic term with a *local* quadratic upper bound (majorization) based on the Lipschitz continuity of its gradient. The outer loop then minimizes a convex surrogate each iteration; convergence is monotone in the surrogate objective (standard MM).

We use (A) in all experiments and state theorems for (A). For completeness, we also include the (B) version (Lemma 1) that requires only boundedness of $\boldsymbol{C}^x, \boldsymbol{C}^y$.

## II. MM LINEARIZATION OF THE FUSED TERM

Let $F(\boldsymbol{T}) = \langle \boldsymbol{C}^x \boldsymbol{T} \, \boldsymbol{C}^y, \boldsymbol{T} \rangle$. At outer iterate $\hat{\boldsymbol{T}}^{(s)}$ we build a first-order majorizer

$$F(\boldsymbol{T}) \;\leq\; F\big(\hat{\boldsymbol{T}}^{(s)}\big) \;+\; \big\langle \boldsymbol{G}^{(s)}, \, \boldsymbol{T} - \hat{\boldsymbol{T}}^{(s)} \big\rangle \;+\; \frac{L}{2} \, \big\| \boldsymbol{T} - \hat{\boldsymbol{T}}^{(s)} \big\|_F^2, \quad \boldsymbol{G}^{(s)} := \nabla F\big(\hat{\boldsymbol{T}}^{(s)}\big), \quad \text{(A3)}$$

where $L$ is any Lipschitz constant of $\nabla F$. Keeping the quadratic term with weight $L/2$ yields a bona fide MM majorizer and theoretical monotonicity (Options A and B). In our implementation, following common FGW practice, we set the prox weight implicitly small and absorb it into the linearized cost (heuristic "pure linearization"); this preserves monotonicity under Option A and works robustly in Option B in practice, although the formal MM upper-bound is then approximate.

$$\widetilde{\boldsymbol{D}}^{(s)} \;=\; (1 - \rho) \, \boldsymbol{C} \;+\; \rho \, \boldsymbol{G}^{(s)}.$$

**When $\boldsymbol{C}^x, \boldsymbol{C}^y$ are symmetric PSD (Option A).** Then $\boldsymbol{G}^{(s)} = 2 \, \boldsymbol{C}^x \hat{\boldsymbol{T}}^{(s)} \boldsymbol{C}^y$ and one can set $L = 2 \, \|\boldsymbol{C}^x\|_2 \, \|\boldsymbol{C}^y\|_2$. This recovers the widely used linearization in FGW.

**When $\boldsymbol{C}^x, \boldsymbol{C}^y$ are not PSD (Option B).** We still have a valid local majorizer: the gradient of $F$ is Lipschitz with $L \leq \|\boldsymbol{C}^x\|_2 \, \|\boldsymbol{C}^y\|_2 + \|(\boldsymbol{C}^x)^\top\|_2 \, \|(\boldsymbol{C}^y)^\top\|_2$, and $\boldsymbol{G}^{(s)} = \boldsymbol{C}^x \hat{\boldsymbol{T}}^{(s)} (\boldsymbol{C}^y)^\top + (\boldsymbol{C}^x)^\top \hat{\boldsymbol{T}}^{(s)} \boldsymbol{C}^y$. Thus, Eq. A3 is a valid MM upper bound without any PSD assumption. We state this explicitly in Lemma 1.

## III. INNER (CONVEX) SUBPROBLEM AND GIBBS KERNEL FORMULATION

We start from the unconstrained KL-regularized formulation (ignoring additive constants). The objective combines (i) linearized cost, (ii) IDM reward, (iii) prior-KL, and (iv) marginal KL penalties (for the unbalanced case).

**General inner problem.** Fixing $\widetilde{\boldsymbol{D}}^{(s)}$ and treating the IDM reward $-\lambda_1 M(\boldsymbol{T})$ as a linear negative cost (i.e., a positive "score" added to the kernel exponent), the inner subproblem at iteration $s$ is

$$\min_{\hat{\boldsymbol{T}} \geq 0} \quad \langle \hat{\boldsymbol{T}}, \, \widetilde{\boldsymbol{D}}^{(s)} \rangle \;-\; \lambda_1 \, M(\hat{\boldsymbol{T}}) \;+\; \lambda_2 \, \text{KL}(\hat{\boldsymbol{T}} \| \hat{\boldsymbol{Q}}) \;+\; \tau \Big( \text{KL}(\hat{\boldsymbol{T}} \, \mathbf{1}_{M+1} \| \boldsymbol{\alpha}) + \text{KL}(\hat{\boldsymbol{T}}^\top \mathbf{1}_{N+1} \| \boldsymbol{\beta}) \Big).$$

$$\text{(A4)}$$

This is strictly convex in $\hat{\boldsymbol{T}}$ (for $\lambda_2 > 0$). Thus, the MM subproblem admits two equivalent perspectives: the constrained $(\xi_1, \xi_2)$ formulation and the penalized $(\lambda_1, \lambda_2)$ formulation, linked through duality.

**Lagrangian.** Dropping constants, the inner optimization problem is:

$$\begin{aligned}
\mathcal{L}(\hat{\boldsymbol{T}}) = &\sum_{i,j} \widetilde{d}_{ij}^{(s)} t_{ij} \;-\; \lambda_1 \sum_{i,j} s_{ij} \, t_{ij} \;+\; \lambda_2 \sum_{i,j} t_{ij} \log \frac{t_{ij}}{q_{ij}} \\
&+ \tau \Big( \sum_i \big[ r_i \log \tfrac{r_i}{\alpha_i} - r_i + \alpha_i \big] \;+\; \sum_j \big[ c_j \log \tfrac{c_j}{\beta_j} - c_j + \beta_j \big] \Big),
\end{aligned} \quad \text{(A5)}$$

with row/column sums: $r_i = (\hat{T}\mathbf{1})_i$ and $c_j = (\hat{T}^\top \mathbf{1})_j$.

$t_{ij}$ is transport $(i,j)$ entry and $q_{ij} > 0$ is prior $(i,j)$ entry of $\hat{T}$ and $\hat{Q}$ respectively and $\lambda_2 > 0$ is the temperature. $s_{ij}$ is the IDM score injection (a linear function of $\hat{T}$):

$$s_{ij} = \lambda_1 \left( \frac{1}{\left(\frac{i}{N+1} - \frac{j}{M+1}\right)^2 + 1} + \frac{1}{\frac{1}{2}d_m + 1} \right), \quad d_m = \left(\frac{i-i_o}{N+1}\right)^2 + \left(\frac{j-j_o}{M+1}\right)^2.$$

**Stationarity (KKT).**

Differentiating Eq. A5 and setting $\partial\mathcal{L}/\partial t_{ij} = 0$ yields:

$$\frac{\partial\mathcal{L}}{\partial t_{ij}} = \widetilde{d}_{ij}^{(s)} - s_{ij} + \lambda_2\left(\log\frac{t_{ij}}{q_{ij}} + 1\right) + \tau\left(\log\frac{r_i}{\alpha_i} + \log\frac{c_j}{\beta_j}\right) = 0. \tag{A6}$$

**Gibbs form.**

Rearranging Eq. A6 and dropping additive constants that are absorbed in scaling, the KKT stationarity yields a Gibbs form

$$t_{ij} = K_{ij}^{(s)} \left(\frac{\alpha_i}{(\hat{T}\mathbf{1})_i}\right)^\kappa \left(\frac{\beta_j}{(\hat{T}^\top\mathbf{1})_j}\right)^\kappa, \qquad \kappa := \frac{\tau}{\lambda_2}, \tag{A7}$$

with strictly positive kernel

$$K_{ij}^{(s)} = q_{ij}\,\exp\left(\frac{s_{ij}^{\lambda_1} - \widetilde{d}_{ij}^{(s)}}{\lambda_2}\right), \quad s_{ij}^{\lambda_1} \text{ is the IDM score for entry } (i,j). \tag{A8}$$

**Unbalanced Sinkhorn Scaling (Partial OT).**

By Sinkhorn's theorem (Theorem A[1], for any matrix with strictly positive entries, there exist unique (up to a scalar) positive scaling vectors that enforce the marginal constraints. Specifically, since $q_{ij} > 0$ and the exponent is finite, each $K_{ij} > 0$. Therefore, there exist unique positive scaling vectors $\boldsymbol{u} \in \mathbb{R}^{N+1}, \boldsymbol{v} \in \mathbb{R}^{M+1}$ such that:

$$\hat{T} = \mathrm{Diag}(\boldsymbol{u})\,\boldsymbol{K}^{(s)}\,\mathrm{Diag}(\boldsymbol{v}), \qquad (\hat{T}\mathbf{1})_i = u_i\,(\boldsymbol{K}^{(s)}\boldsymbol{v})_i, \quad (\hat{T}^\top\mathbf{1})_j = v_j\,((\boldsymbol{K}^{(s)})^\top\boldsymbol{u})_j. \tag{A9}$$

with the *unbalanced* Sinkhorn updates

$$\boldsymbol{u} \leftarrow \left(\frac{\boldsymbol{\alpha}}{\boldsymbol{K}^{(s)}\boldsymbol{v}}\right)^\kappa, \qquad \boldsymbol{v} \leftarrow \left(\frac{\boldsymbol{\beta}}{(\boldsymbol{K}^{(s)})^\top\boldsymbol{u}}\right)^\kappa, \qquad \kappa = \frac{\tau}{\lambda_2} \in (0, \infty). \tag{A10}$$

For $\tau \to \infty$ ($\kappa \to 1$) this reduces to the balanced Sinkhorn updates; for finite $\tau$ it is a standard unbalanced setting.

**Virtual sink frame.** The last row/column of $\hat{T}$ (index $N+1/M+1$) correspond to the virtual mass. They are handled identically by Eq. A10. In practice, we budget sufficient virtual mass in $\boldsymbol{\alpha}, \boldsymbol{\beta}$ and assign large (but finite) appearance costs to discourage non-essential matches unless needed.

**Convergence.**

Assume $\boldsymbol{K}^{(s)}$ has strictly positive entries bounded as $0 < m \le K_{ij}^{(s)} \le M < \infty$ and $\boldsymbol{\alpha}, \boldsymbol{\beta}$ have strictly positive components (including virtual mass). Then the unbalanced updates Eq. A10 are contractive in the Hilbert projective metric and converge to the unique minimizer of Eq. A4. This is a standard result for (un)balanced Sinkhorn with KL-penalized marginals. In practice, this paper uses only 20 iterations, since earlier studies have shown that a small number of iterations is sufficient for effective convergence (Cuturi (2013)).

---

[1]**Balanced Sinkhorn existence (classical).** For any positive matrix $\boldsymbol{A}$, there exist positive diagonal scalings that match prescribed positive marginals (up to a common factor) (Sinkhorn (1967); Borobia & Cantó (1998)). In the unbalanced (KL-penalized) setting used here, the fixed-point equations Eq. A10 arise from KKT stationarity and admit unique positive solutions under bounded positive kernels; see, e.g., unbalanced OT analyses (Chizat et al. (2018)).

**Proposition 1** (Inner convergence). *Under the bounded positive kernel condition above, the iterations Eq. A10 converge to the unique optimizer of Eq. A4.*

*Proof Proposition.* The updates are compositions of positive linear maps with entrywise powers $\kappa \in (0, 1]$; the former are contractive in the Hilbert projective metric with diameter bounded by $\log(M/m)$, and the latter are nonexpansive. Banach's fixed-point theorem yields convergence to the unique fixed point, which is the KKT solution of Eq. A4. $\qquad\square$

**Balanced vs unbalanced cases.**

- In the *balanced* setting ($\tau \to \infty$, hence $\kappa \to 1$), the KL penalties enforce exact marginal constraints, and the updates reduce to classical Sinkhorn scaling Eq. A10.
- In the *unbalanced (partial)* setting ($0 < \tau < \infty$), the generalized exponent $\kappa = \tau/\lambda_2$ appears, yielding the unbalanced Sinkhorn iterations Eq. A10.

IV. OUTER MM: MONOTONE DECREASE

Let $\mathcal{J}$ be the full objective Eq. A2 and $\hat{\boldsymbol{T}}^{(s)}$ the current iterate. Define the surrogate at $\hat{\boldsymbol{T}}^{(s)}$ by replacing $F(\boldsymbol{T})$ with its majorizer Eq. A3 and solving the inner problem exactly to get $\hat{\boldsymbol{T}}^{(s+1)}$.

Recall the fused quadratic form $F(\hat{\boldsymbol{T}}) = \langle \boldsymbol{C}^x \hat{\boldsymbol{T}} \boldsymbol{C}^y, \hat{\boldsymbol{T}} \rangle$, and its linearization at $\hat{\boldsymbol{T}}^{(s)}$:

$$\widetilde{F}^{(s)}(\hat{\boldsymbol{T}}) \ := \ F(\hat{\boldsymbol{T}}^{(s)}) + \langle \boldsymbol{G}^{(s)}, \hat{\boldsymbol{T}} - \hat{\boldsymbol{T}}^{(s)} \rangle, \quad \boldsymbol{G}^{(s)} = 2\, \boldsymbol{C}^x \hat{\boldsymbol{T}}^{(s)} \boldsymbol{C}^y.$$

Define the residual $\Delta^{(s)}(\hat{\boldsymbol{T}}) := F(\hat{\boldsymbol{T}}) - \widetilde{F}^{(s)}(\hat{\boldsymbol{T}})$.

**PSD (Option A).** If $\boldsymbol{C}^x, \boldsymbol{C}^y$ are symmetric PSD, then

$$F(\boldsymbol{T}) - \left( F(\hat{\boldsymbol{T}}^{(s)}) + \left\langle 2\, \boldsymbol{C}^x \hat{\boldsymbol{T}}^{(s)} \boldsymbol{C}^y, \boldsymbol{T} - \hat{\boldsymbol{T}}^{(s)} \right\rangle \right) \ \leq \ \|\boldsymbol{C}^x\|_2 \, \|\boldsymbol{C}^y\|_2 \, \|\boldsymbol{T} - \hat{\boldsymbol{T}}^{(s)}\|_F^2,$$

so the surrogate is a global upper bound tight at $\hat{\boldsymbol{T}}^{(s)}$, and $\mathcal{J}(\hat{\boldsymbol{T}}^{(s+1)}) \leq \mathcal{J}(\hat{\boldsymbol{T}}^{(s)})$.

**Non-PSD (Option B).** Even without PSD, we have a *local* quadratic majorizer:

**Lemma 1** (Local MM majorizer without PSD). *Let $\boldsymbol{C}^x, \boldsymbol{C}^y$ be arbitrary bounded matrices. Then $\nabla F$ is Lipschitz with some finite $L$ and*

$$F(\boldsymbol{T}) \ \leq \ F(\hat{\boldsymbol{T}}^{(s)}) + \left\langle \boldsymbol{G}^{(s)}, \boldsymbol{T} - \hat{\boldsymbol{T}}^{(s)} \right\rangle + \frac{L}{2} \|\boldsymbol{T} - \hat{\boldsymbol{T}}^{(s)}\|_F^2,$$

*with $\boldsymbol{G}^{(s)} = \boldsymbol{C}^x \hat{\boldsymbol{T}}^{(s)} (\boldsymbol{C}^y)^\top + (\boldsymbol{C}^x)^\top \hat{\boldsymbol{T}}^{(s)} \boldsymbol{C}^y$. Minimizing this surrogate yields a monotone decrease in the surrogate objective; hence, the outer MM produces a non-increasing sequence of surrogate values with standard MM convergence guarantees to a stationary point.*

In both options, the exact solution of the strictly convex inner problem yields a unique $\hat{\boldsymbol{T}}^{(s+1)}$.

*Proof of Lemma 1.* Using $\langle \boldsymbol{C}^x \boldsymbol{T} \boldsymbol{C}^y, \boldsymbol{T} \rangle = \langle \boldsymbol{T}, (\boldsymbol{C}^x)^\top \boldsymbol{T} \boldsymbol{C}^y \rangle$, the Frechet derivative is $\nabla F(\boldsymbol{T}) = \boldsymbol{C}^x \boldsymbol{T} (\boldsymbol{C}^y)^\top + (\boldsymbol{C}^x)^\top \boldsymbol{T} \boldsymbol{C}^y$. For any $\boldsymbol{T}_1, \boldsymbol{T}_2$,

$$\|\nabla F(\boldsymbol{T}_1) - \nabla F(\boldsymbol{T}_2)\|_F \leq \|\boldsymbol{C}^x\|_2 \|(\boldsymbol{C}^y)^\top\|_2 \|\boldsymbol{T}_1 - \boldsymbol{T}_2\|_F + \|(\boldsymbol{C}^x)^\top\|_2 \|\boldsymbol{C}^y\|_2 \|\boldsymbol{T}_1 - \boldsymbol{T}_2\|_F,$$

so one can take $L = \|\boldsymbol{C}^x\|_2 \|(\boldsymbol{C}^y)^\top\|_2 + \|(\boldsymbol{C}^x)^\top\|_2 \|\boldsymbol{C}^y\|_2$. The descent lemma then gives the quadratic upper bound. $\qquad\square$

Combining either option with the strict convexity and coercivity of the inner program gives:

**Theorem 1** (Outer monotonicity). *If each inner subproblem Eq. A4 is solved exactly, then the sequence $\{\hat{\boldsymbol{T}}^{(s)}\}$ generated by the MM procedure satisfies $\mathcal{J}(\hat{\boldsymbol{T}}^{(s+1)}) \leq \mathcal{J}(\hat{\boldsymbol{T}}^{(s)})$ in Option (A), and it monotonically decreases the MM surrogate in Option (B). Every limit point is a stationary point of the respective (true or surrogate) objective.*

## V. Algorithmic Summary (Practical Implementation)

---

**Algorithm 1** R-FPGWOT with IDM Priors and Unbalanced Sinkhorn

---

1: **Input:** costs $C, C^x, C^y$, prior $\hat{Q}$, weights $\rho, \lambda_1, \lambda_2, \tau$, annealing schedule $\phi$.
2: Initialize $\hat{T}^{(0)}$ (e.g., $\hat{Q}$), set $s \leftarrow 0$.
3: **repeat**                                                                    ▷ Outer MM
4:     **Outer gradient (linearization):**

$$G^{(s)} \leftarrow \begin{cases} 2\,C^x \hat{T}^{(s)} C^y, & \text{(Option A: symmetric PSD)} \\ C^x \hat{T}^{(s)} (C^y)^\top + (C^x)^\top \hat{T}^{(s)} C^y, & \text{(Option B)} \end{cases}$$

5:     $\widetilde{D}^{(s)} \leftarrow (1-\rho)\,C + \rho\,G^{(s)}$.
6:     $s_{ij}^{\lambda_1} \leftarrow \lambda_1 \left( \left[ (\frac{i}{N+1} - \frac{j}{M+1})^2 + 1 \right]^{-1} + \left[ \frac{1}{2} d_m + 1 \right]^{-1} \right)$
7:     Build kernel $K_{ij}^{(s)} \leftarrow q_{ij} \exp\left( \frac{s_{ij}^{\lambda_1} - \widetilde{D}_{ij}^{(s)}}{\lambda_2} \right)$                  ▷ cf. Eq. A8
8:     Initialize $u, v \leftarrow \mathbf{1}$;   $\kappa \leftarrow \tau/\lambda_2$
9:     **repeat**                                              ▷ Unbalanced Sinkhorn (Eq. A10)
10:        $u \leftarrow \left( \alpha./(K^{(s)} v) \right)^\kappa$,      $v \leftarrow \left( \beta./((K^{(s)})^\top u) \right)^\kappa$
11:     **until** Inner converged
12:     $\hat{T}^{(s+1)} \leftarrow \mathrm{Diag}(u)\,K^{(s)}\,\mathrm{Diag}(v)$
13:     $s \leftarrow s + 1$;   anneal $\phi$
14: **until** Outer convergence
15: **Return** $\hat{T}^{(s)}$.

---

## VI. Computational Complexity

Each inner iteration costs two matrix-vector products with $K^{(s)}$ and $(K^{(s)})^\top$, i.e., $O((N+1)(M+1))$. Forming $G^{(s)} = C^x \hat{T}^{(s)} (C^y)^\top + (C^x)^\top \hat{T}^{(s)} C^y$ (or $2\,C^x \hat{T}^{(s)} C^y$ in Option A) costs $O((N+1)(M+1))$ if $C^x, C^y$ are banded (temporal kernels), since it reduces to two banded-dense multiplies; otherwise it is $O((N+1)^2(M+1))$ but we avoid explicit dense Kronecker constructions. We use a small, fixed number of inner iterations (e.g., $\leq 25$) and 4–7 outer steps in practice. We stop the inner loop by relative marginal change ($\leq 10^{-3}$) and the outer loop by relative objective decrease ($\leq 10^{-4}$).

## VII. Additional Lemmas and Proofs

**Practical construction of structure matrices.** We provide two safe choices for the temporal/relational structure matrices $C^x \in \mathbb{R}_+^{(N+1)\times(N+1)}$ and $C^y \in \mathbb{R}_+^{(M+1)\times(M+1)}$.

*Option A (recommended; PSD kernels).* Kernelize temporal proximity so that the resulting Toeplitz matrices are symmetric positive semidefinite (PSD):

$$\begin{aligned} (C^x)_{ii'} &= \exp\left( -\frac{|i-i'|}{b_x} \right) \quad \text{or} \quad \exp\left( -\frac{(i-i')^2}{2\sigma_x^2} \right), \\ (C^y)_{jj'} &= \exp\left( -\frac{|j-j'|}{b_y} \right) \quad \text{or} \quad \exp\left( -\frac{(j-j')^2}{2\sigma_y^2} \right). \end{aligned} \tag{A11}$$

This guarantees symmetry and PSD, which validates the global convexity route used in the MM derivation.

*Option B (non-PSD distances; surrogate MM).* If raw temporal *distances* must be used (which are generally not PSD), keep them bounded and rely on the local quadratic majorizer in Lemma 2 (Option B case). The algorithm remains an MM scheme on a surrogate upper bound and enjoys a monotone decrease of the surrogate.

**First-order majorization of the fused term.** Let $F(\hat{\boldsymbol{T}}) = \langle \boldsymbol{C}^x \hat{\boldsymbol{T}} \boldsymbol{C}^y, \hat{\boldsymbol{T}} \rangle$.

**Lemma 2** (First-order majorization of $F$). *At an outer iterate $\hat{\boldsymbol{T}}^{(s)}$, define*

$$
\boldsymbol{G}^{(s)} = \begin{cases} 2\,\boldsymbol{C}^x \hat{\boldsymbol{T}}^{(s)} \boldsymbol{C}^y, & \text{if } \boldsymbol{C}^x, \boldsymbol{C}^y \text{ are symmetric PSD (Option A)}, \\ \boldsymbol{C}^x \hat{\boldsymbol{T}}^{(s)} (\boldsymbol{C}^y)^\top + (\boldsymbol{C}^x)^\top \hat{\boldsymbol{T}}^{(s)} \boldsymbol{C}^y, & \text{otherwise (Option B)}. \end{cases}
$$

*Then there exists $L > 0$ (a Lipschitz constant of $\nabla F$) such that for all $\hat{\boldsymbol{T}}$,*

$$
F(\hat{\boldsymbol{T}}) \leq F(\hat{\boldsymbol{T}}^{(s)}) + \left\langle \boldsymbol{G}^{(s)}, \hat{\boldsymbol{T}} - \hat{\boldsymbol{T}}^{(s)} \right\rangle + \frac{L}{2} \|\hat{\boldsymbol{T}} - \hat{\boldsymbol{T}}^{(s)}\|_F^2. \tag{A12}
$$

*In Option A, one can take $L = 2\|\boldsymbol{C}^x\|_2 \|\boldsymbol{C}^y\|_2$, and the inequality is a* global *upper bound with $\boldsymbol{G}^{(s)} = 2\,\boldsymbol{C}^x \hat{\boldsymbol{T}}^{(s)} \boldsymbol{C}^y$.*

*Proof of Lemma 2.* Option A. If $\boldsymbol{C}^x, \boldsymbol{C}^y \succeq 0$ and symmetric, then $F(\hat{\boldsymbol{T}}) = \text{vec}(\hat{\boldsymbol{T}})^\top (\boldsymbol{C}^y \otimes \boldsymbol{C}^x)\,\text{vec}(\hat{\boldsymbol{T}})$ with a PSD Kronecker factor; $F$ is convex and $\nabla F(\hat{\boldsymbol{T}}) = 2\,\boldsymbol{C}^x \hat{\boldsymbol{T}} \boldsymbol{C}^y$. The descent lemma for convex $L$-smooth functions gives Eq. A12 with $L = 2\,\|\boldsymbol{C}^x\|_2 \|\boldsymbol{C}^y\|_2$.

*Option B.* Without PSD, $F$ is still smooth with $\nabla F(\hat{\boldsymbol{T}}) = \boldsymbol{C}^x \hat{\boldsymbol{T}} (\boldsymbol{C}^y)^\top + (\boldsymbol{C}^x)^\top \hat{\boldsymbol{T}} \boldsymbol{C}^y$. For any $\boldsymbol{T}_1, \boldsymbol{T}_2$,

$$
\|\nabla F(\boldsymbol{T}_1) - \nabla F(\boldsymbol{T}_2)\|_F \leq \|\boldsymbol{C}^x\|_2 \|(\boldsymbol{C}^y)^\top\|_2 \|\boldsymbol{T}_1 - \boldsymbol{T}_2\|_F + \|(\boldsymbol{C}^x)^\top\|_2 \|\boldsymbol{C}^y\|_2 \|\boldsymbol{T}_1 - \boldsymbol{T}_2\|_F,
$$

so one can take $L = \|\boldsymbol{C}^x\|_2 \|(\boldsymbol{C}^y)^\top\|_2 + \|(\boldsymbol{C}^x)^\top\|_2 \|\boldsymbol{C}^y\|_2$. Applying the descent lemma yields Eq. A12. $\qquad\square$

**Unique inner minimizer and KKT structure.** Fix $\widetilde{\boldsymbol{D}}^{(s)} = (1-\rho)\boldsymbol{C} + \rho\,\boldsymbol{G}^{(s)}$ and consider the inner convex subproblem:

$$
\min_{\hat{\boldsymbol{T}} \geq 0} \quad \langle \hat{\boldsymbol{T}}, \widetilde{\boldsymbol{D}}^{(s)} \rangle - \lambda_1 M(\hat{\boldsymbol{T}}) + \lambda_2 \text{KL}(\hat{\boldsymbol{T}} \| \hat{\boldsymbol{Q}}) + \tau\Big( \text{KL}(\hat{\boldsymbol{T}} \boldsymbol{1}_{M+1} \| \boldsymbol{\alpha}) + \text{KL}((\hat{\boldsymbol{T}})^\top \boldsymbol{1}_{N+1} \| \boldsymbol{\beta}) \Big),
$$
$$\tag{A13}$$

where $\hat{\boldsymbol{Q}} > 0$ elementwise, $\lambda_2 > 0$, and $\boldsymbol{\alpha}, \boldsymbol{\beta} > 0$ (including virtual mass entries).

**Lemma 3** (Unique inner minimizer). *Problem Eq. A13 is strictly convex on $\{\hat{\boldsymbol{T}} \geq 0\}$ and admits a unique minimizer. It is characterized by the KKT system that yields the Gibbs kernel form:*

$$
\hat{\boldsymbol{T}} = \text{Diag}(\boldsymbol{u})\, \boldsymbol{K}^{(s)}\, \text{Diag}(\boldsymbol{v}), \qquad \boldsymbol{u} = \Big( \boldsymbol{\alpha}./(\boldsymbol{K}^{(s)} \boldsymbol{v}) \Big)^\kappa, \quad \boldsymbol{v} = \Big( \boldsymbol{\beta}./((\boldsymbol{K}^{(s)})^\top \boldsymbol{u}) \Big)^\kappa, \quad \kappa = \frac{\tau}{\lambda_2},
$$

*with a positive kernel*

$$
K_{ij}^{(s)} = q_{ij}\, \exp\Big( \frac{s_{ij}^{\lambda_1} - \widetilde{d}_{ij}^{(s)}}{\lambda_2} \Big),
$$

*where $s_{ij}^{\lambda_1}$ denotes the IDM score (linear reward) and $\widetilde{d}_{ij}^{(s)}$ the linearized fused cost.*

*Proof of Lemma 3.* The function $X \mapsto \text{KL}(X \| \hat{\boldsymbol{Q}}) = \sum_{ij} X_{ij} \log \frac{X_{ij}}{q_{ij}} - X_{ij} + q_{ij}$ is strictly convex in $X$ on $\{X \geq 0\}$ provided $\hat{\boldsymbol{Q}} > 0$. The marginals map $\hat{\boldsymbol{T}} \mapsto (\hat{\boldsymbol{T}} \boldsymbol{1}, \hat{\boldsymbol{T}}^\top \boldsymbol{1})$ is linear, and $x \mapsto \text{KL}(x \| a)$ is strictly convex on $\mathbb{R}_+^n$ for $a > 0$, hence $\hat{\boldsymbol{T}} \mapsto \text{KL}(\hat{\boldsymbol{T}} \boldsymbol{1} \| \boldsymbol{\alpha}) + \text{KL}(\hat{\boldsymbol{T}}^\top \boldsymbol{1} \| \boldsymbol{\beta})$ is convex in $\hat{\boldsymbol{T}}$. Consequently, if $\lambda_2 > 0$ then $\Psi$ is a sum of a linear term and a strictly convex term, hence *strictly convex* in $\hat{\boldsymbol{T}}$. (Strict convexity is also ensured if $\lambda_2 = 0$ but additional entropic regularization is present directly on $\hat{\boldsymbol{T}}$; here we take $\lambda_2 > 0$.)

The coercivity holds because $\text{KL}(\cdot \| \hat{\boldsymbol{Q}})$ and the marginal KLs diverge when the masses grow unbounded or stray from $\alpha, \beta$, while all other terms are linear. Therefore, $\Psi$ admits a unique minimizer.

The KKT stationarity yields the Gibbs form:

$$
0 = \frac{\partial \Psi}{\partial t_{ij}} = \big( \widetilde{d}_{ij}^{(s)} - s_{ij}^{\lambda_1} \big) + \lambda_2 \Big( \log \frac{t_{ij}}{q_{ij}} \Big) + \tau \Big( \log \frac{(\hat{\boldsymbol{T}} \boldsymbol{1})_i}{\alpha_i} + \log \frac{(\hat{\boldsymbol{T}}^\top \boldsymbol{1})_j}{\beta_j} \Big).
$$

Rearranging gives the fixed-point equations above with $\kappa = \tau/\lambda_2$

$$\hat{t}_{ij} = q_{ij} \exp\left(-\frac{1}{\lambda_2}\left(\widetilde{d}_{ij}^{(s)} - s_{ij}^{\lambda_1}\right)\right) \left(\frac{\alpha_i}{(\hat{\boldsymbol{T}}\boldsymbol{1})_i}\right)^{\kappa} \left(\frac{\beta_j}{(\hat{\boldsymbol{T}}^{\top}\boldsymbol{1})_j}\right)^{\kappa}, \qquad \kappa = \frac{\tau}{\lambda_2}.$$

Define the positive kernel

$$K_{ij}^{(s)} = q_{ij} \exp\left(-\frac{1}{\lambda_2}\left(\widetilde{d}_{ij}^{(s)} - s_{ij}^{\lambda_1}\right)\right),$$

and scaling vectors $u_i = \left(\alpha_i/(\hat{\boldsymbol{T}}\boldsymbol{1})_i\right)^{\kappa}$, $v_j = \left(\beta_j/((\hat{\boldsymbol{T}})^{\top}\boldsymbol{1})_j\right)^{\kappa}$. Then the stationarity condition is equivalent to the multiplicative form

$$\hat{\boldsymbol{T}} = \operatorname{Diag}(\boldsymbol{u})\,\boldsymbol{K}^{(s)}\operatorname{Diag}(\boldsymbol{v}),$$

together with the self-consistency equations $\boldsymbol{u} = (\alpha./(\boldsymbol{K}^{(s)}\boldsymbol{v}))^{\kappa}$, $\boldsymbol{v} = (\beta./((\boldsymbol{K}^{(s)})^{\top}\boldsymbol{u}))^{\kappa}$, which are exactly the fixed-point relations stated in Eq. A10. By strict convexity, this solution is unique.

$\square$

**Lipschitz gap and data-term deviation.**

**Lemma 4** (Lipschitz gap for the fused term). *Let $F(\hat{\boldsymbol{T}}) = \langle \boldsymbol{C}^x \hat{\boldsymbol{T}} \boldsymbol{C}^y, \hat{\boldsymbol{T}} \rangle$. With $\boldsymbol{G}^{(s)}$ as in Lemma 2, there exists $L > 0$ such that for all $\hat{\boldsymbol{T}}$,*

$$F(\hat{\boldsymbol{T}}) \leq F(\hat{\boldsymbol{T}}^{(s)}) + \langle \boldsymbol{G}^{(s)}, \hat{\boldsymbol{T}} - \hat{\boldsymbol{T}}^{(s)} \rangle + \frac{L}{2}\|\hat{\boldsymbol{T}} - \hat{\boldsymbol{T}}^{(s)}\|_F^2.$$

*In Option A, one can take $L = 2\,\|\boldsymbol{C}^x\|_2\,\|\boldsymbol{C}^y\|_2$.*

**Corollary 1** (Deviation of the linearized data term). *Let $\widetilde{\boldsymbol{D}}(\hat{\boldsymbol{T}}) = (1-\rho)\boldsymbol{C} + \rho\,\boldsymbol{G}(\hat{\boldsymbol{T}})$ with $\boldsymbol{G}(\cdot)$ from Lemma 2. Then for all $\hat{\boldsymbol{T}}$,*

$$\langle \widetilde{\boldsymbol{D}}(\hat{\boldsymbol{T}}), \hat{\boldsymbol{T}} \rangle - \langle \widetilde{\boldsymbol{D}}^{(s)}, \hat{\boldsymbol{T}} \rangle \leq \rho\,\frac{L}{2}\,\|\hat{\boldsymbol{T}} - \hat{\boldsymbol{T}}^{(s)}\|_F^2,$$

*with $L$ as in Lemma 4. In Option A, $L = 2\|\boldsymbol{C}^x\|_2\|\boldsymbol{C}^y\|_2$.*

**Theorem 2** (Monotone decrease of the outer MM.). *Let $\mathcal{J}$ denote the full objective (Eq. A2). At outer step $s$, replace $F$ by the quadratic majorizer of Lemma 2 with constant $L$, and solve the inner problem exactly to obtain $\hat{\boldsymbol{T}}^{(s+1)}$.*

- *Option A (PSD): $\mathcal{J}(\hat{\boldsymbol{T}}^{(s+1)}) \leq \mathcal{J}(\hat{\boldsymbol{T}}^{(s)})$ (global upper bound; tight at $\hat{\boldsymbol{T}}^{(s)}$).*

- *Option B (non-PSD): the MM surrogate decreases monotonically; every limit point is a stationary point of the surrogate, and a first-order stationary point of the original objective under standard MM conditions.*

*Proof of Theorem 2.* By Lemma 4 and Corollary 1, the quadratic surrogate upper-bounds the fused term (globally in Option A; locally with an $L$-smooth bound in Option B) and is tight at $\hat{\boldsymbol{T}}^{(s)}$. Minimizing this surrogate plus convex KL penalties and the linear IDM reward cannot increase the (true or surrogate) objective. Coercivity of KL and nonnegativity of costs give existence of limit points; standard MM arguments then yield stationarity. $\square$

**Extensions and remarks.**

1. **Alternative $L$ and gradients.** If the loss is $L(a, b) = (a - b)^2$ with symmetric $\boldsymbol{C}^x, \boldsymbol{C}^y$, one obtains
$$\nabla_T F(T) = 2\,\boldsymbol{C}^x T \boldsymbol{1}\boldsymbol{1}^{\top} + 2\,\boldsymbol{1}\boldsymbol{1}^{\top} T \boldsymbol{C}^y - 4\,\boldsymbol{C}^x T \boldsymbol{C}^y,$$
and the same linearization/scaling applies after substituting $\widetilde{\boldsymbol{D}}$. The Lipschitz constant enters the same bounds.

2. **Stability.** The prior $\hat{Q}$ (strictly positive) prevents numerical underflow at low temperature $\lambda_2$, and the IDM reward injects mass near plausible alignments directly in the kernel exponent, which accelerates convergence.

3. **Stopping criteria.** We stop inner (unbalanced Sinkhorn) iterations when the relative marginal change is $\leq 10^{-3}$; the outer loop stops when the relative decrease in $\mathcal{J}$ is $\leq 10^{-4}$ or after a small maximum number of outer steps (e.g., 4-7).

4. **Consistency checks.** When $\tau \to \infty$ (balanced) and $\rho = 0$, we recover entropically regularized KOT with IDM and prior-KL; when $\tau \to \infty$ and $\rho > 0$, we recover entropic FGWOT; if $\lambda_1 = 0$, IDM is absent.

5. **Temperature and priors.** Smaller $\lambda_2$ sharpens the kernel; the prior $q_{ij}$ steers mass to the virtual sink when matches are weak and keeps all $K_{ij}^{(s)}$ strictly positive.

6. **Complexity.** Each inner iteration performs two matrix-vector products with $\boldsymbol{K}^{(s)}$ and $(\boldsymbol{K}^{(s)})^{\top}$, costing $O((N+1)(M+1))$. Forming $\boldsymbol{G}^{(s)} = 2\,\boldsymbol{C}^x\hat{\boldsymbol{T}}^{(s)}\boldsymbol{C}^y$ (Option A) or $\boldsymbol{C}^x\hat{\boldsymbol{T}}^{(s)}(\boldsymbol{C}^y)^{\top} + (\boldsymbol{C}^x)^{\top}\hat{\boldsymbol{T}}^{(s)}\boldsymbol{C}^y$ (Option B) is $O((N+1)(M+1))$ when $\boldsymbol{C}^x, \boldsymbol{C}^y$ are banded/sparse (typical for temporal kernels), as it reduces to two banded-dense multiplies; otherwise one should avoid explicit dense Kronecker constructions. Empirically, we use $\leq 25$ inner iterations and 3-6 outer steps.

## A.2 HYPER-PARAMETER SETTINGS

Table A1 lists the hyperparameters used for REALIGN.

Table A1: Hyper-parameter settings for REALIGN.

| Hyper-parameter | Value |
|---|---:|
| No. of key-steps ($k$) | 7 |
| No. of sampled frames ($N, M$) | 32 |
| No. of epochs | 10000 |
| Batch Size | 2 |
| Learning Rate ($\theta$) | $10^{-4}$ |
| Weight Decay | $10^{-5}$ |
| Window size ($\delta$) | 15 |
| No. of context frames | 2 |
| Context stride | 15 |
| Embedding Dimension | 128 |
| Gromov-Wasserstein weight ($\alpha$) | 0.5 |
| Entropy regularization weight ($\epsilon$) | 0.07 |
| Laplace scale parameter ($b$) | 3.0 (MECCANO, EPIC-Tents) |
| Laplace scale parameter ($b$) | 2.0 (for all other datasets) |
| Temperature | 0.5 |
| $\lambda_1$ | $\frac{1}{N+M}$ |
| $\lambda_2$ | $\frac{0.1*N*M}{4.0}$ |
| Margin ($\lambda_3$) | 2.0 |
| Threshold for virtual frame ($\zeta$) | $\frac{2*5}{N+M}$ |
| Optimizer | Adam (Adam et al. (2014) |
| $c_1$ | $\frac{1}{N*M}$ |
| $c_2$ | 0.5 |
| Coefficient for loss_inter ($c_3$) | 0.0001 |
| Maximum Sinkhorn Iterations | 20 |

## A.3 COMPUTE RESOURCES FOR EXPERIMENTS

For our experiments, appropriate computational resources were required to ensure efficient model training. We employed a single Nvidia A40 GPU; however, its full memory capacity was not nec-

essary. GPU memory usage was primarily determined by the batch size (bs). For instance, with a bs of 2, approximately 16 GB of GPU memory was sufficient. Training time depended on both the dataset size and the number of epochs (set to 10,000 in our case). Under this configuration, a dataset consisting of 15-20 videos (e.g., within the PC assembly or MECCANO domain) could be processed in approximately 12 hours. These resources enabled us to conduct the experiments effectively, ensuring optimal performance and reliable outcomes.

## A.4 DETAILED STATISTICS OF DATASET

Table A2 presents statistical analyses for each of the 16 (5+7+4) tasks in the EgoProceL dataset (Bansal et al. (2022)). Here, $N$ denotes the total number of videos, while $K$ represents the number of key-steps for each task. $u_n$ indicates the number of unique key-steps, and $g_n$ denotes the number of annotated key-steps for the $n^{th}$ video. Following the methodology in (Elhamifar & Naing (2019)), we report the following metrics:

*Foreground Ratio:* This metric measures the proportion of the total video duration occupied by key-steps. It reflects the prevalence of background actions in a task. A higher foreground ratio (closer to 1) corresponds to fewer background actions. It is defined as:

$$F = \frac{\sum_{n=1}^{N} \frac{t_k^n}{t_v^n}}{N} \tag{A14}$$

where $t_k^n$ and $t_v^n$ denote the durations of key-steps and the full video for the $n^{th}$ instance, respectively.

Table A2: Statistics of the EgoProceL dataset across different tasks.

| Task | Videos Count | Key-steps Count | Foreground Ratio | Missing Key-steps | Repeated Key-steps |
|---|---|---|---|---|---|
| PC Assembly (Bansal et al. (2022)) | 14 | 9 | 0.79 | 0.02 | 0.65 |
| PC Disassembly (Bansal et al. (2022)) | 15 | 9 | 0.72 | 0.00 | 0.60 |
| MECCANO (Ragusa et al. (2021)) | 20 | 17 | 0.50 | 0.06 | 0.32 |
| Epic-Tents (Jang et al. (2019)) | 29 | 12 | 0.63 | 0.14 | 0.73 |
| CMU-MMAC (De la Torre et al. (2009)) | | | | | |
| Brownie | 34 | 9 | 0.44 | 0.19 | 0.26 |
| Eggs | 33 | 8 | 0.26 | 0.05 | 0.26 |
| Pepperoni Pizza | 33 | 5 | 0.53 | 0.00 | 0.26 |
| Salad | 34 | 9 | 0.32 | 0.30 | 0.14 |
| Sandwich | 31 | 4 | 0.25 | 0.03 | 0.37 |
| EGTEAGAZE+ (Li et al. (2018)) | | | | | |
| Bacon and Eggs | 16 | 11 | 0.15 | 0.22 | 0.51 |
| Cheese Burger | 10 | 10 | 0.22 | 0.22 | 0.65 |
| Continental Breakfast | 12 | 10 | 0.23 | 0.20 | 0.36 |
| Greek Salad | 10 | 4 | 0.25 | 0.18 | 0.77 |
| Pasta Salad | 19 | 8 | 0.25 | 0.19 | 0.86 |
| Hot Box Pizza | 6 | 8 | 0.31 | 0.13 | 0.62 |
| Turkey Sandwich | 13 | 6 | 0.21 | 0.01 | 0.52 |

*Missing Key-steps (M):* This metric quantifies the proportion of omitted key-steps in each video.

$$M = 1 - \frac{\sum_{n=1}^{N} u_n}{KN}; \tag{A15}$$

Values range from 0 to 1, with higher values indicating more missing steps. This measure helps assess task feasibility when certain steps are skipped.

*Repeated Key-steps:* This metric captures the frequency of key-step repetition across videos:

$$R = 1 - \frac{\sum_{n=1}^{N} u_n}{\sum_{n=1}^{N} g_n} \tag{A16}$$

## A.5 THIRD-PERSON VIDEO PERSPECTIVE

In this study, we evaluate the performance of REALIGN across multiple third-person perspectives from CMU-MMAC (De la Torre et al. (2009)). Table A3 reports the per-frame F1-score and IoU for different exocentric views. Our experiments on exocentric videos yielded consistently strong results, confirming the robustness of the model when trained and tested under this setting. These findings not only highlight the effectiveness of our approach but also emphasize its relevance for practical scenarios involving both egocentric and exocentric video data.

Table A3: Comparison of third-person perspectives from CMU-MMAC (De la Torre et al. (2009)) against egocentric recordings. Egocentric view demonstrates markedly superior alignment quality, underscoring the strength of OT in capturing first-person task dynamics.

| View | **P** | **R** | **F1** | **IoU** |
|---|---|---|---|---|
| TP (Top) | 41.5 | 46.3 | 43.8 | 28.2 |
| TP (Back) | 44.7 | 49.8 | 47.1 | 31.0 |
| TP (LHS) | 50.2 | 55.4 | 52.7 | 35.9 |
| TP (RHS) | 43.0 | 48.2 | 45.5 | 29.4 |
| Egocentric | **61.2** | **58.4** | **59.7** | **43.7** |

## A.6 QUANTITATIVE RESULTS OF REALIGN ON DIFFERENT SUBTASKS ACROSS THE DATASETS

We present results for individual subtasks from egocentric datasets, including CMU-MMAC (De la Torre et al. (2009)) and EGTEA-GAZE+ (Li et al. (2018)), in Table A4, and for third-person exocentric datasets such as ProceL (Elhamifar & Huynh (2020)) and CrossTask (Zhukov et al. (2019)) in Table A5. This analysis provides a detailed evaluation across diverse settings, highlighting the performance of our model under different perspectives and task domains. The results demonstrate the versatility and effectiveness of our approach in handling a wide range of video types, thereby advancing the state of research in procedure learning.

Table A4: Results on individual subtasks of egocentric datasets.

(a) EGTEA-GAZE+ (Li et al. (2018))

| Method | Bacon Eggs | | Cheeseburger | | Breakfast | | Greek Salad | | Pasta Salad | | Pizza | | Turkey | |
|---|---|---|---|---|---|---|---|---|---|---|---|---|---|---|
| | F1 | IoU | F1 | IoU | F1 | IoU | F1 | IoU | F1 | IoU | F1 | IoU | F1 | IoU |
| R-FGWOT | 62.15 | 47.88 | 63.02 | 47.37 | 56.50 | 41.09 | 66.78 | 51.65 | 68.90 | 54.34 | 53.87 | 37.61 | 65.84 | 50.33 |
| R-FPGWOT | 66.74 | 53.62 | 66.98 | 51.95 | 57.95 | 42.50 | 66.79 | 51.65 | 70.99 | 56.78 | 53.97 | 37.67 | 66.23 | 50.68 |

(b) CMU-MMAC (De la Torre et al. (2009))

| Method | Brownie | | Eggs | | Pizza | | Salad | | Sandwich | |
|---|---|---|---|---|---|---|---|---|---|---|
| | F1 | IoU | F1 | IoU | F1 | IoU | F1 | IoU | F1 | IoU |
| R-FGWOT | 58.29 | 41.67 | 56.23 | 40.71 | 47.27 | 31.51 | 63.95 | 48.53 | 65.67 | 50.37 |
| R-FPGWOT | 58.52 | 41.92 | 56.72 | 41.11 | 48.00 | 32.22 | 69.23 | 52.27 | 66.16 | 50.89 |

Table A5: Results on individual subtasks of Third-person exocentric datasets.

(a) ProceL (Elhamifar & Huynh (2020))

| Methods | Clarinet | | PB&J Sandwich | | Salmon | | Jump Car | | Toilet | | Tire Change | |
|---|---|---|---|---|---|---|---|---|---|---|---|---|
| | F1 | IoU | F1 | IoU | F1 | IoU | F1 | IoU | F1 | IoU | F1 | IoU |
| R-FGWOT | 67.85 | 54.12 | 55.69 | 40.42 | 57.03 | 41.59 | 67.30 | 54.85 | 53.18 | 38.16 | 50.85 | 35.09 |
| R-FPGWOT | 68.48 | 54.82 | 56.46 | 40.97 | 58.57 | 43.25 | 67.99 | 55.65 | 55.27 | 40.10 | 51.69 | 36.13 |

| Methods | Tie-Tie | | Coffee | | iPhone Battery | | Repot Plant | | Chromescast | | CPR | |
|---|---|---|---|---|---|---|---|---|---|---|---|---|
| | F1 | IoU | F1 | IoU | F1 | IoU | F1 | IoU | F1 | IoU | F1 | IoU |
| R-FGWOT | 55.79 | 40.37 | 63.10 | 48.91 | 49.47 | 33.90 | 60.34 | 45.40 | 48.74 | 32.64 | 50.87 | 35.64 |
| R-FPGWOT | 55.97 | 40.53 | 64.52 | 50.15 | 49.49 | 33.90 | 60.74 | 45.92 | 49.71 | 33.41 | 52.30 | 36.82 |

(b) CrossTask (Zhukov et al. (2019))

| Methods | 16815 | | 23521 | | 40567 | | 44047 | | 44789 | | 53193 | |
|---|---|---|---|---|---|---|---|---|---|---|---|---|
| | F1 | IoU | F1 | IoU | F1 | IoU | F1 | IoU | F1 | IoU | F1 | IoU |
| R-FGWOT | 64.4 | 50.0 | 61.3 | 46.1 | 58.9 | 43.7 | 56.7 | 41.9 | 60.6 | 46.6 | 66.0 | 51.9 |
| R-FPGWOT | 65.1 | 50.4 | 61.5 | 46.3 | 59.6 | 44.5 | 57.7 | 42.7 | 61.9 | 48.1 | 66.5 | 52.3 |

| Methods | 59684 | | 71781 | | 76400 | | 77721 | | 87706 | | 91515 | |
|---|---|---|---|---|---|---|---|---|---|---|---|---|
| | F1 | IoU | F1 | IoU | F1 | IoU | F1 | IoU | F1 | IoU | F1 | IoU |
| R-FGWOT | 55.0 | 40.0 | 62.6 | 49.5 | 63.0 | 48.4 | 64.4 | 49.9 | 55.3 | 39.7 | 58.5 | 43.2 |
| R-FPGWOT | 56.1 | 40.1 | 63.6 | 50.3 | 63.5 | 48.9 | 65.2 | 50.6 | 55.9 | 40.3 | 58.9 | 43.8 |

| Methods | 94276 | | 95603 | | 105222 | | 105253 | | 109972 | | 113766 | |
|---|---|---|---|---|---|---|---|---|---|---|---|---|
| | F1 | IoU | F1 | IoU | F1 | IoU | F1 | IoU | F1 | IoU | F1 | IoU |
| R-FGWOT | 57.2 | 41.7 | 58.5 | 43.2 | 60.6 | 45.2 | 61.3 | 46.6 | 62.9 | 48.5 | 64.2 | 49.6 |
| R-FPGWOT | 57.6 | 42.0 | 58.6 | 43.3 | 61.6 | 46.0 | 62.5 | 47.8 | 63.9 | 49.7 | 65.1 | 50.5 |

## A.7 ADDITIONAL FUTURE APPLICATIONS

Leveraging multiple videos of the same task enables several practical applications. In procedure monitoring, the system can automatically verify whether each key step is performed correctly, flagging errors or deviations. For assistive guidance, it can localize the current step in real time and suggest the next, serving as an intelligent instruction system. In robotic automation, the framework learns procedural knowledge directly from observation, allowing robots to replicate tasks without explicit programming.

Beyond execution, the model also supports cross-modal transfer: annotations or cues (e.g., text or audio) can be propagated across aligned videos. The embedding space further enables fine-grained retrieval and anomaly detection. Nearest-neighbor search surfaces frames corresponding to specific actions, while deviations from expected trajectories indicate abnormal behavior, ensuring correct procedural order.

Figure A1 illustrates these capabilities: retrieving filled-container frames in water-filling (Row 1), distinguishing pre- vs. post-assembly in tent assembly (Row 2), identifying hard disk insertion in PC assembly (Row 3), and detecting chopping actions across different vegetables (Row 4).

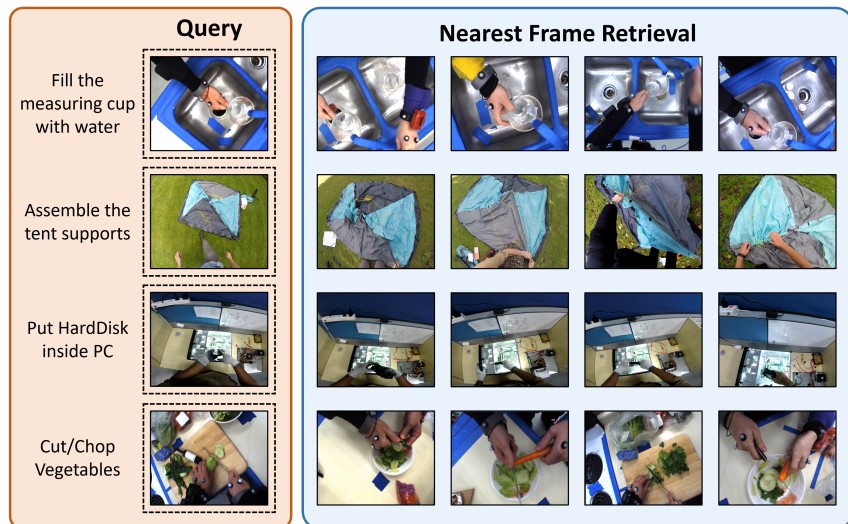

Figure A1: Nearest-neighbor retrieval in the embedding space enables precise frame-level alignment across tasks.

## A.8   ADDITIONAL ABLATION STUDIES

### A.8.1   KEY-STEP LOCALIZATION AND ORDERING USING GRAPHCUT SEGMENTATION

After obtaining frame embeddings through our R-FPGWOT alignment framework, we localize key steps and infer their temporal order to capture procedural structure. Following prior works, we model key-step localization as a multi-label graph cut segmentation problem (Greig et al. (1989)), where the node set includes $K$ terminal nodes representing key steps and non-terminal nodes corresponding to frame embeddings.

Formally, given frame embeddings $\{z_i\}_{i=1}^{T}$ and $K$ key-step prototypes $\{c_k\}_{k=1}^{K}$, we construct a graph $G = (\mathcal{V}, \mathcal{E})$ with $\mathcal{V} = \{1, \ldots, T\}$ and label set $\mathcal{L} = \{1, \ldots, K\}$. The multi-label segmentation $y \in \mathcal{L}^T$ is obtained by minimizing a standard Potts-model graph cut energy:

$$E(\boldsymbol{y}) \;=\; \underbrace{\sum_{i=1}^{T} D_i(y_i)}_{\text{T-links: data term}} \;+\; \beta \underbrace{\sum_{(i,j)\in\mathcal{E}_t} w_{ij}\,\mathbb{I}[y_i \neq y_j]}_{\text{N-links: temporal smoothness}}. \tag{A17}$$

Here, the T-links implement the data term

$$D_i(k) \;=\; \|\boldsymbol{z}_i - \boldsymbol{c}_k\|_2^2, \tag{A18}$$

which encourages frame $i$ to attach to the key-step prototype with the most similar embedding, enforcing structural consistency between clusters and the learned embedding geometry.

The N-links connect temporally adjacent frames $(i,j) \in \mathcal{E}_t$ (typically $j = i + 1$) with weights

$$w_{ij} \;=\; \exp\!\Big(-\frac{\|\boldsymbol{z}_i - \boldsymbol{z}_j\|_2^2}{2\sigma^2}\Big), \tag{A19}$$

so that label changes between nearby frames with similar embeddings incur a higher penalty. This term encourages contiguous, temporally smooth segments while still allowing boundaries where the embeddings change significantly.  The resulting submodular Potts energy $E(\boldsymbol{y})$ is approximately minimized via $\alpha$-Expansion (Boykov et al. (2002)), yielding piecewise-constant key-step segments that are consistent with both the embedding structure and the temporal ordering. To determine the sequential order, we normalize frame timestamps within each video and compute the mean normalized time of frames in each cluster, following Chowdhury et al. (2024). Clusters are then sorted in ascending order of their average time, yielding the predicted key-step sequence for that video.  Finally, across all videos of the same task, we aggregate the discovered orders and rank

them by frequency of occurrence, outputting the most consistent order as the canonical procedural sequence. This pipeline not only identifies salient steps but also resolves their temporal ordering in a robust, data-driven manner.

---

**Algorithm 2** Temporal Ordering of Key Steps

---

**Require:** $R$: predicted key-step assignment for each frame, $k$: number of key steps
**Ensure:** indices: sequential order of tasks
1: $M \leftarrow \text{len}(R)$                                                                                     ▷ Number of frames
2: $T \leftarrow \frac{\{1,2,...,M\}}{M}$                                                                       ▷ Normalized timestamps
3: Initialize $cluster\_time \leftarrow \mathbf{0}_k$
4: **for** $i = 1$ to $k$ **do**
5:     $cluster\_time[i] \leftarrow \text{mean}(T[R == i])$
6: $\_, indices \leftarrow \text{sort}(cluster\_time)$
7: **return** $indices$

---

**Example.**
**Sample Input (R):** [6, 2, 1, 3, 5, 1, 1, 0, 0, 6, 4, 4, 6, 1, 2, 3, 0, 4, 0, 4, 5, 5, 3, 1, 3, 2, 0, 4, 3, 6, 0, 1, 2, 4, 2, 3, 5, 4, 6, 2, 5, 1, 2, 4, 3, 2, 2, 3, 4, 1]
**Sample Output (indices):** [6, 1, 0, 5, 3, 2, 4]

---

### A.8.2 CHOICE OF KEY-STEP $K$

We performed an ablation study to examine the effect of the hyperparameter $K$ on the alignment results. When $K$ was set to small values, the model tended to under-segment the sequence, merging distinct task boundaries and failing to capture fine-grained transitions. In contrast, larger values of $K$ (e.g., 10 or 15) caused over-segmentation, breaking continuous actions into many short intervals as shown in Fig. A2. This excessive fragmentation introduced temporal jitter and decreased the interpretability of the resulting timelines. Selecting $K = 7$ provided the most favorable trade-off: it preserved the major task boundaries while avoiding spurious splits. Empirically, this choice yielded timelines that were both faithful to the ground truth and more robust for downstream analysis.

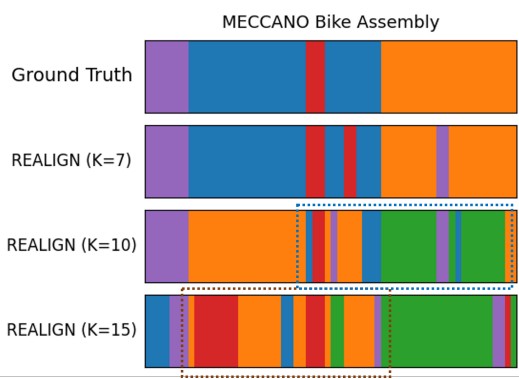

Figure A2: Ablation study on the choice of $K$. With $K = 7$, the model achieves the best balance between capturing essential task boundaries and avoiding over-segmentation. Increasing $K$ leads to more fragmented and jittery segmentations.

### A.8.3 PRIOR DISTRIBUTIONS AND SENSITIVITY TO THE LAPLACE SCALE PARAMETER

**Laplace vs. Gaussian vs. Uniform priors.** In R-FPGWOT, the temporal and optimality structure matrices are instantiated via a parametric prior distribution over pairwise distances, which we implement as a symmetric kernel. We compare three choices of prior: Uniform, Gaussian, and Laplace as shown in Fig. A3. All three induce time-local structure, but with different trade-offs between concentration around the mode and tolerance to deviations. The Uniform prior (Eq. A20) assigns equal mass within a fixed window and zero outside, enforcing a hard locality constraint while failing to differentiate more and less plausible alignments within that window. The Gaussian prior (Eq. A21) yields a strongly peaked kernel that decays rapidly in the tails, encouraging strictly local, near-diagonal alignments but heavily penalizing moderate temporal shifts or non-monotonic correspondences. By contrast, the Laplace prior (Eq. A22) combines a sharp mode with heavier tails, preserving a strong diagonal bias while still assigning non-negligible probability to moderately misaligned frames. This makes it better suited to realistic egocentric videos, where small local jitter and occasional non-monotonic jumps are common.

$$\boldsymbol{Q}(i,j) = \mathcal{N}(x; \mu, \sigma^2) = \frac{1}{\sqrt{2\pi\sigma^2}} \exp\left(-\frac{(x-\mu)^2}{2\sigma^2}\right) \tag{A20}$$

$$\boldsymbol{Q}(i,j) = f(x; a, b) = \begin{cases} \frac{1}{b-a} & \text{if } a \le x \le b, \\ 0 & \text{otherwise.} \end{cases} \tag{A21}$$

$$\boldsymbol{Q}(i,j) = f(x; \mu, b) = \begin{cases} \frac{1}{2b} \exp\left(-\frac{|x-\mu|}{b}\right) & \text{if } -\infty < x < \infty, \\ 0 & \text{otherwise.} \end{cases} \tag{A22}$$

Table A6 reports F1/IoU on all Ego-ProceL datasets for the three priors. The Laplace prior (our default R-FPGWOT configuration) achieves the best or tied-best performance across datasets, consistently outperforming both Uniform and Gaussian variants. These results, together with the convergence analysis in the main text, support our choice of Laplace as the default prior: it satisfies the PSD requirements of Option A (like the Gaussian kernel) while providing a more favorable balance between diagonal sharpness and robustness to temporal variability.

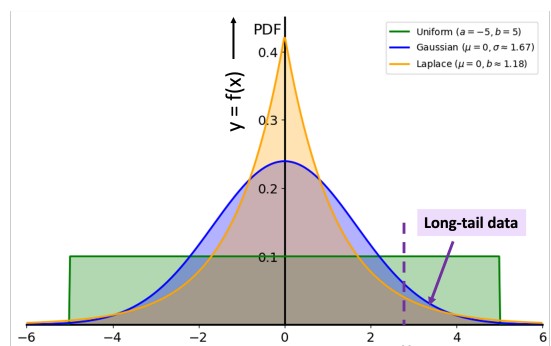

Figure A3: Importance of choosing Laplace distribution as a prior over Gaussian and Uniform distribution.

Table A6: Ablation on the choice of prior distribution (Uniform, Gaussian, Laplace) for the temporal and optimality kernels in R-FPGWOT on EgoProceL. We report F1/IoU (%) for each dataset.

| | EgoProceL | | | | | | | |
| | CMU-MMAC | | EGTEA-GAZE+ | | MECCANO | | EPIC-Tents | |
| | F1 | IoU | F1 | IoU | F1 | IoU | F1 | IoU |
|---|---|---|---|---|---|---|---|---|
| Uniform | 53.0 | 37.0 | 57.5 | 42.5 | 53.0 | 36.0 | 33.0 | 18.5 |
| Gaussian | 57.0 | 41.0 | 61.5 | 46.5 | 57.0 | 40.0 | 37.0 | 22.5 |
| Laplace | **59.7** | **43.7** | **64.2** | **49.3** | **59.6** | **42.7** | **39.8** | **25.0** |

**Sensitivity to the Laplace scale parameter** $b$. Since the Laplace prior is parameterized by a scale $b$, which controls the decay rate around the mode, we further analyze the robustness of R-FPGWOT to this hyperparameter. Intuitively, very small $b$ produces an overly peaked kernel that restricts alignments to a narrow local neighborhood, while very large $b$ flattens the kernel and weakens the temporal guidance, approaching a weakly informative prior. We vary $b \in \{1, 1.5, 2, 2.5, 3, 3.5\}$ and report F1/IoU on CMU-MMAC, EGTEA-GAZE+, and MECCANO in Table A7. As seen in Fig. A4 performance consistently peaks at $b = 2$ across CMU-MMAC and EGTEA-GAZE+ datasets and $b = 3$ across MECCANO, but the changes in F1/IoU across the tested range remain relatively modest, indicating that the method is not overly sensitive to the precise choice of $b$. $b = 2$ *or* 3 provides a good trade-off between temporal concentration around the alignment diagonal and tolerance to realistic temporal jitter and non-monotonicity.

### A.8.4 SEQUENCE ALIGNMENT ROBUSTNESS

To further assess the robustness of our approach, we evaluate the alignment of pairs of sequences that exhibit temporal variations. As shown in Fig. A5, our framework successfully aligns corresponding action frames even when their execution speeds differ across videos. The correct matches demonstrate that the model consistently identifies shared key actions, while redundant or stretched portions of the sequence are effectively handled. This result affirms the reliability of our model in maintaining coherent procedural alignment across temporally diverse sequences.

Table A7: Ablation on the Laplace scale parameter $b$ for the temporal and optimality priors. We report F1/IoU (%) on CMU-MMAC, EGTEA-GAZE+, and MECCANO.

| $b_{\text{Laplace}}$ | CMU-MMAC | | EGTEA-GAZE+ | | MECCANO | |
|---|---|---|---|---|---|---|
| | F1 | IoU | F1 | IoU | F1 | IoU |
| 1.0 | 57.64 | 42.18 | 63.04 | 47.95 | 57.71 | 40.94 |
| 1.5 | 58.23 | 42.98 | 63.23 | 48.57 | 57.96 | 41.36 |
| 2.0 | **59.73** | **43.68** | **64.23** | **49.26** | 58.52 | 41.57 |
| 2.5 | 58.10 | 42.72 | 62.10 | 48.31 | 58.61 | 42.18 |
| 3.0 | 57.98 | 42.12 | 61.98 | 47.75 | **59.61** | **42.67** |
| 3.5 | 57.83 | 41.95 | 61.84 | 47.23 | 58.28 | 41.71 |

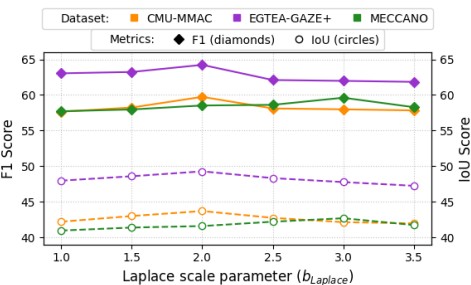

Figure A4: Sensitivity of R-FPGWOT to the Laplace scale parameter $b$.

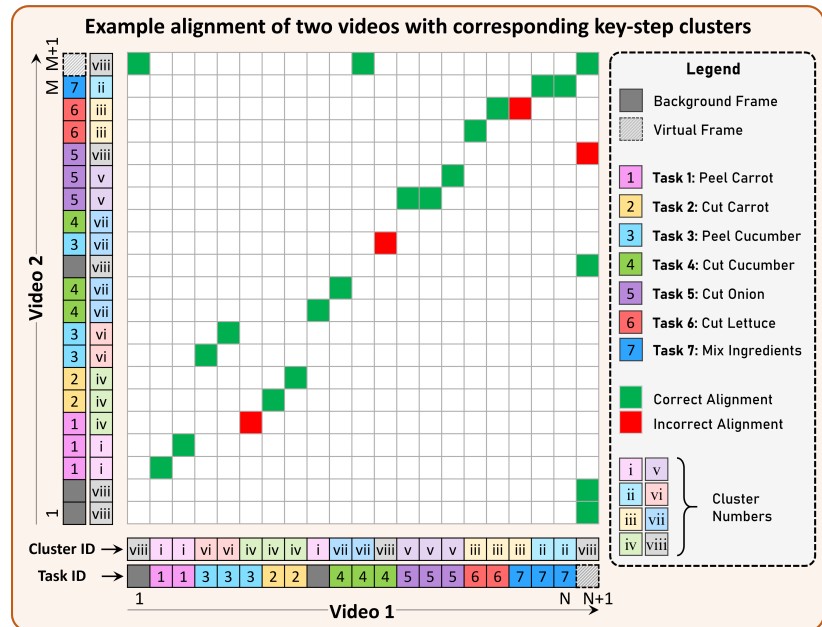

Figure A5: Illustration of sequence alignment of two 'salad making' videos with different temporal dynamics using our framework "REALIGN". Despite variations in execution speed, corresponding action frames are matched accurately, thereby managing redundancy and robustness of our method.

### A.8.5 SENSITIVITY ANALYSES OF OTHER HYPERPARAMETERS

We further investigate the sensitivity of *REALIGN* to several key hyperparameters. Fig. A6 reports F1/IoU on the three EgoProceL datasets (CMU-MMAC, EGTEA-GAZE+, MECCANO) as we vary the Gromov–Wasserstein weight $\alpha$, the marginal constraint penalty $\tau$, the regulariser prior weight $\lambda_1$ and $\lambda_2$, the contrastive margin $\lambda_3$, the contrastive regularization weight $c_2$, the inter-sequence contrastive weight $c_3$, the entropy regularization weight $\epsilon$, the graph-cut smoothness weight $\beta$, and the number of key-steps $K$. Each subplot shows how F1 (solid curves) and IoU (dashed curves) evolve when adjusting a single hyperparameter while keeping the others fixed.

Overall, R-FPGWOT exhibits stable behaviour across the explored ranges. The performance curves are typically smooth and unimodal, with peaks around $\alpha \approx 0.5$ (balancing KOT and GWOT), $\tau \approx 0.8$, $\lambda_1 \approx 1.0$, $\lambda_2 \approx 0.1$, $\lambda_3 \approx 2.0$, $c_2 \approx 0.5$, $c_3 \approx 10^{-4}$, $\epsilon \approx 0.07$, and $\beta \approx 0.2$. For the number of key-steps, performance is best around $K = 7$ and degrades only gradually for larger $K$. Importantly, even when each hyperparameter is perturbed substantially away from its default setting, the change in F1/IoU remains moderate, indicating that R-FPGWOT is robust to the precise choice of these hyperparameters. Consequently, we fix the above values as our default configuration in all reported experiments.

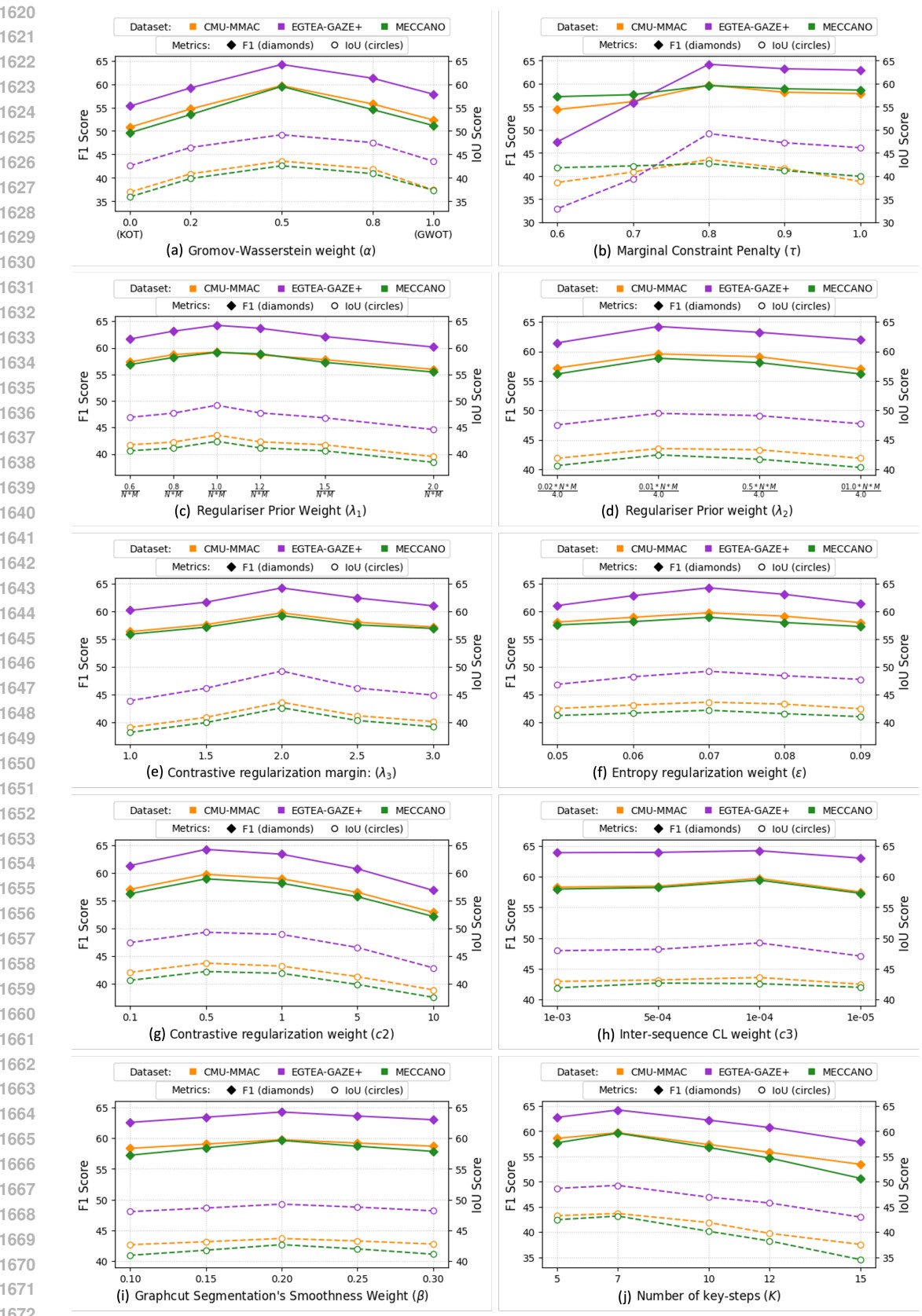

Figure A6: Sensitivity analysis of the various hyperparameters used in *REALIGN*.

