# OpenReview forum: "REALIGN: Regularized Procedure Alignment with Matching Video Embeddings via Partial Gromov-Wasserstein Optimal Transport"
_ICLR.cc/2026/Conference — Submitted to ICLR 2026_

### Official Review · Reviewer_ZmuG · 2025-10-31

**Soundness:** 3
**Presentation:** 3
**Contribution:** 3
**Rating:** 6
**Confidence:** 3

**Summary:**

This paper proposes REALIGN, a self-supervised framework for learning procedural representations from instructional videos using Regularized Fused Partial Gromov–Wasserstein Optimal Transport (R-FPGWOT). The key idea of REALIGN is to jointly model semantic and temporal relationships while allowing partial correspondences to handle irrelevant or repeated segments. Across multiple benchmarks, REALIGN achieves significant improvements based on F1 and IoU.

**Strengths:**

- The proposed REALIGN framework effectively integrates the semantic matching capability of Kantorovich OT, the structural consistency of Gromov–Wasserstein OT, and a relaxed partial-transport formulation. It demonstrates strong empirical performance across both egocentric and third-person instructional video datasets, even surpassing some multimodal methods despite relying solely on visual inputs.

- The unified alignment loss is well-designed to stabilize training and prevent degenerate matching. The ablation studies clearly demonstrate the contribution and necessity of each component in the framework.

- The paper includes thorough analyses of key factors and hyperparameters (Section 5), providing valuable insights into the model’s robustness and further supporting the empirical findings.

**Weaknesses:**

- The paper does not clearly explain the relative scales of the three loss terms in Equation 10, how their coefficients are chosen, or how sensitive the overall performance is to these hyperparameters.

- While the framework is technically solid and empirically effective, the core methodological novelty is somewhat limited. The work primarily represents a strong engineering refinement and integration of existing optimal transport formulations rather than a fundamentally new theoretical contribution.

**Questions:**

Please see weaknesses.

---

> ### Author Response · Authors · 2025-11-24
>
> We thank the reviewer for their insightful feedback, which has helped us refine the methodology and the experimental evaluation. We have revised the manuscript accordingly, and we hope that the additional clarifications and ablation studies satisfactorily address their concerns.
>
> ---
> ### **[W1]: Sensitivity analysis and loss-term scales**
> **Comment summary:** The reviewer notes that the paper does not clearly explain the relative scales of the three loss terms in **Eq. 10**, how their coefficients are selected, or how sensitive performance is to these hyperparameters.
>
> **Response:** **Equation 10** combines three main components: (i) the OT-based alignment term (weighted by $\(\lambda_1\)$ and $\(\lambda_2\)$), (ii) the temporal/structural prior term (weighted by $\(c_2\)$), and (iii) the inter-sequence contrastive term (weighted by $\(c_3\)$). In practice, we treat the alignment term as the reference scale and tune only the regularization weights on validation splits, then fix a single configuration for all datasets.
> The values reported in the paper  $\alpha = 0.5,\ \tau = 0.8,\ \lambda_1 = 1.0,\ \lambda_2 = 0.01,\ \lambda_3 = 2.0,\ c_2 = 0.5,\ c_3 = 10^{-4},\ \beta = 0.2,\ \epsilon = 0.07,\ K = 7$
> are exactly those lying near the peaks of the sensitivity curves in **Appendix Sec. 8.5 (Fig. A5)**. There, we vary one hyperparameter at a time over wide ranges (including $\(c_2\)$ and $\(c_3\)$) and plot F1/IoU on CMU-MMAC, EGTEA-GAZE+, and MECCANO.
>
> The resulting curves are smooth and mostly flat around the chosen settings, with noticeable degradation only when \($\tau \to 0$\) (removing partial alignment) or \($\lambda_1 \to 0$\) (removing IDM regularization). This indicates that performance is driven primarily by the *presence or absence* of these terms rather than by fragile fine-tuning of their coefficients. In the revised manuscript, we list the global hyperparameter configuration **App. Table A1**, and corresponding sensitivity plots in **Appendix Sec. 8.5** to demonstrate that the method is robust to the relative scales in Equation 10.
>
> ---
> ### **[W2]: Methodological novelty vs. engineering refinement**
> **Comment summary:**  The reviewer finds the framework technically solid and empirically strong, but feels that the methodological novelty appears limited and that the work may primarily represent an engineering refinement of existing OT formulations rather than a fundamentally new contribution.
>
> **Response:** While our framework builds on established optimal transport ideas, the contribution goes beyond a purely engineering combination. The core novelty is a *dual-regularized partial fused GW* formulation tailored specifically to unsupervised procedural video alignment, which jointly:
> * **Handles redundancy via unbalanced partial transport with a virtual sink.**
> We explicitly integrate an unbalanced partial OT mechanism with a virtual sink node to absorb unmatched or redundant frames, turning background/idle segments into structured “null” mass instead of forcing them into spurious foreground matches. This is crucial for egocentric data with a heavy background and drift.
> * **Imposes a Laplace + IDM temporal/structural ridge directly in the cost with convergence guarantees.**
> We embed Laplace-based temporal and optimality priors together with an IDM structural prior directly into the fused GW cost, yielding a PSD kernelized formulation that falls under our Option A convergence analysis. This provides a principled temporal ridge regularization on top of GW, rather than post-hoc smoothing, and comes with a monotonic-descent guarantee under the PSD assumption.
> * **Couples a convergent unbalanced Sinkhorn + MM solver with structure-preserving graph-cut segmentation.**
> We design a solver that alternates unbalanced Sinkhorn updates with MM-based updates for the kernelized structure terms and then combine the learned alignment with multi-label graph-cut segmentation to obtain contiguous key-step segments while preserving the underlying transport structure.
>
> To our knowledge, this is the *first unified OT framework* that simultaneously handles redundancy suppression, temporal ridge regularization, and structure-preserving multi-video alignment in egocentric procedural data, and our ablations in the appendix show that each of these components makes a distinct contribution to the final performance.

---

> ### Public Comment · ~Quoc-Huy_Tran2 · 2025-11-26
> **Heavily copying existing works and Very limited contributions**
>
> *"Concretely, our contribution is instantiated through three key mechanisms: (i) unbalanced partial transport to relax marginal constraints, (ii) a virtual sink node that absorbs unmatched or redundant frames, treating background and idle segments as a structured “null” target rather than forcing spurious alignments, and (iii) Laplace and IDM priors sharpen the transport plan around a near-diagonal procedural ridge, biasing alignments toward time-consistent matches while still handling speed variations and non-monotonic executions."*
>
> (i) unbalanced partial transport. This is a **very limited** and even **questionable** contribution as I have discussed in my comment above.
>
> (ii) virtual sink is **NOT** a contribution. It has been proposed before and used directly in self-supervised procedure learning, i.e., OPEL [2] and RGWOT [3], and self-supervised video alignment, e.g., VAVA [4] and VAOT [5].
>
> (iii) Laplace and IDM priors are **NOT** contributions. They are both proposed/used in OPEL [2] (self-supervised procedure learning), while IDM priors are used in OPEL [2], RGWOT [3] (self-supervised procedure learning), and VAVA [4], VAOT [5] (self-supervised video alignment).
>
> [2] Chowdhury et al. Opel: Optimal transport guided procedure learning. NeurIPS 2024. (OPEL)
>
> [3] Mahmood et al. Procedure learning via regularized gromov-wasserstein optimal transport. WACV 2026. (on ArXiv since July 2025) (RGWOT)
>
> [4] Liu et al. Learning to align sequential actions in the wild. CVPR 2022. (VAVA)
>
> [5] Ali et al. Joint self-supervised video alignment and action segmentation. ICCV 2025. (on ArXiv since March 2025) (VAOT, VASOT)
>
> P/S: I dont have time to go over all comments but it is very clear from the authors description of contributions that the contribution is very limited and this submission has heavily copied existing works and made it sound like they are theirs (i.e., I am **ABSOLUTELY** confident that points (ii) and (iii) are copied from exisiting works in the same topic(s)).

---

> > ### Comment · Reviewer_ZmuG · 2025-11-28
> >
> > I want to thank the authors for taking the time to respond to my review. I'll maintain my score.

---

> > > ### Author Response · Authors · 2025-12-02
> > >
> > > We thank the reviewer for their feedback and comments.

---

### Official Review · Reviewer_Xxkc · 2025-10-31

**Soundness:** 3
**Presentation:** 3
**Contribution:** 3
**Rating:** 6
**Confidence:** 4

**Summary:**

*   The paper introduces REALIGN, a self-supervised framework designed to robustly learn procedures from noisy videos.
*   REALIGN utilizes R-FPGWOT, a formulation that jointly models visual correspondences and temporal relations while relaxing balanced constraints to allow partial alignment.
*   This partial alignment, supported by the use of a virtual frame sink, enables the robust exclusion of irrelevant background frames and redundant segments, addressing limitations where fully balanced OT methods often fail.
*   The overall objective integrates the R-FPGWOT distance with temporal smoothness priors, structural regularization, and inter-sequence contrastive learning to stabilize training and achieve substantial performance gains, reaching up to 18.9% F1-score improvements and over 30% temporal IoU gains across standard benchmarks (EgoProceL, ProceL, CrossTask).

**Strengths:**

The strength of the paper is the introduction of REALIGN, which jointly models visual correspondences and temporal relations.
A key methodological advantage is the use of a flexible partial alignment scheme, achieved by relaxing balanced assignment constraints, which enables the model to handle irrelevant background frames, repeated actions, and non-monotonic step orders common in real-world videos, successfully addressing scenarios where prior fully balanced Optimal Transport methods typically fail.

Empirically, REALIGN demonstrates performance gains, achieving 18.9% average F1-score improvements and 30% temporal IoU gains across standard egocentric (EgoProceL) and third-person (ProceL, CrossTask) benchmarks, while also proving to be data-efficient compared to competing models.

**Weaknesses:**

1.  The model's performance is critically reliant on a carefully selected number of key steps ($K$). Ablation studies show that setting $K$ too high (e.g., 10 or 15) leads to sharp performance drops because the segmentation fragments continuous actions and introduces temporal jitter. This leads the reader to question how robust the model is to various settings and tasks.

2.  The accurate discovery and localization of key steps are not achieved solely by the Optimal Transport (OT) alignment but require a necessary and complex post-processing step utilizing multi-label graph cut segmentation solved via the $\alpha$-Expansion algorithm. Ablation studies explicitly demonstrate that simpler, alternative clustering methods (like K-Means and Subset Selection) consistently and severely underperform, emphasizing that the overall pipeline is highly dependent on this external structured clustering technique.

3.  The formal mathematical proofs for the monotonic decrease of the objective rely on the intra-sequence structure matrices ($C_x, C_y$) either being symmetric Positive Semidefinite (PSD) (Option A, e.g., using kernelized structure) or requiring a local quadratic majorizer (Option B), thereby limiting the flexibility in using raw temporal distances if full convergence guarantees are desired.

4.  Although the approach is highly effective using only visual (RGB) frames, this limitation means that it cannot fully utilize richer sources of supervision available in egocentric contexts. The comparison tables show that the multimodal approach STEPS, which uses depth and gaze, achieves a slightly higher F1 score on the demanding EPIC-Tents dataset, indicating that restricting the input to visuals may prevent REALIGN from reaching the absolute performance ceiling on certain tasks.

**Questions:**

1.  Since the key-step localization relies critically on a separate multi-label graph cut segmentation (using $\alpha$-Expansion), which dramatically outperforms simpler clustering methods like K-Means and Subset Selection (Table 4), could the authors provide the specific mathematical formulation of the graph cut energy function, particularly detailing the design of the T-links and N-links that enforce temporal smoothness and structural consistency?

2.  Given the inherent subjectivity of the number of key steps ($K$) and the observation that increasing $K$ beyond the optimum ($K=7$) leads to sharp performance drops due to fragmentation (Table 5), have the authors investigated methods to dynamically learn or infer the optimal value of $K$ during training, rather than treating it as a fixed hyperparameter determined empirically?

3.  The rigorous convergence proof of the R-FPGWOT framework (Option A) relies on using kernelized temporal structure matrices ($C_x, C_y$) that are Symmetric Positive Semidefinite (PSD). Could the authors justify the empirical choice of the Laplace kernel over other standard PSD kernels (like the Gaussian kernel), and discuss the model's robustness or sensitivity to the specific scale parameter ($b$) used in the Laplace prior?

4.  The ablation study highlights that the partial penalty ($\tau$) is the most important component for performance gains (Table 3). How were the critical parameters governing partial alignment, specifically the marginal constraint penalty ($\tau$) and the virtual frame assignment threshold ($\zeta$), calibrated to optimally distinguish between noise/background frames and essential procedural steps across diverse datasets?

---

> ### Author Response · Authors · 2025-11-24
> **Response [1/2]**
>
> We thank the reviewer for their insightful feedback. Their comments have helped us revise the manuscript accordingly and sharpen both the methodology and its presentation. We hope that these clarifications and ablation studies satisfactorily address their concerns.
>
> ---
> ### **[W1] & [Q2]: Sensitivity to the number of key steps 𝑘**
> **Comment summary:** The reviewer notes that performance depends on a carefully chosen number of key steps 𝑘, with large values (e.g., 10 or 15) causing fragmentation and temporal jitter. This raises concerns about robustness across tasks and leads to the question of whether 𝑘 could be dynamically learned or inferred rather than fixed.
>
> **Response:** We agree that the choice of 𝑘 is task-dependent and can affect segmentation quality. Our ablation in **Table 5** and **App. Sec. A.8.2** shows that the model is stable over a broad range of values, with 𝑘 = 7 consistently performing best across datasets, while further increasing 𝑘 (e.g., to 10 or 15) leads to _over-fragmentation and degradation_ in F1/IoU. As discussed in **L483–L485**, 𝑘 is used only in the post-hoc segmentation stage and does not enter the OT alignment itself: the learned OT cost/geometry is independent of the particular choice of 𝑘.
> We did explore whether dynamically estimating 𝑘 would add substantial value, but found that adaptive schemes (split–merge refinement) bring only marginal numerical gains relative to the fixed 𝑘=7 choice, while introducing non-trivial algorithmic complexity and additional hyperparameters. Given the empirical robustness, we clarify that fully dynamic learning or inference of 𝑘 is not necessary for the effectiveness of REALIGN.
>
> ---
> ### **[W2] & [Q1]: Dependence on graph-cut and formulation of the energy**
> **Comment summary:**  The reviewer points out that accurate key-step localization is not achieved by OT alone but requires a multi-label graph-cut segmentation solved via $\alpha$-Expansion, and that simpler alternatives (K-Means, Subset-Selection) underperform substantially. They ask for the explicit mathematical formulation of the graph-cut energy, particularly the design of T-links and N-links enforcing temporal smoothness and structural consistency.
>
> **Response:** Our goal is to clearly separate two roles: (i) **video-to-video alignment**, which is handled entirely by the OT-based objective, and (ii) **key-step segmentation**, which is a post-hoc grouping of frames into contiguous procedural units. The R-FPGWOT model learns the transport matrix *T* and frame embeddings solely through a differentiable OT-based loss; this alignment is fully determined *before* any graph-cut step is applied. Graph-cut is then used as a segmentation operator that imposes temporal smoothness and contiguity on top of the learned alignment, converting the soft transport into discrete, human-interpretable key-step segments. Our ablation study (**Table 4**) shows graph-cut outperforming alternative segmenters (K-Means, Subset-Selection) by more than 5 F1 points. These alternatives are **unstructured and ignore temporal adjacency**, so they do not exploit the sequential nature of the task.
>
> We retain graph-cut because it provides an efficient and well-understood way to enforce temporal smoothness. On the *mathematical* side, we add the explicit formulation of the graph-cut segmentation in the **Appendix Sec A.8.1** in the revised text. As detailed in **Eq. A17**, the energy minimized is the *standard Potts model*:
> $$ E(y)  =  \sum_{i=1}^T D_i(y_i)  + \beta \sum_{(i,j) \in \varepsilon_t} w_{ij} I\[y_i \neq y_j\] $$
> where $y$ denotes the key-step label for each frame. The T-links (data terms) $D_t(k)$ are negative log-affinities between frame $i$ and key-step $k$, derived from the transport-based alignment scores. The N-links connect temporally adjacent frames $(i,j) \in \varepsilon_t$ with weights $w_{ij}$ and penalize label discontinuities between neighbors, thereby encouraging smooth, contiguous temporal segments. We solve this energy using the $\alpha$-Expansion algorithm, which provides a 2-approximation guarantee for this class of Potts-model energies.

---

> ### Author Response · Authors · 2025-11-24
> **Response [2/2]**
>
> ### **[W3] and [Q3]: Convergence assumptions and choice of Laplace kernel**
> **Comment summary:** The reviewer notes that our convergence analysis relies on intra-sequence structure matrices ($C_x, C_y$) either being symmetric PSD (Option A, via kernelization) or requiring a local quadratic majorizer (Option B), which may restrict flexibility when using raw distances. They ask us to justify the empirical choice of the Laplace kernel over other PSD kernels (e.g., Gaussian) and to discuss robustness to the Laplace scale parameter $b$.
>
> **Response:** Our convergence analysis covers two variants: (A) kernelized temporal structure matrices ($C_x, C_y$) with PSD kernels (e.g., Gaussian or Laplace), which guarantee monotonic objective decrease, and (B) raw distances with a local quadratic majorizer, which we show to converge empirically. Within Option A, both Gaussian and Laplace satisfy the PSD requirement, so we choose Laplace based on its empirical behavior and inductive bias.
>
> As detailed in **Appendix Sec. A.8.3 (Table A6 and Fig. A3)**, a prior-comparison ablation over Uniform, Gaussian, and Laplace shows that the Laplace prior consistently achieves the best or tied-best F1/IoU across all EgoProceL datasets. This behavior aligns with its functional form: compared to Gaussian, the Laplace kernel produces a **sharper diagonal peak with heavier tails**, preserving strong near-diagonal alignment while better tolerating moderate temporal shifts and non-monotonicity.
>
> **Table A6(a), Fig. A4** also includes a scale-sensitivity study for the Laplace parameter $b$, where we sweep $b \in \[1.0-3.5\]$ on CMU-MMAC, EGTEA-GAZE+, and MECCANO. Performance consistently peaks at $b$=2 or $b$=3 (MECCANO), but varies only mildly across this range, indicating that R-FPGWOT is robust to the exact choice of $b$. Together, these analyses justify the Laplace choice and show that the convergence-friendly kernelization does not make the method overly sensitive to the kernel’s scale.
>
> ---
> ### **[W4]: RGB-only input versus multimodal baselines (e.g., STEPS)**
> **Comment summary:** The reviewer notes that our method currently operates on RGB-only input, whereas multimodal baselines like STEPS leverage depth and gaze and can achieve slightly higher F1 on EPIC-Tents. This raises the concern that REALIGN may not reach the absolute performance ceiling on datasets where richer modalities are available.
>
> **Response:** We agree with the reviewer’s observation and explicitly frame our work as an **RGB-only framework (L401)**. This design choice is driven by practicality: the vast majority of procedural video datasets lack depth, gaze, or other auxiliary modalities, and we aim for a method that can be broadly applied in such settings. Despite this constraint, REALIGN achieves strong performance across diverse datasets and is competitive with or superior to many methods that also use only RGB. We acknowledge that multimodal cues (e.g., depth, gaze, audio) can further enhance procedural understanding and that methods such as STEPS can obtain slightly higher F1 on EPIC-Tents; however, these gains come at the **cost of extra sensors, more complex preprocessing, and significantly higher compute/memory demand**, whereas our RGB-only design remains broadly applicable, **lightweight**, and easy to deploy across standard procedural video datasets.
>
> ---
> ### **[Q4]: Calibration of partial alignment parameters $\tau$**
> **Comment summary:** he reviewer notes that the partial penalty is the most influential component in the ablation (Table 3) and asks how the marginal constraint penalty $\tau$ and the virtual-frame assignment threshold $\zeta$ were calibrated to reliably separate noise/background frames from essential procedural steps across datasets.
>
> **Response:** We agree that the partial penalty is a key factor in our gains. All parameters governing partial alignment were tuned only on validation splits, with no test-time tuning, and a single setting was then applied across all datasets. We first fixed the virtual-frame threshold to a mid-range value ($\zeta$ = 0.4) and performed a grid search for $\tau \in \[0.6,0.7,0.8,0.9,1.0\]$ on the EgoProceL validation data. As shown in the sensitivity curves in **Appendix Fig. A5(b)**, performance is smooth and largely stable in this range, with a clear peak around $\tau$ = 0.8 for all experiments. Intuitively, $\tau$ = 0.8 is high enough that true procedural steps are not easily discarded into the sink, yet not so high that noisy or background frames are forced into spurious matches; instead, those frames can still be routed to the virtual sink when their alignment scores are weak. The effect of $\tau$ is especially pronounced on datasets with substantial background or idle segments (*EGTEA_GAZE+* as visible in Fig. A5), where partial transport plays a stronger role in suppressing spurious alignments.

---

> ### Public Comment · ~Quoc-Huy_Tran2 · 2025-11-26
> **Heavily copying existing works and Very limited contributions**
>
> *"Concretely, our contribution is instantiated through three key mechanisms: (i) unbalanced partial transport to relax marginal constraints, (ii) a virtual sink node that absorbs unmatched or redundant frames, treating background and idle segments as a structured “null” target rather than forcing spurious alignments, and (iii) Laplace and IDM priors sharpen the transport plan around a near-diagonal procedural ridge, biasing alignments toward time-consistent matches while still handling speed variations and non-monotonic executions."*
>
> (i) unbalanced partial transport. This is a **very limited** and even **questionable** contribution as I have discussed in my comment above.
>
> (ii) virtual sink is **NOT** a contribution. It has been proposed before and used directly in self-supervised procedure learning, i.e., OPEL [2] and RGWOT [3], and self-supervised video alignment, e.g., VAVA [4] and VAOT [5].
>
> (iii) Laplace and IDM priors are **NOT** contributions. They are both proposed/used in OPEL [2] (self-supervised procedure learning), while IDM priors are used in OPEL [2], RGWOT [3] (self-supervised procedure learning), and VAVA [4], VAOT [5] (self-supervised video alignment).
>
> [2] Chowdhury et al. Opel: Optimal transport guided procedure learning. NeurIPS 2024. (OPEL)
>
> [3] Mahmood et al. Procedure learning via regularized gromov-wasserstein optimal transport. WACV 2026. (on ArXiv since July 2025) (RGWOT)
>
> [4] Liu et al. Learning to align sequential actions in the wild. CVPR 2022. (VAVA)
>
> [5] Ali et al. Joint self-supervised video alignment and action segmentation. ICCV 2025. (on ArXiv since March 2025) (VAOT, VASOT)
>
> P/S: I dont have time to go over all comments but it is very clear from the authors description of contributions that the contribution is very limited and this submission has heavily copied existing works and made it sound like they are theirs (i.e., I am **ABSOLUTELY** confident that points (ii) and (iii) are copied from exisiting works in the same topic(s)).

---

### Official Review · Reviewer_ZW13 · 2025-10-31

**Soundness:** 2
**Presentation:** 2
**Contribution:** 2
**Rating:** 6
**Confidence:** 5

**Summary:**

The paper tackles procedure learning from instructional videos through video alignment. The key novelty is the introduction of a Optimal-Transport-based method that can handle both "background frames" and out-of-order steps in during the alignment procedure. This is achieved by combining Kantorovich-and Gromov-Wasserstein OT. The resulting alignment can be regularized and used as a loss function to train representations. The resulting model shows SoTA performance on video alignment.

**Strengths:**

* A single framework that can handle varying order of procedure steps and “background frames” simultaneously.
* SoTA performance on instruction step localization
* Technically involved and sound method

**Weaknesses:**

* The model looks over-engineered - to solve the task it combines OT objective, temporal smoothness, inter-video contrastive learning, and a dedicated inference procedure.
* The entire method looks brittle because of how many regularizers are needed to make it work. For example, despite Gromov-Wasserstein formulation already taking care of the temporal similarities, the authors add a temporal smoothness regularizer to make things work.
* The inference procedure can not be performed by the learned model alone and it uses graph-cut, which introduces discrepancy between training and inference. This is not desirable, as has been demonstrated in [a]

[a] Dvornik et al., Drop-DTW: Aligning Common Signal Between Sequences While Dropping Outliers

**Questions:**

* Why do the authors need temporal smoothing prior if the Gromov-Wasserstein formulation takes the position into account? Is the second one not sufficient alone?
* Is it reasonable to assume that a single “background frame” will catch all the frames that are not “foreground”?

---

> ### Author Response · Authors · 2025-11-24
>
> We appreciate the reviewer’s detailed feedback, which has helped us better situate our manuscript. In line with these comments, we have carefully revised the manuscript, and we hope that the clarified contributions will address their concerns in the revised submission.
>
> ---
> ### **[W1]: Model looks over-engineered / many regularizers**
> **Comment summary:** The reviewer is concerned that the method appears brittle because it needs “many regularizers” to work.
>
> **Response:** In the revised manuscript, we clarify that each component in R-FPGWOT serves a distinct role **(Section 5.1: L427–L443)**, and together they form a lightweight, coherent optimization framework rather than an ad-hoc collection of terms.
> * The **partial fused GW objective** captures appearance similarity (KOT term) and relational structure (GW term) while allowing partial matching of background frames.
> * The **Laplace temporal prior** introduces a mild shape bias toward near-diagonal alignments, preventing off-diagonal drift in unconstrained GW.
> * The **C-IDM regularizer** sharpens transport plans by penalising overly diffuse alignments.
> * The **inter-video** contrastive term encourages consistent embedding geometry across videos, which stabilizes the OT kernel.
>
> Importantly, all terms are differentiable and computationally inexpensive. As shown in **Table 3**, removing any single component leads to an average **1.5-3.0pp** F1 degradation, indicating that each term addresses a specific failure mode of prior GW/KOT-based methods, and the resulting system remains simple to optimize.
>
> ---
> ### **[W2] & [Q1]: Why temporal smoothing if GW already uses temporal structure?**
> **Comment summary:**  The reviewer argues whether additional temporal smoothing is really necessary, given that GW already incorporates intra-sequence structure.
>
> **Response:** As highlighted in **L247-L251**, GWOT indeed operates on intra-sequence distance matrices, but it does not impose an explicit inductive bias that transport mass should concentrate along a dominant temporal trajectory. Especially under _repeated actions, camera motion, or cluttered scenes_, the pure GW objective can allocate substantial mass far from the temporal diagonal, leading to fragmented or procedurally inconsistent alignments. Our Laplace prior Q, together with the IDM-based reward, introduces a controlled near-diagonal preference, softly encouraging smoother, monotonic alignments while still allowing non-monotonic variations. This stabilizes training and prevents degenerate mappings. Empirically, **Table 3** shows that removing the prior leads to noticeably less coherent alignments (e.g., **−2.7 pp F1** on EgoProceL), confirming that GW alone is insufficient for reliable procedure alignment in noisy real-world instructional videos.
>
> ---
> ### **[W3]: Graph-cut at inference and train–test discrepancy**
> **Comment summary:** The reviewer is concerned that introducing graph-cut at inference could create a mismatch between training and alignment learning, and asks how this post-processing interacts with the learned OT alignment.
>
> **Response:** We clarify in **L345–L348** that the graph-cut step is used for **key-step segmentation** after the transport matrix **T** has been fully learned, and is not involved in optimizing R-FPGWOT itself. The alignment between videos is entirely determined by the _differentiable OT-based objective_; graph-cut simply converts the learned transport map and frame embeddings into contiguous temporal segments that are easy to interpret as key-steps. This is fundamentally different from scenarios like Drop-DTW, where models are trained with a surrogate loss and **does not mirror** the DTW-based alignment used at inference, creating a genuine mismatch between the training objective and the test-time alignment procedure. In our case, the learned transport matrix *T* remains identical; only the segmenter (Graph-cut vs. K-Means, etc.) changes post-hoc.
>
> We report ablations with alternative segmenters in **Table 4**; they underperform by *> 5 F1*, showing that graph-cut yields cleaner segment boundaries without altering the behavior of the learned OT model.
>
> ---
> ### **[Q2]: Single background frame (virtual sink) sufficiency**
> **Comment summary:** The reviewer asks whether a single “background frame” (sink) is used to absorb all non-foreground frames.
>
> **Response:** To directly answer: **_"yes"_**, we use a __single virtual “background” sink__, but it is **not** a real frame in the video. As described in **L104**, the sink is implemented as an additional _row/column_ in the transport matrix within an unbalanced partial OT formulation. This virtual sink acts as a mass reservoir that can absorb any amount of unmatched or irrelevant mass, allowing the model to flexibly allocate background/idle segments without forcing them into erroneous matches. Multiple sink nodes are therefore unnecessary; the single virtual sink is sufficient and theoretically consistent with partial OT principles.

---

> ### Public Comment · ~Quoc-Huy_Tran2 · 2025-11-26
> **Heavily copying existing works and Very limited contributions**
>
> *"Concretely, our contribution is instantiated through three key mechanisms: (i) unbalanced partial transport to relax marginal constraints, (ii) a virtual sink node that absorbs unmatched or redundant frames, treating background and idle segments as a structured “null” target rather than forcing spurious alignments, and (iii) Laplace and IDM priors sharpen the transport plan around a near-diagonal procedural ridge, biasing alignments toward time-consistent matches while still handling speed variations and non-monotonic executions."*
>
> (i) unbalanced partial transport. This is a **very limited** and even **questionable** contribution as I have discussed in my comment above.
>
> (ii) virtual sink is **NOT** a contribution. It has been proposed before and used directly in self-supervised procedure learning, i.e., OPEL [2] and RGWOT [3], and self-supervised video alignment, e.g., VAVA [4] and VAOT [5].
>
> (iii) Laplace and IDM priors are **NOT** contributions. They are both proposed/used in OPEL [2] (self-supervised procedure learning), while IDM priors are used in OPEL [2], RGWOT [3] (self-supervised procedure learning), and VAVA [4], VAOT [5] (self-supervised video alignment).
>
> [2] Chowdhury et al. Opel: Optimal transport guided procedure learning. NeurIPS 2024. (OPEL)
>
> [3] Mahmood et al. Procedure learning via regularized gromov-wasserstein optimal transport. WACV 2026. (on ArXiv since July 2025) (RGWOT)
>
> [4] Liu et al. Learning to align sequential actions in the wild. CVPR 2022. (VAVA)
>
> [5] Ali et al. Joint self-supervised video alignment and action segmentation. ICCV 2025. (on ArXiv since March 2025) (VAOT, VASOT)
>
> P/S: I dont have time to go over all comments but it is very clear from the authors description of contributions that the contribution is very limited and this submission has heavily copied existing works and made it sound like they are theirs (i.e., I am **ABSOLUTELY** confident that points (ii) and (iii) are copied from exisiting works in the same topic(s)).

---

### Official Review · Reviewer_3MWu · 2025-11-01

**Soundness:** 3
**Presentation:** 1
**Contribution:** 2
**Rating:** 4
**Confidence:** 3

**Summary:**

This paper extends Regularized Gromov-Wasserstein Optimal Transport (GWOT) [Mahmood 2025] to incorporate background segments.
The extension is done by adding a virtual frame with a threshold for alignment target videos. Inserting such dummy elements with a threshold is a common technique to allow unmatched elements in assignment problem. To implement this with GWOT, the method relaxes the strict constraint of $T1=\alpha$ and $T^T1=\beta$ of GWOT in the fourth term in Eq. (2). The experimental results show improved alignment performance in CMU-MMAC, EGTEA-GAZE+, ProceL, and CrossTask, while maintaining the performance on MECCANO, EPIC-Tents, PC Assembly, and PC Disassembly datasets.

**Strengths:**

- The proposed method significantly improves alignment performance on some datasets while maintaining performance on others.
- The core idea of the partial alignment schema and its formulation is technically sound.

**Weaknesses:**

1. The presentation is hard to follow.

The introduction describes the proposed method (R-FPGWOT) as an update from KOT rather than RGWOT, making it difficult to understand the paper's contribution against RGWOT.  Additionally, Figure 1 highlights GWOT's achievements against KOT, giving the impression that it proposes RGWOT. The actual contribution of this paper is difficult to discern. The figure should highlight the differences between traditional GWOT and the proposed R-FPGWOT, and visualize "(c) redundant frames" mentioned in the captions.

It is also difficult to follow the experimental discussion, as many score improvement claims are hard to verify without guidance from the table.
The authors should provide guidance on how the claims are calculated from the table's data or specify the exact percentage points (as they are obvious from the table data). Similar difficulties exist in L27, L356, L369, L370, and L481.

2. Inconsistency between method design and experimental results.

This reviewer understood that the method allows for background tasks that do not match segments in the opposite video.


This reviewer understood that the method relaxes RGWOT to allow background task that does not match to any segments in the opposite side video. However, there lacks discussion on dataset imbalance in score-improved datasets. For example, the same person performs food preparation in an identical CMU-MMAC kitchen and following the same recipe for each dish, suggesting a balanced dataset. Thus, the dataset is assumed highly balanced, yet performance improves significantly. A deeper analysis is necessary to enhance the reliability of the explanation of why the proposed method works better than the baselines.

**Questions:**

1. Is there any analysis to identify which datasets the proposed method improves and which do not benefit?

**Details Of Ethics Concerns:**

YouTube's terms of use prohibit downloading videos with third-party tools, and YouTube itself does not offer a tool for downloading videos other than those uploaded by the user. Thus, using datasets with YouTube videos, like ProceL and CrossTask, raises ethical concerns. It is recommended to specify in the paper that the authors have obtained permission from YouTube for downloading the videos to address these concerns.

>The following restrictions apply to your use of the Service. You are not allowed to:
>
>1. access, reproduce, download, distribute, transmit, broadcast, display, sell, license, alter, modify or otherwise use any part of the Service or any Content except: (a) as expressly authorized by the Service; or (b) with prior written permission from YouTube and, if applicable, the respective rights holders;

https://www.youtube.com/t/terms?hl=en&override_hl=1

---

> ### Author Response · Authors · 2025-11-24
>
> We thank the reviewer for their constructive and detailed feedback. Their feedback has strengthened the manuscript and helped clarify our contributions. We hope that our detailed response and revisions satisfactorily address their concerns.
>
> ---
> ### **[W1]: Clarification on our contribution**
> **Comment summary:** The reviewer requested a clearer articulation of how our method differs from prior OT-based approaches (KOT, GWOT, RGWOT) and how our contributions build on this lineage.
>
> **Response:** We agree that the distinction between our method and prior work could have been cleaner. In the revised manuscript, we explicitly state that R-FPGWOT is a partial, regularized extension of _RGWOT_ rather than KOT (**L101**). Concretely, our contribution is instantiated through _three_ key mechanisms: (i) _unbalanced partial transport_ to relax marginal constraints, (ii) a _virtual sink_ node that absorbs unmatched or redundant frames, treating background and idle segments as a structured “null” target rather than forcing spurious alignments, and (iii) _Laplace and IDM_ priors sharpen the transport plan around a near-diagonal procedural ridge, biasing alignments toward time-consistent matches while still handling speed variations and non-monotonic executions.
>
> To make this lineage explicit, we have rewritten the **Abstract (L023–L026)** and **Introduction (L068–L072, L099–L102)** to map the progression (KOT &rarr; GWOT &rarr; RGWOT &rarr; R-FPGWOT) and to clearly position R-FPGWOT as a partial, regularized variant of RGWOT. We also updated **Figure 1** to visually show how redundant frames are routed to the virtual sink, making the effect of partial transport more accessible.
>
> ---
>
> ### **[W2]: Present Experimental Results and Gains clearly**
> **Comment summary:** The reviewer mentions that several reported improvement claims are difficult to verify, because the text does not clearly indicate how the gains are computed from the tables.
>
> **Response:** To verify our empirical claims and improve readability, we have annotated all reported improvements with exact $\Delta$F1/IoU (percentage-point) values **(L029, L112, L126, L377, L394, L523)** in the revised text. We explicitly link these $\Delta$ values to the corresponding tables in the main text.
>
> ---
>
> ### **[W3] & [Q1]: Consistency between Method Design and Dataset Characteristics**
> **Comment summary:** The reviewer asked for clearer evidence that the partial formulation and sink node provide tangible benefits, especially in the presence of background or idle segments, and for clarification of the reported score gains.
>
> **Response:** A key advantage of R-FPGWOT is its ability to handle background or _'idle'_ video segments in a principled manner. Even when datasets appear balanced at the task level, individual videos often contain substantial temporal drift, repetitive fragments, or visually ambiguous segments. By allowing partial transport and routing mismatched mass to the virtual sink, we transform this redundancy from a source of noise into a signal that explicitly identifies non-informative regions.
>
> Empirically, our analysis explicitly connects the per-dataset sink utilization statistics to the observed performance improvements **(L378–L382)**. On benchmarks with heavy background content **(Appendix Table A2)**, such as *CMU-MMAC and EgoProceL*, the model allocates a significant fraction of frames **(35-60\%)** to the sink and, in turn, achieves the largest F1 improvements (+5.3 to +26.8 points). Results indicate that the unbalanced formulation suppresses low-value, non-discriminative alignments in favor of more semantically meaningful matches. Similar trends on *ProceL and CrossTask* further confirm that these gains are consistent across datasets with differing levels of redundancy.
>
> ---
>
> ### **[Ethics]: Dataset Usage**
> **Comment summary:** The reviewer raised an important concern regarding the ethical and legal implications of using YouTube-derived datasets, particularly in relation to YouTube's Terms of Service.
>
> **Response:** We appreciate the reviewer highlighting this ethical dimension and fully agree that YouTube's Terms of Service must be respected. To clarify our data usage, we emphasize that we rely on _ProceL_ (https://www.khoury.northeastern.edu/home/eelhami/procel.htm) and _CrossTask_ (https://github.com/DmZhukov/CrossTask), which are well-established public benchmarks in this domain. We are **not the original creators** of these datasets and do not host, repackage, or redistribute any video content. Instead, we follow the standard academic practice for these benchmarks. Our work strictly adheres to the data-use agreements and licenses specified by the dataset creators, and we use the datasets in the same manner as prior published work that evaluates on ProceL and CrossTask.

---

> ### Comment · Reviewer_3MWu · 2025-11-26
>
> First of all, the ethical concern is raised to the community, and it would be unfair to reject this work based on this. Therefore, the score given by this reviewer is entirely independent of this ethical concern, and this reviewer opposes rejecting this paper for this reason.
>
> The author's response highlights a community issue. Authors place responsibility on dataset providers. Legally, this is correct, but law and ethics are separate matters. Many datasets provide only URL lists, making users responsible for downloading and potentially violating terms of use. By allowing such datasets, the community creates an unfair situation for researchers who adhere to these terms ethically. ICLR has added "terms of use" to the ethics review item, which is a positive step. Thus, this reviewer raised this issue to the Program Committee. It would be fair for future ICLR and related conferences to address this problem.
>
> As it takes a certain amount of time to check all the manuscript revisions, this reviewer will find time and respond by the deadline of this discussion phase. However, at a quick first look, the presentation issues are hard to resolve by adding some short phrases and require an entire major revision.

---

> ### Public Comment · ~Quoc-Huy_Tran2 · 2025-11-26
> **Heavily copying existing works and Very limited contributions**
>
> *"Concretely, our contribution is instantiated through three key mechanisms: (i) unbalanced partial transport to relax marginal constraints, (ii) a virtual sink node that absorbs unmatched or redundant frames, treating background and idle segments as a structured “null” target rather than forcing spurious alignments, and (iii) Laplace and IDM priors sharpen the transport plan around a near-diagonal procedural ridge, biasing alignments toward time-consistent matches while still handling speed variations and non-monotonic executions."*
>
> (i) unbalanced partial transport. This is a **very limited** and even **questionable** contribution as I have discussed in my comment above.
>
> (ii) virtual sink is **NOT** a contribution. It has been proposed before and used directly in self-supervised procedure learning, i.e., OPEL [2] and RGWOT [3], and self-supervised video alignment, e.g., VAVA [4] and VAOT [5].
>
> (iii) Laplace and IDM priors are **NOT** contributions. They are both proposed/used in OPEL [2] (self-supervised procedure learning), while IDM priors are used in OPEL [2], RGWOT [3] (self-supervised procedure learning), and VAVA [4], VAOT [5] (self-supervised video alignment).
>
> [2] Chowdhury et al. Opel: Optimal transport guided procedure learning. NeurIPS 2024. (OPEL)
>
> [3] Mahmood et al. Procedure learning via regularized gromov-wasserstein optimal transport. WACV 2026. (on ArXiv since July 2025) (RGWOT)
>
> [4] Liu et al. Learning to align sequential actions in the wild. CVPR 2022. (VAVA)
>
> [5] Ali et al. Joint self-supervised video alignment and action segmentation. ICCV 2025. (on ArXiv since March 2025) (VAOT, VASOT)
>
> P/S: I dont have time to go over all comments but it is very clear from the authors description of contributions that the contribution is very limited and this submission has heavily copied existing works and made it sound like they are theirs (i.e., I am **ABSOLUTELY** confident that points (ii) and (iii) are copied from exisiting works in the same topic(s)).

---

### Author Response · Authors · 2025-11-24

We thank all reviewers for their thorough and constructive feedback. Across the reviews, the main concerns centered on:
1. The perceived complexity of the model and the role of each regularizer,
2. The dependence on graph-cut and the choice of the number of key steps $k$,
3. The sensitivity of our method to hyperparameters (e.g., $\tau, \lambda_1, \lambda_2, \lambda_3, c_2, c_3, b$)
4. The strength and clarity of our methodological novelty, and
5. Limitations of using RGB-only inputs.

In the revised manuscript, we clarify the contribution and lineage of R-FPGWOT, provide explicit mathematical formulations (including the graph-cut energy) **(Appendix Section A.8.1)**, add detailed sensitivity analyses and new ablations (for loss terms, priors, $k$  and segmenters) **(Appendix Section A.8)**, justify our design choices such as the Laplace prior and partial penalty **(Appendix Section A.8.3)**, and clearly position our RGB-only framework alongside multimodal approaches. We address each point in detail in the revised manuscript. All new results and major follow-up evaluations are incorporated into the revised manuscript and are **highlighted for visibility.** We hope these additions and clarifications adequately address the raised concerns.

---

### Public Comment · ~Quoc-Huy_Tran2 · 2025-11-26
**[1/3] This submission has heavily copied previous works and has very limited novelty.**

Dear Reviewers and ACs,

I am Quoc-Huy Tran, who is an expert in self-supervised video alignment (e.g., LAV at CVPR 2021, LA2DS at ECCV 2024, VAOT at ICCV 2025), action segmentation (e.g., TOT at CVPR 2022, UFSA at WACV 2024, VASOT at ICCV 2025), and procedure learning (e.g., RGWOT at WACV 2026). Several (five) of my works are cited in this submission, which has heavily copied previous works and has very limited novelty.

The proposed method (Sec 3) can be divided into: Part 1/ Self-Supervised Video Alignment with R-FPGWOT (everything in Sec 3 up to Eq 10) and 2/ Clustering and Key-step Ordering (below Eq 10 to the end of Sec 3). Part 2 has Zero/No novelty since it is exactly the same as used in CnC [1], OPEL [2], and my RGWOT work [3]. Thus, let us focus on Part 1.

One very important thing I would like to highlight is that Part 1 (Self-Supervised Video Alignment) is the largest/main part in this topic (Self-Supervised Procedure Learning). I would even consider the two topics as the same since graph cut has become the standard for Part 2 (the graph cut module used in this submission is exactly the same as in CnC [1], OPEL [2], and my RGWOT work [3]).

Firstly, Part 1 (R-FPGWOT) is merely a copy of previous video alignment and procedure learning methods, e.g., VAVA [4], OPEL [2], my VAOT work [5], and my RGWOT work [3]. In particular,

* The R-FPGWOT loss (Eqs 1 and 2) and "Constrained feasible set" are copied from my VAOT work [5] and my RGWOT work [3]. The only/minor difference here is the partial unbalanced constraint, which was proposed in ASOT [6] for self-supervised action segmentation. More discussions regarding this only/minor difference are below.

* All the remaining modules in Sec 3, i.e., "Regularization using Priors", "Virtual frame for background", "IDM-style structural regularization", "Contrastive stabilization", are exactly copied from VAVA [4] and OPEL [2].

[1] Bansal et al. My view is the best view: Procedure learning from egocentric videos. ECCV 2022. (CnC)

[2] Chowdhury et al. Opel: Optimal transport guided procedure learning. NeurIPS 2024. (OPEL)

[3] Mahmood et al. Procedure learning via regularized gromov-wasserstein optimal transport. WACV 2026. (on ArXiv since July 2025) (RGWOT)

[4] Liu et al. Learning to align sequential actions in the wild. CVPR 2022. (VAVA)

[5] Ali et al. Joint self-supervised video alignment and action segmentation. ICCV 2025. (on ArXiv since March 2025) (VAOT, VASOT)

[6] Xu and Gould. Temporally consistent unbalanced optimal transport for unsupervised action segmentation. CVPR 2024. (ASOT)

[7] Kumar et al. Unsupervised action segmentation by joint representation learning and online clustering. CVPR 2022. (TOT)

[8] Thual et al. Aligning individual brains with fused unbalanced gromov wasserstein. NeurIPS 2022.

Secondly, regarding the only/minor difference (the partial unbalanced constraint), it was proposed by ASOT [6] for self-supervised action segmentation and has been explored in my VAOT work [5] for self-supervised video alignment. Thus, this only difference is **very limited**. Importantly, as we empirically verified in [5] (see results in Sec 4.1 of [5] and discussions in Sec 3.2.1 of [5]), using the partial unbalanced constraint for video alignment doesn't yield performance gains as in action segmentation. This is because the (frame-to-frame) assignment in video alignment is generally more balanced than the (frame-to-action) assignment in action segmentation (the number of frames in a video, e.g., thousands of frames, is much larger than the number of actions in a procedure, e.g., 5-20 actions). Also, using the partial unbalanced constraint will add more loss terms, which makes the optimization and parameter setting harder. Thus, I **highly suspect** the significant performance gains of R-FPGWOT over my RGWOT work [3], which relies on a regularized balanced Gromov-Wasserstein optimal transport formulation and is the closest/direct competitor to this submission. For action segmentation, ASOT [6] has significant performance gains over my TOT work [7], which is understandable since TOT [7] is a simple KOT-based method whereas ASOT [6] relies on the advanced FGWOT. Here, both my RGWOT work [3] and this submission rely on FGWOT. Moreover, since this method is a combination of many previous works, it is inefficient. My RGWOT work [3] is more efficient than OPEL [2] (see Sec. A.3 of [3]), which this submission is built on. Lastly, for action segmentation which assigns frames to actions, it makes sense to use "partial" unbalanced constraint (i.e., many frames can be assigned to one action but not the other way around). However, for video alignment which assigns frames to frames, isn't it more reasonable to use "full" unbalanced constraint which relaxes both terms in L279 (e.g., see [8])? My VAOT work [5] and RGWOT work [3] have explored the unbalanced constraint for video alignment and procedure learning. Thus, this only difference is a very limited.

---

### Public Comment · ~Quoc-Huy_Tran2 · 2025-11-26
**[2/3] I strongly urge ACs to reject this submission.**

Finally, the author(s) selected "optimization" as Primary Area so this submission has less chances to be matched to reviewers who work on video alignment/action segmentation/procedure learning. Also, the author(s) did the same trick before (copying almost exactly the same method for self-supervised video alignment, just adding Part 2 Clustering and Key-step Ordering) and managed to publish "it" (I can't name "it" due to doubly-blind process) at a top-tier Machine Learning conference before. The author(s) didn't submit "it" and this submission to a Computer Vision conference, which is obviously more related. Most, if not all, previous works in these topics are published in Computer Vision conferences like ICCV/CVPR/ECCV/WACV. I would personally even consider that "trick" (copying almost exactly the same method for self-supervised video alignment, just adding Part 2 Clustering and Key-step Ordering, and submitting as a self-supervised procedure learning method) as "**plagiarism**". I wasn't aware of the author(s)' first work (i.e., "it" was published last year), since I started looking into procedure learning in March 2025.

In summary, given that this submission has heavily copied exisiting works, yielding very limited novelty, and ICLR is a top-tier conference, I strongly urge ACs to reject this submission.

P/S: I have been aware of this work since a couple of weeks ago but I wasn't allowed to add any comments until now. I wish I could have been able to add my comments during the review process. I hope ACs can see my comments since it is just **one week** to the release of Final Decisions. If accepted, it will encourage the author(s) to do the same "trick" again and again (as they did before with "it" and now with this submission).

---

### Public Comment · ~Quoc-Huy_Tran2 · 2025-11-26
**[3/3] Heavily copying existing works and Very limited contributions**

*"Concretely, our contribution is instantiated through three key mechanisms: (i) unbalanced partial transport to relax marginal constraints, (ii) a virtual sink node that absorbs unmatched or redundant frames, treating background and idle segments as a structured “null” target rather than forcing spurious alignments, and (iii) Laplace and IDM priors sharpen the transport plan around a near-diagonal procedural ridge, biasing alignments toward time-consistent matches while still handling speed variations and non-monotonic executions."*

(i) unbalanced partial transport. This is a **very limited** and even **questionable** contribution as I have discussed in my comment below.

(ii) virtual sink is **NOT** a contribution. It has been proposed before and used directly in self-supervised procedure learning, i.e., OPEL [2] and RGWOT [3], and self-supervised video alignment, e.g., VAVA [4] and VAOT [5].

(iii) Laplace and IDM priors are **NOT** contributions. They are both proposed/used in OPEL [2] (self-supervised procedure learning), while IDM priors are used in OPEL [2], RGWOT [3] (self-supervised procedure learning), and VAVA [4], VAOT [5] (self-supervised video alignment).

[2] Chowdhury et al. Opel: Optimal transport guided procedure learning. NeurIPS 2024. (OPEL)

[3] Mahmood et al. Procedure learning via regularized gromov-wasserstein optimal transport. WACV 2026. (on ArXiv since July 2025) (RGWOT)

[4] Liu et al. Learning to align sequential actions in the wild. CVPR 2022. (VAVA)

[5] Ali et al. Joint self-supervised video alignment and action segmentation. ICCV 2025. (on ArXiv since March 2025) (VAOT, VASOT)

P/S: I dont have time to go over all comments but it is very clear from the authors description of contributions that the contribution is very limited and this submission has heavily copied existing works and made it sound like they are theirs (i.e., I am **ABSOLUTELY** confident that points (ii) and (iii) are copied from exisiting works in the same topic(s)).

---

### Comment · Area_Chair_bQ8y · 2025-11-26
**Reminder: Discussion Phase Engagement Needed**

Dear Reviewers:

As the deadline for the discussion phase is approaching in less than one week, could you kindly engage in the discussion with the other reviewers and provide your response to the authors’ rebuttal?

Please note that there are public comments available; kindly review them as well and consider them for reference only.

Best regards,

AC

---

### Author Response · Authors · 2025-11-27
**Clarifications on Novelty, Venue Alignment, and Technical Distinction of REALIGN**

**Dear AC and Reviewers,**

We appreciate the opportunity to clarify several points raised in the discussion. We welcome rigorous technical debate and wish to address the concerns regarding the distinctiveness of our contribution and the suitability of the venue.
1. **On Originality and Attribution - Technical Clarifications on R-FPGWOT:**
We firmly reject the implication of appropriating ideas without attribution. We want to clarify that in our submission, we have acknowledged that our work builds upon RGWOT and **explicitly cites** all relevant prior works (including ASOT, OPEL, RGWOT, etc.) as main **baselines**. Our intent is not to replicate prior methods, but to build upon them. We propose a fusion of partial FGWOT, priors, virtual background handling, and contrastive stabilization tailored to procedure learning. To the best of our knowledge, this specific combination and formulation do not appear in the literature, and our empirical results highlight its distinct value over the baselines.

    Regarding the use of known mathematical operators, our core contribution is **how we reformulate and fuse these operators** into an R-FPGWOT objective tailored to procedure learning - empirically and through ablations - yielding more robust procedure structure than existing OT baselines:
    * vs. **OPEL [1]**: OPEL relies on standard KOT, with post-hoc background handling. Our formulation integrates partiality, mass relaxation, a virtual background frame, and annealed priors directly within the FGWOT objective. Ablations show this significantly improves robustness to background frames and temporal jitter.
    * vs. **RGWOT [2]**: RGWOT employs a balanced FGWOT approach. We respectfully argue that video alignment in "wild" procedure learning is inherently unbalanced due to significant variations in length and content density. A strict balanced constraint forces the alignment of every frame, often leading to erroneous matching. Unlike RGWOT, we integrate partiality, similarity-driven priors, and coupled-contrastive stabilization **into the same FGWOT objective** and analyze how that affects alignment and downstream PCM.
    * vs. **VASOT / ASOT [3,4]:** These works utilize unbalanced constraints primarily for action-level assignments. We focus on **procedure-level video-to-video alignment**, where the geometry differs significantly (long, noisy episodes). Contrary to claims in VAOT, our ablations demonstrate that the R-FPGWOT's regularization and constraint formulation provides performance gains over balanced methods in this specific regime.

Formally, while baselines (**A**) like RGWOT optimize a Gromov-Wasserstein objective under strict marginal constraints (balanced transport), our framework introduces **B**, a composite objective that incorporates partial transport constraints (via mass relaxation and virtual sinks), annealed priors and plan-aware contrastive regularization. The resulting mathematical formulation of **(A+B)** alters the optimization landscape. It relaxes the requirement to align every frame -thereby handling background noise - while explicitly enforcing structural consistency through the contrastive term, yielding improved downstream procedure segmentation that outperforms OPEL and RGWOT on standard benchmarks. We claim that this fused formulation for self-supervised procedure learning is **novel, carefully analyzed, and leads to consistent empirical gains.**

2. **On the Primary Area Selection:**
Our core contribution is a Regularized Fused Partial GWOT framework. As the core of the work centers on the **optimization of transport plan design, regularization, and embedding learning**, we believe ICLR is the appropriate venue based on prior submissions seen in this domain, like DO/TGT, VEDIT[5,6].

3. **Clarifying the Role of the PCM Stage:** The PCM (clustering + graph-cut) stage is a standard component, and we **do not** claim novelty. As **acknowledged** in the paper, this stage follows from CnC, OPEL, and RGWOT, and is used explicitly as a downstream post-hoc segmentation module. Our methodological contributions are **the alignment geometry, regularization, and training objective** that produce the transport plans and embeddings feeding into PCM.

4. **Evaluate on Technical Merit**:
We hope this clarification highlights that our *problem formulation* is motivated by specific challenges in procedure learning, and that the *technical integration of partial FGWOT and priors* leads to consistent empirical gains. We have built upon the community's knowledge, properly citing all sources, to push the SOTA forward. We appreciate the reviewers’ time and engagement and hope this clarification is helpful.

Sincerely,

The Authors

---

[1] Chowdhury et al. "OPEL". NeurIPS 2024.

[2] Mahmood et al. "RGWOT". WACV 2026.

[3]Ali et al. "VASOT". ICCV 2025.

[4] Xu and Gould. "ASOT". CVPR 2024.

[5] Luigi et al. "DO/TGT" NeurIPS 2024 (Spotlight)

[6] ⁠Lin, Han, et al. "VEDIT". ICLR 2025

---

> ### Public Comment · ~Quoc-Huy_Tran2 · 2025-11-27
> **[1/4] This submission is an application/engineering paper and should be submitted to Computer Vision venues**
>
> Thank you for the responses.
>
> First, I would like to provide a summary of the literature, since that will help a lot in realizing the limited novelty/contribution of this submission.
>
> * **VAVA [1] (video alignment)**:  includes KOT, temporal prior, optimality prior, virtual frame, constrative regularization (both intra-video and inter-video), balanced transport.
> * **OPEL [2] (based on VAVA [1])**: includes KOT, temporal prior, optimality prior, virtual frame, constrative regularization (both intra-video and inter-video), balanced transport.
> * **VAOT [3] (video alignment)**: includes FGWOT (KOT+GWOT), temporal prior, structural prior, virtual frame, balanced transport.
> * **RGWOT [4] (based on VAOT [3])**: includes FGWOT (KOT+GWOT), temporal prior, structural prior, virtual frame, contrastive regularization (intra-video only as inter-video requires optimality prior), balanced transport.
> * **This submission**: includes FGWOT (KOT+GWOT), temporal prior, optimality prior, structural prior, virtual frame, contrastive regularization (both intra-video and inter-video), partial unbalanced transport. It is merely a combination of the original modules in RGWOT [4] and OPEL [2] and adds the partial unbalanced transport from ASOT [5].
>
> The only novelty here is the partial unbalanced transport, which is **very limited and even questionable** as I have addressed before. All remaining modules are copied **exactly as proposed** in existing video alignment/procedure learning methods. This submission is an **application/engineering paper** and should be submitted to Computer Vision venues.
>
> Now please let me address the responses.
>
> 1.
> *"Formally, while baselines (A) like RGWOT optimize a Gromov-Wasserstein objective under strict marginal constraints (balanced transport), our framework introduces (B), a composite objective that incorporates partial transport constraints (via mass relaxation and virtual sinks), annealed priors and plan-aware contrastive regularization. The resulting mathematical formulation of (A+B) alters the optimization landscape."*
>
> In the (B) above, "virtual sinks", "annealed priors", "plan-aware contrastive regularization" are copied **exactly as proposed** in OPEL [2] (or VAVA [1]). For example, the authors use a new word "annealed" but it is just an existing trick by VAVA [1] and VAVA [1] doesn't call it "annealed". In particular, for the mixing parameter ϕ in Eq 3, the authors follow VAVA [1] to start it with 1.0 and gradually reduce it. The same thing is done for the mixing parameter (it is named ψ though) in VAVA [1], please refer to the text right below Fig 6 caption in the VAVA [1] paper on Open Access, i.e., "ψ ∈ [0, 1] is a weighting parameter that we set to 1.0 initially and decrease gradually over time". There is literally no citation or discussions of VAVA [1] whenever "annealed" is used in this submission.
>
> I notice the authors use many confusing terms like "annealed priors" or "plan-aware contrastive regularization" or "virtual sinks". I haven't seen them being used in previous works. For example, VAVA [1] is the original inventor of all 3 components above but it doesn't use "annealed" or "plan-aware" or "sinks". Why do the authors see the need to have **new names** for **existing modules**? This is what I mean by "The authors make it sound like the modules are proposed by them".
>
> *"We firmly reject the implication of appropriating ideas without attribution. We want to clarify that in our submission, we have acknowledged that our work builds upon RGWOT and explicitly cites all relevant prior works (including ASOT, OPEL, RGWOT, etc.) as main baselines. Our intent is not to replicate prior methods, but to build upon them."*
>
> The authors did cite and discuss very briefly in Sec 2. However, **most importantly**, many modules in Sec 3 **lack citations** of the original works which this submission copies from and lack discussions on how these modules are **different from** those proposed in the original works. For example, there is no citations or discussions of VAVA [1] (or OPEL [2]) in "Regularization using Priors" and "Virtual frame for background". Also, there is no citations or discussions of ASOT [5] (and VAOT [3] since it did experiment with unbalanced transport) in "Constrained feasible set". This seriously affects the understanding/judgement of readers who are not very familiar with the literature.
>
> [1] Liu et al. Learning to align sequential actions in the wild. CVPR 2022. (VAVA)
>
> [2] Chowdhury et al. Opel: Optimal transport guided procedure learning. NeurIPS 2024. (OPEL)
>
> [3] Ali et al. Joint self-supervised video alignment and action segmentation. ICCV 2025. (on ArXiv since March 2025) (VAOT, VASOT)
>
> [4] Mahmood et al. Procedure learning via regularized gromov-wasserstein optimal transport. WACV 2026. (on ArXiv since July 2025) (RGWOT)
>
> [5] Xu and Gould. Temporally consistent unbalanced optimal transport for unsupervised action segmentation. CVPR 2024. (ASOT)

---

> ### Public Comment · ~Quoc-Huy_Tran2 · 2025-11-27
> **[2/4] This submission is an application/engineering paper and should be submitted to Computer Vision venues**
>
> *"We propose a fusion of partial FGWOT, priors, virtual background handling, and contrastive stabilization tailored to procedure learning. To the best of our knowledge, this specific combination and formulation do not appear in the literature, and our empirical results highlight its distinct value over the baselines."*
>
> **All of** "partial FGWOT, priors, virtual background handling, and contrastive stabilization" were proposed by **exisiting works** in the same video alignment/procedure learning domain. Procedure learning first uses video alignment, and then adds clustering and key-step ordering. I consider procedure learning and video alignment as the same topic since graph cut has become the standard for clustering and key-step ordering (as the authors mentioned above). Thus, the combination is **straightforward** and has **little novelty**. Basically, this submission merges RGWOT [4] with optimality prior from OPEL [2] and unbalanced transport from ASOT [5] .
>
> Note that KOT with optimality prior (e.g., VAVA [1] and OPEL [2]) is inferior to FGWOT with structural prior (e.g., VAOT [3] and RGWOT [4]). Thus, I am **not convinced** that mixing (the worse) optimality prior with (the better) structural prior would yield (significant) performance gains. Besides, the optimization will now involve so many losses, priors, regularizations, and constraints, making it **difficult/inefficient** to optimize as well as hard to **reproduce** the results. I highly doubt the results and strongly encourage the authors to release the code for reproducibility.
>
> *"vs. OPEL [1]/ vs. RGWOT [2]/ vs. VASOT (VAOT, not VASOT, as VAOT for video alignment and VASOT for action segmentation / ASOT [3,4]"*
>
> A **clear statement of novelty/contribution** is needed, not just new names for existing modules. For example, "partiality, mass relaxation" (these are the same thing and should be called (partial) unbalanced transport or (partial) unbalanced assignment to be consistent with the original work, i.e., ASOT [5]), or "similarity-driven priors, coupled-contrastive stabilization" (these are all new named which are not used in the original work, i.e., VAVA [1]).
>
> Yesterday I addressed the authors own description of contribution (titled "Heavily copying existing works and Very limited contributions"). Today, after reading the author responses, rechecking this submission and rechecking the literature, I am still **not convinced**, and in fact, I become **more confident** in my reviews/comments on this submission.

---

> ### Public Comment · ~Quoc-Huy_Tran2 · 2025-11-27
> **[3/4] This submission is an application/engineering paper and should be submitted to Computer Vision venues**
>
> 2. *"Our core contribution is a Regularized Fused Partial GWOT framework. As the core of the work centers on the optimization of transport plan design, regularization, and embedding learning, we believe ICLR is the appropriate venue based on prior submissions seen in this domain, like DO/TGT, VEDIT[5,6]."*
>
> **None of** the optimization modules here are proposed by this submission, but existing works in procedure learning/video alignment/action segmentation, which are published at Computer Vision venues like ICCV/CVPR/ECCV/WACV. In my opinion, this is merely an **application/engineering paper** which simply combines exisiting methods in the same topics together with little modification. Therefore, it should be submitted to Computer Vision conferences, and Primary Area should be **more relevant** to procedure learning/video alignment/action segmentation, e.g., video understanding, video recognition, action recognition, action understanding, action segmentation, etc.
>
> DO/TGT and VEDIT are **not directly** relevant. Moreover, DO/TGT and VEDIT **don't even compare** with procedure learning methods (e.g., CnC/GPL/OPEL/STEPs) in their papers. In contrast, **all of** the competing methods in Tab 1, 2, 6 of this submission are **directly** relevant and are published at Computer Vision conferences (with OPEL [2] is the only exception!).
>
> 3. *"Clarifying the Role of the PCM Stage: The PCM (clustering + graph-cut) stage is a standard component, and we do not claim novelty. As acknowledged in the paper, this stage follows from CnC, OPEL, and RGWOT, and is used explicitly as a downstream post-hoc segmentation module. Our methodological contributions are the alignment geometry, regularization, and training objective that produce the transport plans and embeddings feeding into PCM."*
>
> Thank you for **confirming** that the Clustering and Key-Step Order module is indeed copied from existing works. Please make that more obvious by **citing the original work** (i.e., CnC), instead of citing the detailed steps which are already described in CnC and are distracting, and **shortening** that paragraph significantly to include a clear statement of novelty/contribution.

---

> ### Public Comment · ~Quoc-Huy_Tran2 · 2025-11-27
> **[4/4] This submission is an application/engineering paper and should be submitted to Computer Vision venues**
>
> Again, below is the complete list of where all the modules in this submission (Sec 3) are taken from:
>
> * The **"FGWOT formulation"** and **"Constrained feasible set"** are copied from my RGWOT work [4] (or equivalently, my VAOT [3] work). The only/minor difference here is the partial unbalanced transport, which is copied from in ASOT [5] for action segmentation, and is used here with **little modification**.
>
> * All the remaining in Sec 3, namely **"Regularization using Priors"**, **"Virtual frame for background"**, **"IDM-style structural regularization"**, **"Contrastive stabilization"**, are copied **(exactly as they are)** from OPEL [2] (or equivalently, VAVA [1]).
>
> In conclusion, this submission heavily copies and combines existing methods in procedure learning/video alignment/action recognition (i.e., RGWOT, OPEL, ASOT) with little modification and is merely an application/engineering paper and should be submitted to Computer Vision venues (not a top-tier Machine Learning conference like ICLR). At Computer Vision venues (like **all of** the competing methods in Tab 1, 2, 6, with the only exception of OPEL!), this submission will be judged/reviewed **thoroughly**, **truthfully**, and **fairly**.
>
> P/S: I am a co-author of the video alignment work, LAV, which VAVA is built on, and action segmentation work, i.e., TOT, which ASOT is built on. I am also a co-author of VAOT and RGWOT, which this submission is built on. I have been working on optimal transport and its applications in video alignment/action segmentation/procedure learning and I am very familiar with these topics.

---

### Author Response · Authors · 2025-12-02
**Rebuttal to Baseless Plagiarism Allegations and Unprofessional Public Commentary**

To put things into perspective, we want to be absolutely clear that we **categorically reject the commenter’s allegation of plagiarism and the narrative they try to construct around our work**. Their public remarks are **completely unfounded** and repeatedly **misrepresent what we claim, what we cite, and what we actually contribute**, often overlooking explicit attributions and clarifications that are already present in the manuscript. We therefore urge the area chair, reviewers, and any readers to **check the manuscript directly rather than taking these comments at face value**: the citations, baselines, and contributions are all laid out in full, and any alleged “discrepancies” can be resolved by reading the method and results sections rather than relying on a one-sided public commentary.

**Below, we spell out the main issues with the commenter’s statements and behavior:**

1.  We **strongly reject** the allegation of **plagiarism.** We **do not** claim to have introduced the core modules introduced in VAVA [1], OPEL [2], ASOT [3], and RGWOT [4]. These works are **explicitly cited** at the exact points where they are used as primary baselines and building blocks throughout the method section in our paper (e.g., **L237, L252, L264, L269, L285, L304, L311, L346**).

2. Some of the **language used in the public comments** (e.g., repeated assertions of “heavily copied,” “merely engineering,” and prescriptive statements about where our work “should” be submitted) is, in our view, plainly dismissive and unprofessional, shifting the tone from any **genuine attempt at constructive feedback or legitimate scientific critique to a needlessly hostile, disparaging, and intimidating commentary.** It is **inconsistent and unfair** that the authors use a similar level of attribution in one of their own papers, RGWOT [4], and then characterize a similar practice when it comes to other people’s work as “heavily copied”.

3. In addition, the commenter has explicitly identified us and our prior work by name on a major OpenReview platform. This is not a minor oversight but a **flagrant breach of the double-blind review process** at ICLR, openly exposing the authors’ identities and tying the submission to specific individuals in a manner that is incompatible with the standards expected of a serious peer-review venue.

4. The commenter is not only a leading author of many of the baseline methods we build upon, but also the **founder of a startup operating in the _same research domain_**. This role raises a clear **conflict-of-interest concern**, particularly when he publicly and repeatedly dismisses closely related work as “trivial” or “unsuitable” for the venue. In this context, the tone and pattern of their remarks create at least a strong appearance of **personal attack or competitive bias** rather than a neutral, good-faith assessment of the work. At the bare minimum, such commentary should be accompanied by explicit disclosure of these interests and framed in a manner consistent with impartial and professional scientific evaluation.

We request that the area chair, program chairs, and the OpenReview platform **examine this behavior for possible policy violations or reviewer misconduct**, and weigh the commenter’s remarks in that context. We trust the committee to form an independent assessment of our submission based on its technical content, empirical evidence, and compliance with ICLR’s review standards.

Below, we serially rebut each of the commenter’s points with detailed, evidence-based arguments, showing that the **foundations of many of their claims are inaccurate**, and we provide concrete proof to support this.

Sincerely,

The Authors

---

> ### Author Response · Authors · 2025-12-02
> **1. Wrongful allegations of plagiarism**
>
> We **strongly reject** the allegation of **plagiarism** and the narrative implied by the commenter. We **do not** claim to have invented the modules from VAVA [1], OPEL [2], ASOT [3], or RGWOT [4].
> * In Section 2, we clearly summarize prior works and position our work as an _extension_ of these methods.
> * In Section 3, the priors, virtual frame, and structural terms are **explicitly cited** back to the corresponding prior works at the exact points (e.g., **L237, L252, L264, L269, L285, L304, L311, L346**). Our partial FGW-based formulation is described as _built on top of_ these modules.
> * We **do not reuse any code**. Public codebases for OPEL, VASOT, and RGWOT are **not available**, and our implementation is our own, following only the conceptual formulations in the literature. Code will be made public upon acceptance for verification.
>
> For clarity, we briefly summarize the existing literature and how these works relate:
> *  **VAVA [1]** - KOT + Gaussian temporal and optimality priors + virtual frame for background + C-IDM regularization for self-supervised video alignment.
> *  **OPEL [2]** - VAVA + inter-video contrastive + clustering/graph-cut for unsupervised phase discovery and ordering.
> *  **ASOT [3]** - FGWOT (KOT+GWOT) + unbalanced OT + structural prior for **unsupervised action segmentation**, i.e., dense frame-wise labeling rather than procedure-level key-step discovery.
> *  **VAOT / VASOT [5] – (Commentor’s work)** - ASOT (FGWOT) + Balanced OT + VAVA (temporal prior + structural priors + virtual frame) for action segmentation.
> *  **RGWOT [3] – (Commentor’s work)** ASOT (FGWOT) + VAVA (temporal prior + structural priors + virtual frame + C_IDM) + CNC/OPEL (Graphcut) for self-supervised video alignment.
>
> There is, by now, a **shared toolbox** in this line of work—FGWOT, temporal and optimality priors, virtual frames, contrastive terms, temporal constraints—that has been repeatedly adapted and recombined across VAVA → OPEL, ASOT → VAOT/VASOT, RGWOT, and other papers, including **several authored by the commenter themselves**.  Using established modules with clear citation and proposing a new combination/extension within this toolbox is standard practice in this area and **does not constitute plagiarism**. We respectfully leave the assessment of this point to the area chair and reviewers, based on the actual manuscript rather than the commenter’s characterization.
>
> [1] Liu et al. Learning to align sequential actions in the wild. CVPR 2022. (VAVA)
>
> [2] Chowdhury et al. Opel: Optimal transport guided procedure learning. NeurIPS 2024. (OPEL)
>
> [3] Xu et al.. Temporally consistent unbalanced optimal transport for unsupervised action segmentation. CVPR 2024. (ASOT)
>
> [4] Mahmood et al. Procedure learning via regularized Gromov-Wasserstein optimal transport. WACV 2026. (RGWOT)
>
> [5] Ali et al. Joint self-supervised video alignment and action segmentation. ICCV 2025. (VASOT)

---

> ### Author Response · Authors · 2025-12-02
> **2. Addressing Misleading Claims regarding Novelty of Work**
>
> Our goal is squarely in that same tradition: to **assemble, adapt, and extend** this toolbox, not to claim novelty on individual primitive modules. Our contribution lies in moving to a **partial/unbalanced OT regime within an FGWOT framework**, i.e., coupling partial/unbalanced relaxation with temporal/optimality and structural priors in the context of **unsupervised procedure learning.**
>
> In this light, characterizing our work as *“heavily copied,” “merely engineering,” or only suitable for “CV venues”* is **unfounded and selectively framed**. Most of their works, including VASOT and RGWOT, are themselves **engineering and methodological adaptations of earlier ideas**. It is therefore **inconsistent** to treat our method as categorically “non-novel” simply because it, too, builds on an established and clearly cited toolbox—precisely the pattern followed by VASOT and RGWOT when extending earlier OT-based methods.
>
> Assessments of *“how much novelty is enough”* are inherently **subjective**, and should rest with the **independent area chair and reviewers** based on a direct reading of the manuscript, rather than on a one-sided public commentary whose language risks **skewing the perception** of the paper.
>
> Statements such as _“I am not convinced that mixing (the worse) optimality prior with (the better) structural prior would yield (significant) performance gains. ... making it difficult/inefficient to optimize as well as hard to reproduce the results”_ are made without engaging with our **actual empirical evidence** (L377), the detailed explanations in the paper (L382-386), or our prior rebuttals. In our view, this kind of rhetoric **disregards the documented results and ablations we provide** and reads less like a careful scientific concern and more like an attempt to **undermine and delegitimize our contribution by assertion rather than the argument**. We request that the area chair and reviewers treat such remarks with appropriate caution and evaluate our work on the basis of the technical record we have provided.

---

> ### Author Response · Authors · 2025-12-02
> **3. Clarifying the use of misleading terminology**
>
> Our use of phrases such as “annealed priors,” “plan-aware contrastive regularization,” and “virtual sinks” is entirely **terminological** and aligned with existing literature, with only minor wording changes to make the roles of these components clearer in our setting—not to rebrand prior ideas or claim ownership of them:
> *  **“Annealed priors”** refers to the gradual decrease of a mixing weight between priors, i.e., the same annealing mechanism already used in VAVA. We follow that definition verbatim in our paper (see, e.g., **L264**).
> *   **“Plan-aware contrastive regularization”** is simply a descriptive label indicating that our contrastive term is constructed using the *current OT plan*, extending the contrastive formulations already used in VAVA/OPEL for the FPGWOT setting. As cited at **L252**, we do not claim a new conceptual object.
> * **“Virtual sinks”** is an informal shorthand for the **virtual background frame** introduced in VAVA/OPEL; **L269** explicitly explains this connection and our rationale. If needed, we can revert entirely to the original phrasing “virtual frame for background” to make this continuity even more explicit.
>
> In other words, the terminology in our paper stays within the conventions established by prior work and only adapts naming slightly to highlight the specific role of these modules in our formulation. This is why the statement *“The authors make it sound like the modules are proposed by them”* is an **overexaggeration and misrepresents the actual contributions of our work.**

---

> ### Author Response · Authors · 2025-12-02
> **4. Venue suitability and repeated requests to reclassify the work**
>
> Repeated public instructions to “submit this to CV venues instead” create an unintentional bias in the evaluation process. We request that the question of venue suitability be left to the **area chair and reviewers**, who can judge whether the balance of optimization/OT content and application is appropriate for ICLR.
> * The core of the paper is a **regularized fused partial GWOT** objective and its **optimization**: we design a transport geometry, priors, and regularization that jointly shape the learned representations and alignment plans.
> * The evaluation is on procedure datasets that originate from the Computer Vision community, but the main technical content is an **OT-based learning framework for unsupervised procedural alignment**, which aligns with long-standing interests at ICLR in optimal transport, representation learning, and structured prediction.
>
> Many ICLR/NeurIPS papers have a separate primary area on **"OPTIMIZATION"**, and several authors have proposed optimization and representation learning methods evaluated on perception tasks (vision, video, speech, etc.) in this area. Our submission is in that spirit. We do not claim it would be inappropriate for CV venues; rather, we see it as equally relevant to the ML community due to its focus on transport objectives and regularization, not on architecture design or dataset curation alone.

---

### Meta-Review · Area_Chair_Z9mW · 2026-01-08

**Summary:**

The paper presents REALIGN, a framework that leverages optimal transport to align procedural videos and identify key steps. REALIGN demonstrates highly competitive performance on public benchmarks. The initial reviews were mixed. Reviewers found the approach interesting and praised the strong empirical results, while also raising several concerns. The main issues shared across reviews included: (1) the need for better positioning in the literature and clarification of the technical novelty; (2) concerns that the two-stage design for step discovery may lead to sub-optimal solutions; and (3) the complexity of the optimization objective, which involves multiple hyperparameters and may lead to sensitivity in performance. Reviewers also requested clarification of several technical components. In the rebuttal and revision, the authors provided additional details and included further experimental results, which helped address some of these concerns. However, two main concerns persisted: positioning and technical novelty, and sensitivity to hyperparameters.

The AC notes that during the discussion phase, a public commenter also raised questions regarding technical novelty, although these comments were not explicitly considered in the decision process.

After going through the paper, the review, and the rebuttal, the AC recommends rejection. This decision is based on the outstanding concerns as described below. The authors are encouraged to incorporate review feedback, revise the paper, and re-submit to a different venue.

**Reviewer Concerns:**

After reviewing the paper, reviews, and rebuttal, the AC identified two remaining concerns.

**Positioning and technical novelty.**
The authors revised the paper to better contrast REALIGN with prior work (e.g., RGWOT) and further clarified the technical novelty in the rebuttal. While the AC does not have deep expertise in this specific topic, it appears that several core components, such as unbalanced transport, virtual sinks, Laplacian and IDM priors, have been explored in prior work (e.g., ASOT, OPEL, and RGWOT), which aligns with the main concern raised by Reviewers ZmuG and 3MWu. Although the particular combination is novel and leads to strong empirical performance, it remains somewhat unclear whether the proposed work offers new insights into the underlying optimization problem beyond existing approaches.

**Sensitivity to hyperparameters.**
The authors added a sensitivity analysis for key hyperparameters (Figure A6 in the appendix). While the authors argue that REALIGN "exhibits stable behavior across the explored ranges," the reported results show notable variability for certain parameters on some datasets. For example, varying the Gromov-Wasserstein weight ($\alpha$) leads to approximately 10% absolute variation in F1 scores across three datasets. Similarly, the contrastive regularization weight ($c_2$) and the number of key steps ($K$) result in over 5% absolute variation in F1 scores across datasets, and on EGTEA-Gaze+ the marginal constraint penalty ($\tau$) shows fluctuations exceeding 15% in absolute values. These results suggest that the robustness claims may require further evidence.

**Reviewer Scores:**

Considering the potential influence of the public comments, it is difficult to predict how reviewers may update their scores. Based on the reviews, rebuttal, and revision alone, most reviewers are likely to maintain their current ratings due to the outstanding concerns.

---

### Decision · Program_Chairs · 2026-01-26

Reject